# On the information content in linear horizontal delay gradients estimated from space geodesy observations

Gunnar Elgered[1], Tong Ning[2], Peter Forkman[1], and Rüdiger Haas[1]

[1]Department of Space, Earth and Environment, Chalmers University of Technology, Onsala Space Observatory, SE-439 92 Onsala, Sweden.
[2]Lantmäteriet (The Swedish Mapping, Cadastral and Land Registration Authority), SE-801 82, Gävle, Sweden

**Correspondence:** Gunnar Elgered (gunnar.elgered@chalmers.se)

**Abstract.** We have studied linear horizontal gradients in the atmospheric propagation delay above ground-based stations receiving signals from the Global Positioning System (GPS). Gradients were estimated from 11 years of observations from five sites in Sweden. Comparing these gradients with the corresponding ones from the European Centre for Medium-Range Weather Forecasts (ECMWF) analyses shows that GPS gradients detect effects over different time scales caused by the hydrostatic and the wet components. The two stations equipped with microwave absorbing material below the antenna in general show higher correlation coefficients with the ECMWF gradients compared to the other three stations. We also estimated gradients using 4 years of GPS data from two collocated antenna installations at the Onsala Space Observatory. Correlation coefficients for the east and the north wet gradients estimated with a temporal resolution of 15 minutes from GPS data can for specific months reach up to 0.8 when compared to simultaneously estimated wet gradients from microwave radiometry. The best agreement is obtained when an elevation cutoff angle of 3° is applied in the GPS data processing, in spite of the fact that the radiometer does not observe below 20°. We also note a strong seasonal dependence in the correlation coefficients, from 0.3 during months with smaller gradients to 0.8 during months with larger gradients, typically during the warmer, and more humid, part of the year. Finally, a case study using a 15-day long continuous Very Long Baseline Interferometry (VLBI) campaign was carried out. The comparison of the gradients estimated from VLBI and GPS data indicates that a homogeneous and frequent sampling of the sky is a critical parameter.

## 1 Introduction

Space geodetic techniques, where the fundamental observable is a radio signal's time of arrival at a station on the surface of the Earth, are affected by variations in the propagation velocity in the atmosphere. Because time measurements avoid problems related to accurate calibration, which are common for systems measuring different types of emissions, it is a common view that Global Navigation Satellite Systems (GNSS) have a long term stability and are well suited for climate monitoring, e.g. in terms of the atmospheric water vapour content. Estimates of the total propagation delay above a GNSS station can be used to determine the integrated amount of water vapour. It is also common practice to estimate two-dimensional horizontal linear gradients for each station in the GNSS data processing, because it improves the reproducibility of estimated geodetic parameters, see e.g. (Bar-Sever et al., 1998).

We have studied estimated gradients primarily from GPS data from Swedish GNSS stations by comparing these gradients to independent measurements. An important site is the Onsala Space Observatory where a geodetic Very Long Baseline Interferometry (VLBI) telescope and a water vapour radiometer (WVR) are installed and collocated with GNSS receiver stations. The overall goal was to study the usefulness of GPS-derived gradients in atmospheric and climate research. Previous studies have been carried out using GPS/GNSS data from Onsala. Comparing the gradients derived from VLBI, GPS, and a WVR, Gradinarsky et al. (2000) found that when varying the constraint for the gradient variability from 0.2 to 5.6 mm/$\sqrt{\text{h}}$ the weighted root-mean-square (RMS) difference compared to the WVR gradients varied between 0.8 and 1.0 mm for both the GPS and the VLBI gradients. Using multi-GNSS observations, Li et al. (2015) found a significant increase in the correlation coefficient to about 0.6 when compared to ECMWF gradients, while the one for the GPS only was typically below 0.5. In addition, they found that the RMS difference of the gradient was reduced to about 25–35 % by multi-GNSS processing.

There are some interesting questions actualised by previous work which we tried to take further. Of specific interest in our study was to investigate if there is any systematic seasonal behaviour in the estimated gradients in Sweden, and if they can be explained by the influence of regional-scale weather systems. The question about the seasonal changes of gradients was previously studied by Koulali et al. (2012). Another issue is that comparisons of estimated GPS gradients with a high temporal resolution are rather sparse, and have to our knowledge so far not covered periods of many years. Here we report on comparisons between GPS and WVR gradients, with a temporal resolution of 15 minutes, over a more or less continuous 4-year period. With such a resolution it is for example possible to study convective systems (Brenot et al., 2013) and the relation between the temporal variability of the gradients and the zenith wet delay (ZWD) during the passage of weather fronts.

In Section 2 we give a short background on the cause of gradients that are sensed by the space geodetic techniques and the model used to estimate them. In Section 3 instruments, techniques, and their data are described. The results are presented in two sections. First, in Section 4, we compare 11 years of total gradients from five Swedish GNSS stations to gradients originating from the European Centre for Medium-Range Weather Forecasts (ECMWF) analyses. Here we study seasonal dependence. In Section 5, we use data from two collocated GNSS stations (with different antenna installations) and one WVR to assess the station performances and differences between different GPS processing variants. We also study the seasonal dependence of the estimated wet gradients over a 4-year period. Finally, within this 4-year period a 15-day long VLBI campaign occurred which we use as a case study. In Section 6 we present our conclusions and suggest possible future studies of gradients.

## 2  Cause of horizontal gradients and models

The delay of space geodetic signals propagating through the atmosphere depends of the refractive index. For space geodetic applications it is meaningful to define one hydrostatic and one wet component (Davis et al., 1985). For a horizontally stratified atmosphere it is then common practise to use equivalent zenith values for these components. Additionally we may define a horizontal linear gradient, that can be inferred from ground-based observations (Davis et al., 1993), consisting of one east and one north component, which in turn also can be separated, into one hydrostatic and one wet component.

Hydrostatic gradients are determined by pressure and temperature gradients and exist mainly over regional scales (e.g. persistent high and low pressure systems) and synoptic scales (e.g. weather systems). Using a European and a global GPS network, including three of the GPS stations used in this study, Meindl et al. (2004) have shown that the north gradient has a clear dependence on latitude when averaged over long time scales. For the area of interest in this study we specifically mention the Icelandic low pressure system that typically evolves in the winter and disappears in the summer (Hewson and Longley, 1944). This is a component in the North Atlantic Oscillation and the Arctic Oscillation (Thompson and Wallace, 1998; Sanchez-Franks et al., 2016).

Temperature and especially water vapour can show strong horizontal gradients over small (kilometre) scales and the temporal variability is typically also much higher than that of the hydrostatic gradients, see e.g. Li et al. (2015). Hence, the large local gradients over a station are mainly caused by the variability in water vapour and the wet refractivity. Gradients can be significant during a passage of a weather front, e.g. Kačmařík et al. (2018) report gradient amplitudes of up to 3–4 mm during the passage of an occlusion front over Germany. Nahmani et al. (2019) have studied gradients during the passage of mesoscale convective systems in West Africa and Koulali et al. (2012) have shown correlations between gradients and precipitation and moisture fluxes in Morocco. Other specific weather phenomena that can cause horizontal variability in the partial pressure of water vapour, and hence also the wet refractivity, are sea breeze (Craig et al., 1945; Miller et al., 2003), cloud rolls (Brown, 1970) and convection processes in general.

We note that none of the known processes is expected to be strictly horizontally linear, but the strength in the geometry, the distribution of the observations on the sky, and the GNSS data quality makes it difficult to determine additional atmospheric parameters of higher order.

The atmospheric parameters that are normally estimated when processing space geodesy data are an equivalent zenith wet delay and linear horizontal delay gradients in the east and the north directions. The uncertainties of the estimates depend on the geometry of the observations and the accuracy of the so called mapping functions, used to describe the estimated parameters dependence on the elevation angle, given the specific weather conditions at the site, at the time, see e.g. Boehm et al. (2006) and Kačmařík et al. (2018). The common model used to relate the observed delay along the line-of-sight, $\Delta L(\alpha, \varepsilon)$, and the estimated parameters (IERS Conventions, 2010) is also used in this study, i.e.

$$\Delta L(\alpha,\varepsilon) = m_h(\varepsilon)\,\Delta L_{hz} + m_w(\varepsilon)\,\Delta L_{wz} + m_g(\varepsilon)\,[\Xi_e \sin\alpha + \Xi_n \cos\alpha] \tag{1}$$

where $m_h$, $m_w$, and $m_g$ are the mapping functions, depending on the elevation angle $\varepsilon$, for the hydrostatic and the wet delays, and the gradients, respectively; $\Delta L_{hz}$ and $\Delta L_{wz}$ are the equivalent hydrostatic and wet delays in the zenith direction; $\alpha$ is the azimuth angle, measured clockwise from the north, implying that $\Xi_e$ and $\Xi_n$ are the gradients in the east and in the north directions. While total gradients are estimated, they can be interpreted as the sum of hydrostatic and wet components as well. In the following we will subtract the hydrostatic component computed from ECMWF from the total GPS gradient to get the GPS wet gradient.

In addition to the east and the north gradient components we also studied the gradient amplitude, defined as

$$|\Xi| = \sqrt{\Xi_e^2 + \Xi_n^2} \tag{2}$$

The gradient amplitude is defined for the hydrostatic, the wet, and the total gradients.

## 3   Instrumentation and data

We compared gradients estimated from GPS observations acquired at five sites and six antenna/receiver installations: Kiruna (KIR0), Mårtsbo (MAR6), Borås (SPT0), Visby (VIS0), and Onsala (ONSA and ONS1) with respect to VLBI, WVR, and ECMWF estimates. These stations are also part of the EUREF network (Bruyninx et al., 2012). Their geographic locations are shown in Figure 1. In this section we first describe the different datasets. Thereafter, we summarise their use and characterise them in terms of formal errors, advantages, and disadvantages.

### 3.1   GPS

We used 11 years of GPS data (2006–2016) from the five Swedish GNSS sites mentioned above. Gradients in the east and the north directions were estimated with a temporal resolution of 5 min. Two GNSS stations are operating continuously at the Onsala Space Observatory, on the west coast of Sweden. The primary station, ONSA, was established already in 1987 and the other station, ONS1, was taken into operation in 2011. The six antenna installations are shown in Figure 2. The antennas of ONSA and ONS1 are located within 100 m from each other and should observe almost identical atmospheric gradients. For the time period 2013–2016 we compared gradients from these two stations with simultaneously estimated gradients using data from a WVR.

The analysis of the GPS data followed the same lines as described by Ning et al. (2013) and is summarised in Table 1. Specifically we mention that each day was analysed independently after adding 3 h of data from the previous day and 3 h from the following day, i.e. in total 30 h. The reason was to avoid discontinuities at midnight in the estimated time series.

In order to investigate the impact of different constraints on the estimated gradients we also reprocessed two days of GPS data for ONSA where large changes (2–3 mm) in both the east and the north gradient components were observed over a couple of hours by both the GPS and the WVR data. In addition to the constraint value of 0.3 mm/$\sqrt{\mathrm{h}}$ suggested by Bar-Sever et al. (1998), the values 0.6, 0.9, and 1.2 mm/$\sqrt{\mathrm{h}}$ were used. The use of these different values shows only small differences compared to the WVR in the daily averaged gradient amplitudes, from 0.02 mm to 0.12 mm, although the short term variability in the GPS gradients increases when weaker constraints (larger values) are used. This is consistent with the result presented by Gradinarsky et al. (2000) for the same site. We note that by using a stronger constraint (small value) we remove the possibility to follow rapid variations in the gradients but at the same time reduce unwanted noise to be absorbed into the gradient estimates. This is a possible explanation why differences between GPS and WVR gradients are less sensitive to the constraint value used in the GPS processing. The impact of the constraint value is further discussed in Sections 5.2 and 6.

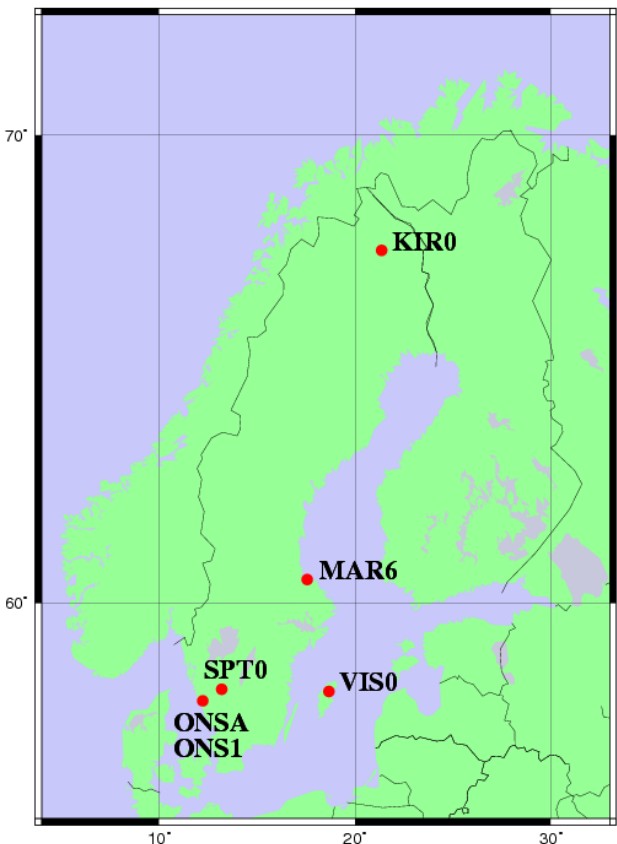

**Figure 1.** The five sites used in the study. Two antenna installations, ONSA and ONS1, are collocated together with the VLBI telescope and the WVR at the Onsala site. An antenna installation is referred to as a station.

Recent work by Kačmařík et al. (2018) compared estimated gradients with those from a numerical weather model using different gradient mapping functions and elevation cutoff angles. They found the best agreement for an elevation cutoff angle equal to 3°. They also showed that the Bar-Sever et al. (1998) gradient mapping function resulted in 17 % smaller gradient amplitudes compared to the Chen and Herring (1997) mapping function. For the 11-year study presented in the next section we used a 10° elevation cutoff angle only, whereas we used several different elevation cutoff angles in the comparison with the WVR data from the Onsala site for a 4-year period.

Based on the five-minute gradients we calculated mean values over 15 min, 6 h, 1 day, and 1 month in order to match the temporal resolution of the comparison data and to study the variability of the wet and the hydrostatic gradients over different time scales.

Examples of the sky coverage of the GPS observations are shown in Figure 3 for the Onsala site. At this latitude there is a significant part of the sky that is never sampled, just north of the zenith direction. It is reasonable to assume that this will have a negative impact on the estimated gradients, and especially in the north direction.

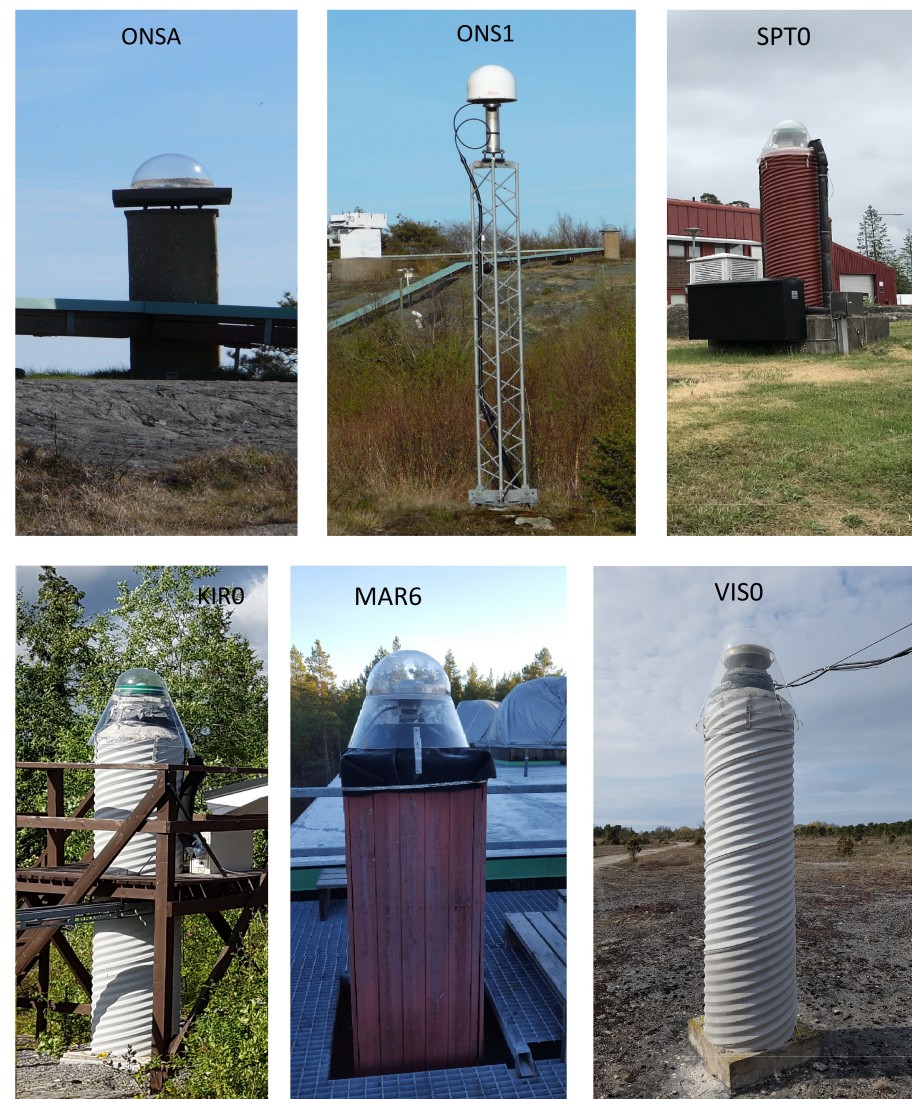

**Figure 2.** The six antenna installations used to acquire the GPS data. See Figure 1 for their geographical location.

**Table 1.** Processing of GPS data.

| Parameter | Description / Value |
| --- | --- |
| Processing software | GIPSY v6.2 (Webb and Zumberge, 1993) |
| Strategy | Precise Point Positioning (Zumberge et al., 1997) final orbit and clock products were provided by JPL obtained from the legacy GIPSY-OASIS software[a] |
| Reference frame | IGS08 |
| Mapping functions for $\Delta L_z$ | Vienna 1 2006 (VMF1) (Boehm et al., 2006)[b] |
| Mapping function for $\Xi$ | Bar-Sever et al. (1998) |
| Elevation cutoff angle | $10^{\circ}$ [c] |
| Zenith delay | Estimated every 5 min, constraint 10 mm/$\sqrt{h}$ (Jarlemark et al., 1998) |
| Linear horizontal gradient | Estimated every 5 min, constraint 0.3 mm/$\sqrt{h}$ (Bar-Sever et al., 1998) |
| Ocean tide model | FES2004 (Lyard et al. , 2006) |
| Antenna phase centre | igs08_1740.atx (Schmid et al., 2007)[d] |
| Ambiguity resolution | Yes (Bertiger et al., 2010) |
| Ionosphere model | 2nd order (IGRF)[e] (Matteo and Morton, 2011) |

[a] For the 11-year dataset, for the 4-year dataset, the products were obtained from a new GipsyX software. We noted that the difference in the products due to the change of software is small (Sibois et al., 2017).

[b] For the 11-year dataset, for the 4-year dataset also the weighted ($\sin(\varepsilon)$) VMF1 and the NMF (Niell, 1996) were used.

[c] For the 11-year dataset, for the 4-year dataset also $3^{\circ}$ and $20^{\circ}$ were used.

[d] For the 11-year dataset, for the 4-year dataset igs08_1869.atx were used.

[e] International Geomagnetic Reference Field

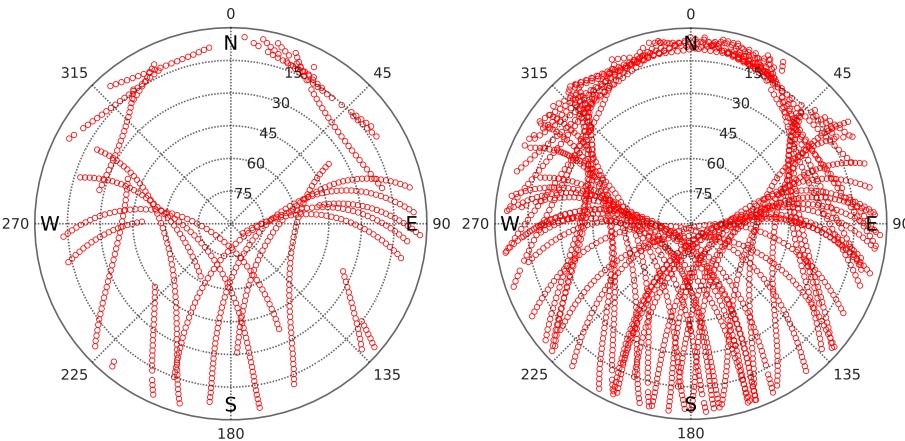

**Figure 3.** Sky plots of GPS observations at Onsala from 6 to 12 UT (left) and from 0 to 24 UT (right) on May 12, 2014. This particular day was chosen because it is included in the CONT14 campaign presented in Section 5.3. The sky distribution of observations is very similar, although not identical, for all days.

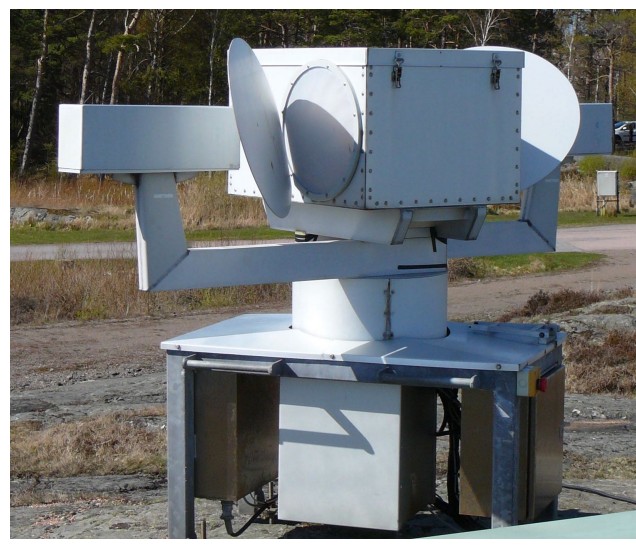

**Figure 4.** The water vapour radiometer (WVR) Konrad at the Onsala Space Observatory.

## 3.2 Microwave radiometer

The microwave radiometer, shown in Figure 4, was designed in order to provide independent estimates of the wet propagation delays for space geodetic applications. It measures the emission from the sky, on and off the water vapour line at 22.2 GHz. Its specifications are summarised in Table 2 and the data processing was carried out as was described for another WVR by
Elgered and Jarlemark (1998).

During the time period 2013–2016 the WVR was observing in a sky mapping mode as is illustrated in Figure 5. A disadvantage of a WVR is that the algorithm for calculation of the wet propagation delay fails for data acquired during rain or when large liquid drops are present in the sensed atmosphere. Typically such conditions imply large positive errors in the wet delay, and the water vapour content (Westwater and Guiraud, 1980). Therefore, data taken during rain, or when the estimated equivalent
amount of liquid water in the zenith direction was $> 0.7$ mm, were discarded from the gradient analysis. In addition there were also time periods when the WVR hardware has failed. The amount of analysed data are shown in Figure 6 as the number of individual observations per day. The first long data gap, in 2014–2015, was caused by a broken mechanical waveguide switch and the second long gap, in 2015–2016, was due to broken cables in the so called cable wrap. As a consequence the cable wrap was redesigned to avoid similar failures in the future.

In order to avoid ground-noise pickup the WVR provided observations of the wet delay in the different directions above $20°$. Therefore a simple $\sin(\varepsilon)$ mapping function was used to relate these slant wet delays to the equivalent ZWD. The WVR gradients were estimated based on all observations carried out during a period of 15 min using the method of least squares and the Bar-Sever gradient mapping function. We used a four-parameter model, fitting a ZWD, a ZWD rate, and an east and a north gradient to the data (Davis et al., 1993). This means that the estimated gradients are independent of the successive estimates,
which is different from the gradients estimated from the space geodetic techniques, where temporal constraints are applied.

**Table 2.** Specifications for the Konrad WVR.

| Parameter | Value |
|---|---|
| Frequencies | 20.6 GHz and 31.6 GHz |
| Antenna type (one for each channel) | Conical horn with lens |
| Antenna beam FWHM$^a$, E-plane, ch.1 / ch.2 | 2.9°/ 2.0° |
| Antenna beam FWHM$^a$, H-plane, ch.1 / ch.2 | 3.4°/ 2.3° |
| Reference temperatures (both channels) | 313 and 373 K |
| System noise temperatures, channel 1 / 2 | 450 / 550 K |
| RF bandwidth (double sideband) | 320 MHz (both channels) |
| Absolute accuracy (weather dependent due to the quality of tip curves) | 1–3 K |
| Repeatability | 0.1 K |

[a] FWHM = Full Width Half Maximum

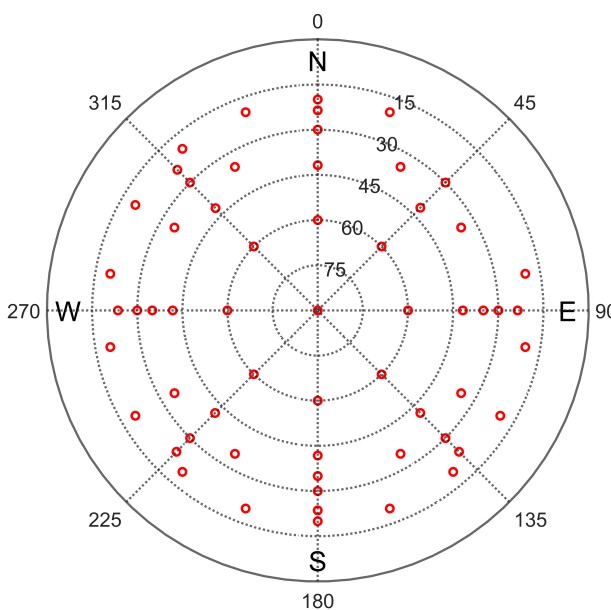

**Figure 5.** A measurement cycle of the WVR begun with two azimuth scans. In order to avoid emission from the ground the lowest elevation angle observed was $20°$. Starting in the north, first turning at an elevation angle of $20°$ clockwise to the north (excluding the azimuth angles of $40°$ and $60°$ due to a nearby radio telescope), and then turning counterclockwise at an elevation angle of $35°$. Thereafter four tip curves were made over the zenith direction (implying four observations in the zenith direction during each cycle): from the north to the south, from the southwest to the northeast, from the east to the west, and from the northwest to the southeast. The cycle was about 8 min long and was repeated continuously, implying that almost two complete cycles with a total of $\approx 100$ observations were used when estimating gradients every 15 min.

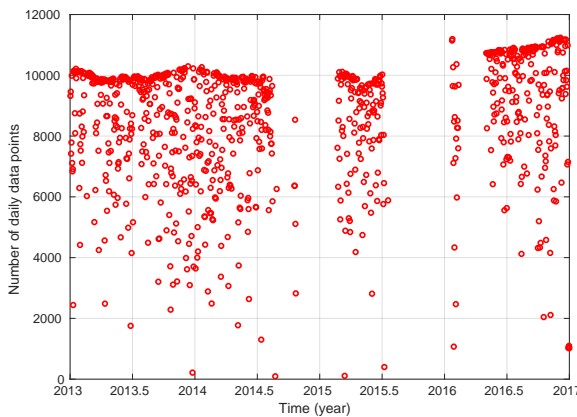

**Figure 6.** Number of data points per day observed by the WVR. During days without data loss, e.g. due to rain, each estimated gradient was based on $\approx 100$ observations in the directions illustrated in Figure 5. Observations close to the sun were removed from the raw data before the data analysis was carried out which causes the seasonal variation in the maximum number of observations per day. During the last year the measurement cycle was optimised by reducing some of the time delays inserted between samples but the observational sequence shown in Figure 5 was used during the whole period.

### 3.3 Very long baseline interferometry

We used the VLBI data from the CONT14 campaign coordinated by the International VLBI Service (Nothnagel et al., 2017). The IVS organises continuous (CONT) VLBI campaigns every third year in order to acquire state-of-the-art VLBI data over a time period of two weeks and to demonstrate the highest accuracy of which the current VLBI system is capable. The primary goal of these CONT campaigns is to support research concerning high resolution Earth rotation (Haas et al., 2017), reference frame stability, and daily to sub-daily site motions, but also other aspects. A concise overview of the IVS CONT campaigns is given by MacMillan (2017).

The CONT14 campaign was observed during May 6–20, 2014. The VLBI data were analysed with the calc/solve analysis software (Ma et al., 1990). Station positions, ZWD, atmospheric gradients, relative clock parameters w.r.t. a reference station, as well as earth rotation parameters were estimated. The relative clock parameters were estimated as a piecewise linear functions every hour, with a constraint of $5 \cdot 10^{-14}$ s/s between clock rate segments. The ZWD and atmospheric gradients were estimated as piecewise linear functions (i.e. not stochastic processes) with a temporal resolution of 30 min and 6 h, respectively. Constraints for the variability of 15 mm/h for the ZWD rate segments, and 2 mm/day for gradient rates were applied. The NMF (Niell, 1996) mapping functions for ZWD and the Chen and Herring (1997) mapping function for gradients were used in the analysis, together with meteorological information recorded at the VLBI stations. An elevation cutoff angle of $5°$ was used, and no elevation-dependent weighting.

Figure 7 depicts the sampling of the sky for a 6 h period, which is the highest temporal resolution of the gradient estimates from VLBI, as well as all observations scheduled for a 24 h experiment. This schedule was repeated every day with only minor modifications.

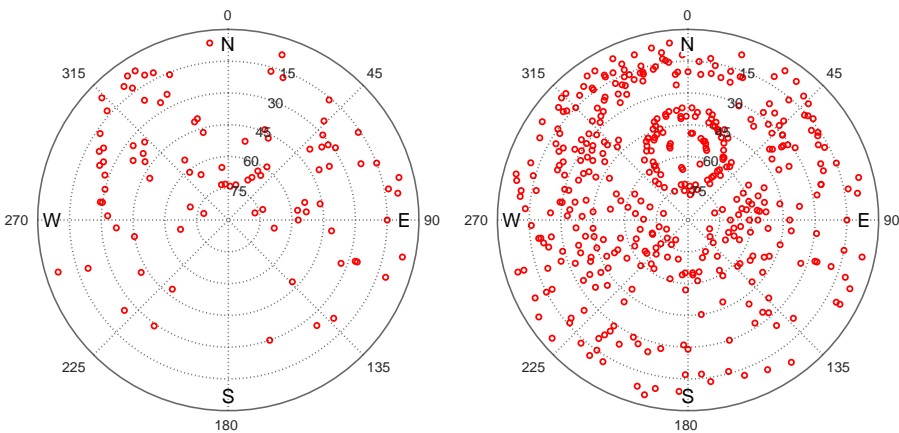

**Figure 7.** The directions of the VLBI observations for the time period from 6 to 12 UT (left) and from 0 to 24 UT (right), both on May 12, 2014.

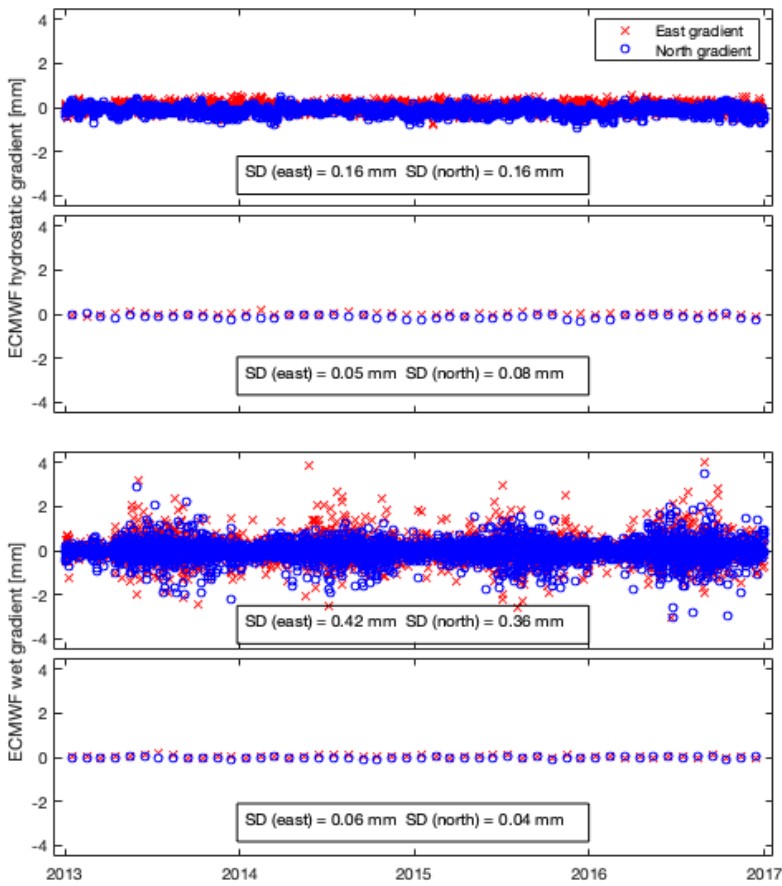

**Figure 8.** The ECMWF gradients for the Onsala (ONSA) site during the 4-year time period studied in Section 5. From the top: hydrostatic gradients every 6 h, their monthly averages, wet gradients every 6 h, and their monthly averages.

### 3.4 ECMWF

The Technical University of Vienna provides hydrostatic and wet gradients based on ECMWF data for many space geodetic sites globally. The product used here is usually referred to as LHG (linear horizontal gradients) and is described by Boehm and Schuh (2007). It is available during certain time periods from the mid of 2005 and is more continuous from 2006. It is computed
5   from profiles of hydrostatic and wet refractivity with a temporal resolution of 6 h, and a spatial resolution of 0.25° (∼30 km). The profile closest to the site is used together with one profile to the east and one profile to the north to calculate the refractivity gradient profiles. These are thereafter integrated to give the delay gradients. Because it was observed that on average the gradients computed in this way overestimate the more accurate gradients estimated from slant profiles, they are scaled by empirically derived factors, 0.53 for the hydrostatic gradients and 0.71 for the wet gradients (Boehm and Schuh, 2007). This computation
10   method and rescaling provide gradient estimates of limited accuracy but they still represent valuable and independent source of information which are used here for comparisons with estimated GPS gradients.

There are alternative methods to derive gradients from Numerical Weather Model data using ray tracing methods, see e.g. (Zus et al., 2015) and references therein. More recently the Technical University of Vienna also introduced a new gradient product based on a least-squares adjustment of the ERA-Interim analyses (Landskron and Böhm, 2018).

In this study we used the LHG data from 2006 to 2016, resulting in a time series of 11 years. As an introduction, examples of the ECMWF hydrostatic and wet gradients are illustrated in Figure 8. Worth noting is that the wet gradients dominate for the temporal resolution of 6 h and vary with the season, whereas the wet and the hydrostatic gradients show similar standard deviations (SD) for the monthly averages.

## 3.5 Summary of datasets

The results of comparisons between the gradients from these datasets are presented in the next two sections. The usage is defined in Table 3. In Section 4 GPS gradients estimated using the $10°$ elevation cutoff angle are compared to the ECMWF gradients. The temporal resolution is limited to 6 h in the ECMWF data. On the other hand the time series are 11 years long. The results in Section 5 focus on comparisons of the wet gradients at the Onsala site. These have a temporal resolution of 15 minutes when comparing to WVR data and the ECMWF data are only used to subtract the hydrostatic gradients from the total gradients estimated by the GPS and the VLBI techniques. In Table 4 we summarise the typical formal errors of the remote sensing techniques. Worth noting is the larger formal error for the north GPS gradient, compared to the east gradient, using the elevation cutoff angle of $20°$. The reason is that we lose many observations of satellites located in the north, see Figure 3. Other important comments are that WVR gradients are not estimated during rain events and are not based on observations below $20°$ elevation angles, but have a more homogeneous sky coverage compared to the GPS and the VLBI observations. Gradients from GPS and WVR have a superior temporal resolution, 5 and 15 min, respectively, compared to the 6 h of the VLBI and the ECMWF gradients.

**Table 3.** Summary of used datasets.

| Dataset | Resolution | Time period | ONS1 | ONSA | SPT0 | VIS0 | MAR6 | KIR0 |
|---------|-----------|-------------|------|------|------|------|------|------|
| GPS [a] | 5 min | 2006–2016 | – | √ | √ | √ | √ | √ |
| ECMWF[b] | 6 h | 2006–2016 | – | √ | √ | √ | √ | √ |
| GPS [c] | 5 min | 2013–2016 | √ | √ | – | – | – | – |
| WVR | 15 min | 2013–2016 | √ | √ | – | – | – | – |
| VLBI | 6 h | 6–20 May 2014 | √ | √ | – | – | – | – |

[a] The GPS data were processed with elevation cutoff angles equal to $10°$.

[b] (Boehm and Schuh, 2007)

[c] The GPS data were processed with elevation cutoff angles equal to $3°$, $10°$, and $20°$, different mapping functions, and elevation angle dependent weighting.

**Table 4.** Formal errors of the remote sensing techniques

| Data set | Elev. cutoff angle (°) | Formal error | | |
|---|---|---|---|---|
| | | Gradient | | ZWD |
| | | East (mm) | North (mm) | (mm) |
| GPS | 3 | 0.14 | 0.13 | 1.7 |
| GPS | 10 | 0.19 | 0.20 | 2.3 |
| GPS | 20 | 0.35 | 0.43 | 4.0 |
| WVR | 20 | 0.04 | 0.04 | 0.2 |
| VLBI | 5 | 0.14 | 0.13 | 1.7 |

# 4 Comparison of gradients from GPS and ECMWF data for the time period 2006–2016

## 4.1 Seasonal variations of horizontal gradients

We start by investigating the characteristics of the gradients over the year. In Figure 9 we present the monthly mean gradients for the time period 2006–2016 estimated from ECMWF data and GPS data from the Onsala (ONSA) station. In the top graphs, comparing ECMWF and GPS gradients, we note that the GPS gradients show a larger variability. There are also differences between the east and the north gradients both in the mean over the year and in the seasonal variations.

We can clearly see negative north gradients in the winter, with a mean value around $-0.2$ mm, both in the GPS and the ECMWF results. When the ECMWF gradients are separated into the hydrostatic and the wet components this variation appears in the hydrostatic component. We interpret this effect as the influence of the Icelandic low pressure system mentioned in Section 2. The winter feature is clearly seen in the analyses of the mean sea level pressure in the ERA-40 Atlas (https://software.ecmwf.int/static/ERA-40_Atlas/docs/section_B/parameter_mslp.html)

The results for the other four stations (KIR0, MAR6, SPT0, and VIS0) show similar systematic features. One exception is KIR0, which is at a higher latitude and has a less humid climate. At KIR0 the average monthly wet gradients are much smaller except during the summer months. Furthermore, the influence of the Icelandic low pressure in the winter is not as large as it is at the other four stations. Another exception is seen in the ECMWF wet gradients for ONSA in Figure 9. They are larger in the summer when the wet refractivity is higher. This is also seen at the other stations, but at ONSA there is a tendency of a positive east gradient in the summer. The ONSA GPS station is located a few hundred metres from the coastline, see Figure 1, suggesting that the air on the average is more humid over land compared to over the sea. One possible cause could be the sea breeze that occurs during the summer (Craig et al., 1945; Miller et al., 2003). The issue of wet gradients is studied further using a higher temporal resolution and comparisons with the WVR data in Section 5.

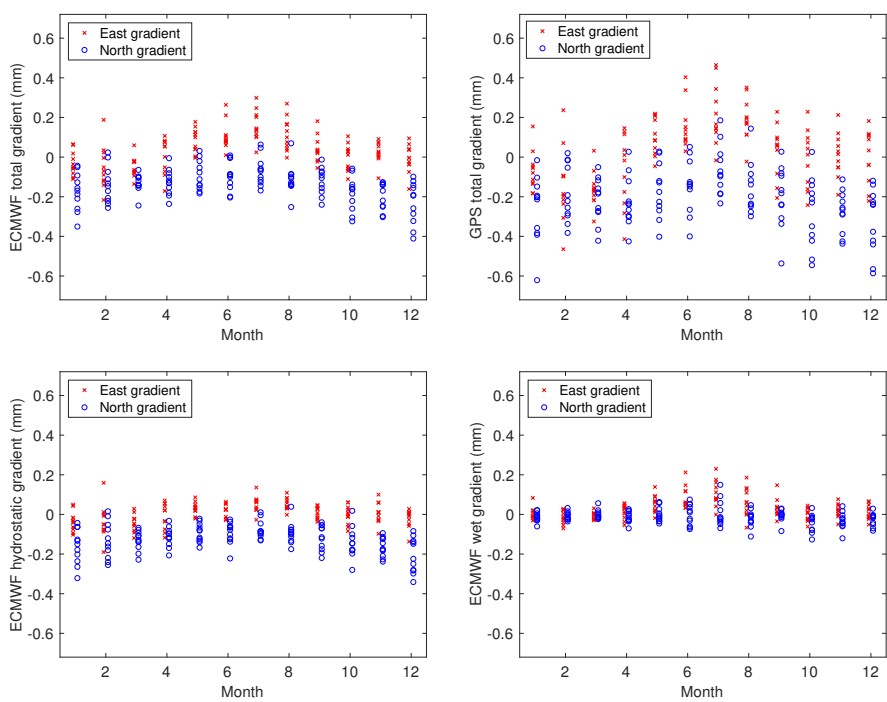

**Figure 9.** Monthly means of estimated gradients at the ONSA station for the period 2006–2016. The top graphs show the total gradients from ECMWF (left) and GPS (right). The graphs at the bottom show the ECMWF gradients when separated into the hydrostatic (left) and the wet gradient (right).

**Table 5.** Correlation coefficients for the total east and north gradients estimated from GPS data and compared to ECMWF data.

| Station | Six hourly | | Daily | | Monthly | |
|---|---|---|---|---|---|---|
| | East | North | East | North | East | North |
| Kiruna (KIR0) | 0.55 | 0.53 | 0.76 | 0.75 | 0.77 | 0.82 |
| Mårtsbo (MAR6) | 0.58 | 0.51 | 0.75 | 0.72 | 0.83 | 0.80 |
| Borås (SPT0) | 0.58 | 0.58 | 0.74 | 0.74 | 0.88 | 0.85 |
| Visby (VIS0) | 0.55 | 0.56 | 0.71 | 0.75 | 0.84 | 0.81 |
| Onsala (ONSA) | 0.60 | 0.60 | 0.75 | 0.78 | 0.90 | 0.90 |

## 4.2 Comparing GPS and ECMWF gradients over different time scales at the five stations

We study the agreement, in terms of correlation coefficients, between the total GPS and ECMWF gradients from 5 GPS stations using data from 2006 to 2016. These are shown in Table 5.

The correlations seen in all cases confirm that a consistent atmospheric signal in terms of gradients is detected by the
5 GPS observations and ECMWF analyses. We note that the correlation coefficients increase for longer averaging time periods. Our interpretation is that by long term averaging we compare a larger fraction of the gradient that is caused by large scale temperature and pressure gradients. Unfortunately, the temporal resolution of 6 h in the ECMWF data is not sufficient to resolve neither rapid changes in the pressure related to moving weather systems nor many of the short lived small-scale gradients associated with the variability in the water vapour.

Another result worth noting is that the two stations with the highest correlation coefficients, especially for the monthly averages, are ONSA and SPT0. The 95 % confidence interval is +0.03/−0.04 for the correlation coefficient of 0.90 obtained at station ONSA, based on 131 data points (12 months over 11 years). These two stations are the only ones that are equipped with microwave absorbing material below the antenna and above the metal plate used for the antenna mounting. This could reduce the impact from unwanted multipath effects. The phenomenon calls for further studies.

The mean values and the SD of the gradients, for the three different temporal resolutions, are presented in Tables 6 and 7 from GPS and ECMWF data, respectively. For the 6-hour temporal resolution the GPS gradients estimated at the same time epoch as the ECMWF gradients are included in the calculations. The daily and monthly values are averages using these 6-hour data. When comparing the two tables it is clear that there are differences in the mean values of up 0.2 mm. These differences are mainly in the east component whereas there are consistent negative values for the north component. The SD of the GPS
gradients are larger than the ECMWF gradients by a factor of 2. The differences may be explained by at least two reasons. First, the ECMWF gradient data used here have some intrinsic shortcomings (see Section 3.4). Second, not all variations in the water vapour content observed by the GPS receivers are actually represented in the ECMWF model due to its rather coarse spatial and temporal resolutions (Bock and Parracho, 2019).

**Table 6.** Mean values and standard deviations (SD) over the 11 years of estimated total gradients from GPS data for different temporal resolutions.

| Station | ZWD[a] | | Horizontal gradient | | | | | | | |
|---|---|---|---|---|---|---|---|---|---|---|
| | | | Mean[b] | | Six hourly SD | | Daily SD | | Monthly SD | |
| | Mean | SD | East | North | East | North | East | North | East | North |
| | (mm) | (mm) | (mm) | (mm) | (mm) | (mm) | (mm) | (mm) | (mm) | (mm) |
| Kiruna (KIR0) | 62 | 36 | −0.21 | −0.14 | 0.47 | 0.47 | 0.32 | 0.31 | 0.13 | 0.13 |
| Mårtsbo (MAR6) | 88 | 46 | −0.23 | −0.13 | 0.55 | 0.58 | 0.37 | 0.36 | 0.14 | 0.15 |
| Borås (SPT0) | 87 | 45 | −0.24 | −0.12 | 0.56 | 0.49 | 0.38 | 0.38 | 0.16 | 0.17 |
| Visby (VIS0) | 88 | 47 | −0.07 | −0.23 | 0.60 | 0.56 | 0.40 | 0.37 | 0.16 | 0.13 |
| Onsala (ONSA) | 92 | 47 | 0.01 | −0.20 | 0.59 | 0.55 | 0.41 | 0.38 | 0.18 | 0.15 |

[a] The Zenith Wet Delay (ZWD) is included to illustrate the amount of water vapour in the atmosphere above the station and its SD is based on the 6 h gradients.

[b] The mean gradient values are based on the 6 h gradients.

We note that the SD obtained for the KIR0 station for 6 h and one day are smaller. This is likely a consequence of the lower humidity at the station. For monthly averages, these differences are reduced and the SD for all stations are in the range 0.13–0.18 mm indicating that the hydrostatic gradients and other effects, e.g. signal multipath effects, become relatively more important. Variations in the electromagnetic environment that change the impact of the signal multipath at a station may be due to e.g. snow, rain, vegetation, and soil moisture. The relative importance of hydrostatic and wet gradients was illustrated in Figure 8 using four years of data from the ONSA station. Using all eleven years of ECMWF data, all stations have standard deviations of the hydrostatic east and north monthly gradients in the range from 0.05 mm to 0.07 mm, whereas the standard deviations for the monthly wet gradients show a dependence with latitude, from 0.03 mm at KIR0 in the north to 0.06 mm at ONSA in the south.

**Table 7.** Mean values and standard deviations (SD) over the 11 years of estimated total gradients from ECMWF data for different temporal resolutions.

| Station | Horizontal gradient | | | | | | | |
| --- | --- | --- | --- | --- | --- | --- | --- | --- |
| | Mean[a] | | Six hourly SD | | Daily SD | | Monthly SD | |
| | East | North | East | North | East | North | East | North |
| | (mm) | (mm) | (mm) | (mm) | (mm) | (mm) | (mm) | (mm) |
| Kiruna (KIR0) | 0.00 | −0.14 | 0.28 | 0.26 | 0.20 | 0.19 | 0.07 | 0.07 |
| Mårtsbo (MAR6) | −0.22 | −0.13 | 0.38 | 0.34 | 0.25 | 0.23 | 0.08 | 0.09 |
| Borås (SPT0) | −0.00 | −0.13 | 0.39 | 0.35 | 0.25 | 0.24 | 0.09 | 0.09 |
| Visby (VIS0) | −0.01 | −0.14 | 0.42 | 0.37 | 0.26 | 0.25 | 0.08 | 0.08 |
| Onsala (ONSA) | 0.03 | −0.14 | 0.43 | 0.37 | 0.27 | 0.25 | 0.10 | 0.09 |

[a] The mean gradient values are based on the 6 h gradients.

## 5   Wet gradients at the Onsala site

For the Onsala site we study total gradients from the two GPS stations and one VLBI station and wet gradients from the WVR for the time period 2013–2016. We use the hydrostatic gradients from ECMWF to calculate wet gradients from GPS and VLBI total gradients. The designs of the two GPS stations are different, see Figure 2, which motivates to include both of them in the comparisons. Three different studies are made using these data: (1) assessment of the impact of using different processing of the GPS data, primarily varying the elevation cutoff angle, by comparison to the WVR gradients; (2) using the GPS gradients from the processing variant showing the best agreement with the WVR gradients, the seasonal variations in the wet gradient are characterised; and (3) a 15-day long period with VLBI data is used as a case study for comparisons with GPS and WVR wet gradients and the ZWD.

### 5.1   Test of GPS processing variants relative to WVR data

Gradients in the east and the north directions are estimated from the GPS data for five different solutions. We use three different elevation cutoff angles for the VMF1 zenith delay mapping functions. One additional solution is carried out with elevation dependent weighting $(\sin(\varepsilon))$ and in the fifth solution the VMF1 mapping functions are replaced by the NMF. As stated earlier the gradient mapping function presented by Bar-Sever et al. (1998) is used in all cases.

The GPS wet gradients for ONSA and ONS1 are computed by subtracting the hydrostatic gradients from ECMWF (see Figure 8), linearly interpolated to match the time epochs of the GPS gradients, from the total GPS gradients. Thereafter, we form 15 min averages for the east and the north wet gradients from GPS and compare to the corresponding WVR results.

The results for the different GPS solutions are summarised in Tables 8 and 9. Because of the different gradient amplitudes from the WVR and GPS, we present mean values and SD of the differences as well as correlations coefficients. Table 8 shows the results when the total gradients from the stations ONSA and ONS1 are compared to each other. Table 9 shows the results when the wet gradients from ONSA and ONS1 are compared to the WVR gradients. We note that in both tables the best agreement between the gradients estimated is obtained for an elevation cutoff angle equal to $3°$. The 95 % confidence interval for correlation coefficients around 0.65 and approximately 80,000 data pairs is $\pm$ 0.004. This result was not expected by us, given that the WVR has an elevation cutoff angle of $20°$ (in order to avoid ground-noise pickup) the GPS solution using the same cutoff angle would show a better agreement. Our interpretation is that for the temporal resolutions of 5–15 min the low elevation observations are important in order to distinguish the gradient parameters relative to other estimated parameters in the GPS analysis. A higher elevation cutoff angle will remove many observations towards the north, and especially for a cutoff angle of $20°$, see Figure 3 and Table 4 with the formal errors.

The solution giving the best agreement, when comparing gradients from ONSA and ONS1 data with each other, is the one with elevation dependent weighting, whereas the comparisons with the WVR, for both ONSA and ONS1, give the best agreement without weighting. The choice of elevation cutoff angle is a compromise between having a good geometry and avoiding effects of signal multipath. Our interpretation is that the gradients from ONSA and ONS1 are estimated based on very similar observational directions and have common error sources, such as orbit errors, resulting in correlations around 0.9.

**Table 8.** Assessment of the different GPS solutions comparing total gradients from the two GPS stations ONSA and ONS1.

| GPS Solution | Mean Difference[a] | | Standard Deviation | | Correlation Coefficient | |
|---|---|---|---|---|---|---|
| | East | North | East | North | East | North |
| | (mm) | (mm) | (mm) | (mm) | (mm) | (mm) |
| VMF 3° | −0.01 | 0.03 | 0.22 | 0.25 | 0.91 | 0.87 |
| VMF 3°[b] | 0.03 | 0.02 | 0.15 | 0.16 | 0.95 | 0.92 |
| NMF 3° | −0.01 | 0.05 | 0.23 | 0.26 | 0.91 | 0.86 |
| VMF 10° | 0.02 | 0.04 | 0.25 | 0.27 | 0.91 | 0.88 |
| VMF 20° | 0.33 | 0.36 | 0.39 | 0.47 | 0.82 | 0.70 |

[a] The mean difference is ONS1−ONSA.

[b] Elevation dependent weighting, $\sin(\varepsilon)$

In order to increase an already high correlation the observations at the lowest elevation angles are not that important, since multipath effects will be more and more different the closer to the horizon the observations are made. When ONSA and ONS1 gradients are compared to those from the WVR the situation is different, because these gradients are independent and the geometry of the GPS observations becomes more important in order to estimate a more accurate gradient. Although we note that the correlation coefficients are here reduced, to 0.68 and 0.64 for the east and the north component, respectively. Since the WVR provides independent gradients, we will in the following focus on the VMF 3° solution without elevation dependent weighting.

**Table 9.** Assessment of the different GPS solutions for the wet gradients from the two GPS stations ONSA and ONS1 relative to the WVR data.

| GPS Solution | Mean Difference[a] | | Standard Deviation | | Correlation Coefficient | |
|---|---|---|---|---|---|---|
| | East (mm) | North (mm) | East (mm) | North (mm) | East (mm) | North (mm) |
| **ONSA** | | | | | | |
| VMF 3° | 0.23 | −0.07 | 0.64 | 0.57 | 0.68 | 0.64 |
| VMF 3°[b] | 0.21 | −0.06 | 0.71 | 0.62 | 0.58 | 0.55 |
| NMF 3° | 0.22 | −0.07 | 0.64 | 0.57 | 0.68 | 0.64 |
| VMF 10° | 0.20 | −0.10 | 0.65 | 0.59 | 0.66 | 0.62 |
| VMF 20° | −0.02 | −0.28 | 0.75 | 0.73 | 0.54 | 0.42 |
| **ONS1** | | | | | | |
| VMF 3° | 0.22 | −0.04 | 0.64 | 0.58 | 0.68 | 0.64 |
| VMF 3°[b] | 0.24 | −0.02 | 0.71 | 0.63 | 0.58 | 0.55 |
| NMF 3° | 0.21 | −0.02 | 0.64 | 0.58 | 0.68 | 0.63 |
| VMF 10° | 0.22 | −0.04 | 0.66 | 0.59 | 0.66 | 0.62 |
| VMF 20° | 0.36 | 0.15 | 0.79 | 0.73 | 0.49 | 0.42 |

[a] The mean difference is the offset referenced to the corresponding WVR time series.

[b] Elevation dependent weighting, $\sin(\varepsilon)$

**Table 10.** The impact of the elevation cutoff angle on the estimated 15-minute GPS gradient amplitude[a]

| Elev. cutoff angle | Mean gradient amplitude | | SD of gradient amplitude | |
| | ONSA (mm) | ONS1 (mm) | ONSA (mm) | ONS1 (mm) |
| --- | --- | --- | --- | --- |
| 3° | 0.51 | 0.50 | 0.41 | 0.41 |
| 10° | 0.58 | 0.59 | 0.45 | 0.45 |
| 20° | 0.75 | 0.70 | 0.45 | 0.46 |

[a] The corresponding WVR values are only available for the 20° elevation cutoff angle: the mean is 0.87 mm and SD is 0.78 mm.

## 5.2 Wet gradients from GPS and WVR

An overview of the data in terms of monthly means of the wet gradient amplitude and the ZWD is presented in Figure 10. The GPS solution with a 3° elevation cutoff angle, no weighting, and the VMF1 mapping functions is used. When forming monthly means the correlations are obvious, both between GPS and WVR estimates, and between the variability, in terms of the SD, and the gradient amplitudes and the ZWD. Here we also note that the WVR gives much larger gradients. Factors that can cause a difference in gradient amplitude are:

(1) The WVR is sensitive to liquid water in the atmosphere. This is a cause for positive systematic errors in the ZWD as well as occasional overestimates of gradient amplitudes. We investigated this possibility by deleting all WVR observations implying a liquid water content larger than 0.3 mm. The impact was however insignificant. The average gradient amplitude decreased by 0.01 mm. The reason being that large liquid contents are rather infrequent, given that already data acquired during rain (which was assumed to occur when liquid water content was larger than 0.7 mm) have been removed.

(2) The WVR gradients for one 15-minute period do not depend on earlier or later estimates whereas the GPS gradients are estimated using constraints on the variability. A constraint has a similar impact as a low-pass filter (peaks with a short duration will be reduced).

(3) The fact that the WVR and the GPS gradients are computed for different elevation cutoff angles has two possible impacts: (i) the larger volume sensed by GPS (with a 3° cutoff angle) includes different air masses and introduces an averaging effect that reduces the mean amplitude and the variability of the gradients, similar to averaging over longer time periods as shown in Table 6; (ii) the higher cutoff angle (20°) results in larger formal errors, and thus larger variability and larger gradient amplitudes. Table 10 shows the impact of changing the elevation cutoff angle for the GPS observations for ONSA and ONS1 over the 4-year period 2013–2016. We also note from Table 4 that the formal errors of the WVR gradients are significantly smaller than those for the GPS gradients.

We conclude that the constraints and the sampling of different air masses are the likely explanations for the differences in gradient amplitudes estimated from GPS and WVR data but cannot based on these results determine their relative importance.

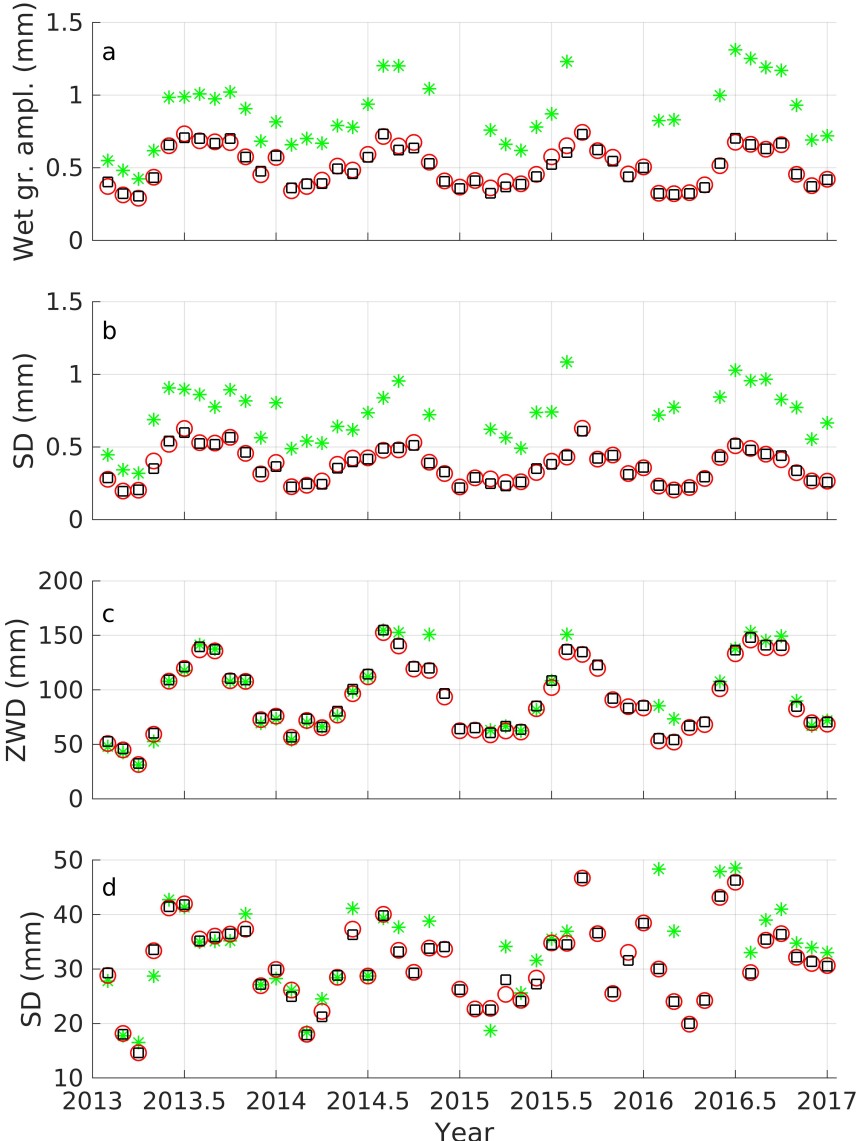

**Figure 10.** Time series of (a) monthly means of wet gradient amplitudes, $\sqrt{\Xi_{e,wet}^2 + \Xi_{n,wet}^2}$, (b) their SD, (c) monthly means of the ZWD, and (d) the ZWD SD from GPS and WVR. The GPS results are from the $3°$ solution without weighting and the temporal resolution in the time series used to calculate the monthly mean and the SD is 15 min. The green stars denote WVR data. The ONSA and ONS1 data are denoted by red circles and black squares, respectively.

A correlation plot for the total gradients from ONSA and ONS1 for the VMF1 solution with a 3° elevation cutoff angle is shown in Figure 11. As in the previous section we see a slightly higher correlation for the east gradients, possibly because of the poorer sampling on the sky north of the zenith direction due to the geometry of the GPS satellite constellation at this latitude (see Figure 3). The two GPS stations share several error sources, such as clock and orbit errors of the observed satellites, and the use of the same mapping functions, meaning that the rather high correlation is overoptimistic due to a common mode suppression of errors.

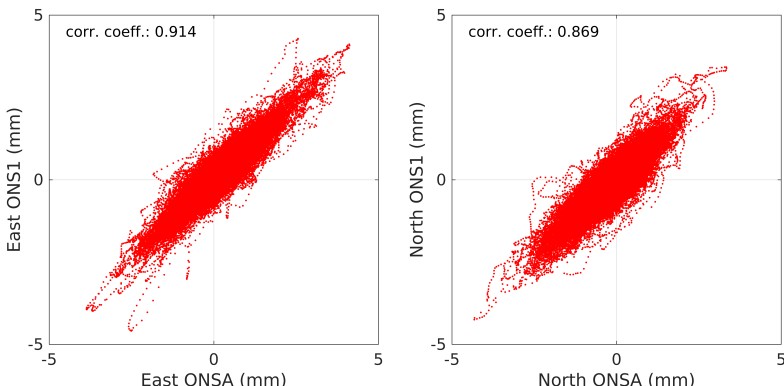

**Figure 11.** Correlations between estimated total gradients from the GPS stations ONSA and ONS1 using all data with a 5 min resolution from the period 2013–2016.

Correlation plots for the wet gradients from ONSA, ONS1, and the WVR are presented in Figure 12. As seen previously from total gradients the correlations between the estimated gradients from the two GPS stations are significantly higher compared to when the GPS gradients are correlated with the gradients from the WVR. It is also not surprising that the correlation between the wet gradients from ONSA and ONS1 are slightly lower compared to the correlation between the total gradients (Figure 11). When subtracting the hydrostatic gradients, a common signal is removed and the dynamic range is reduced, which affects the correlation coefficients.

The reasons for the lower correlation coefficients between the WVR and the GPS gradients are almost identical with the reasons above why the WVR gradient amplitudes are higher: (1) they do not have common sources of errors; (2) the WVR data suffer both from white noise and algorithm errors, especially when liquid water is present; (3) the WVR data for each 15-minute period are independent of the successive periods, whereas there are temporal constraints on the gradients estimated from the GPS data; (4) the sampling on the sky agrees also much better between the two GPS stations, assuming that in general the directions of the observations are towards the same satellites, whereas the WVR observations are evenly spread over the sky and above an elevation angle of 20°.

Concerning the sampling of the atmosphere, the use of a multi-GNSS constellation has been shown to improve the agreement between GNSS gradients with those estimated from a WVR (Li et al., 2015). In this context it should be noted that with many

more GNSS observations the optimum elevation cutoff angle may not be as low as 3° because of an improved sampling of the atmosphere.

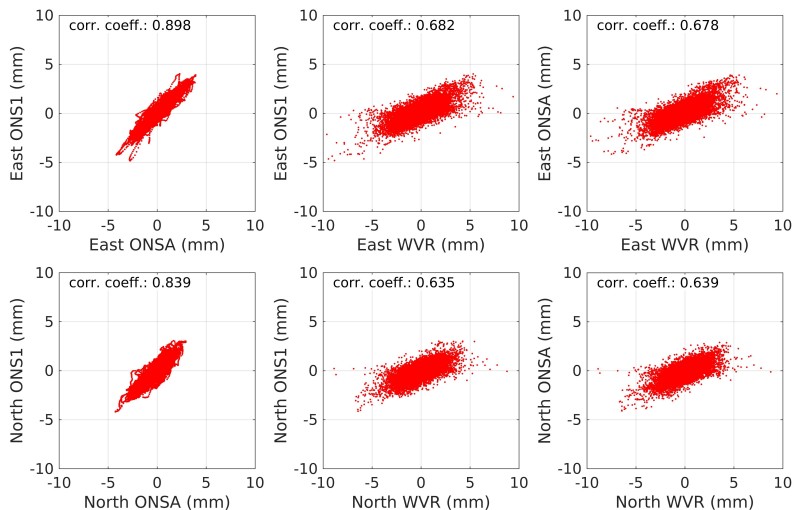

**Figure 12.** Correlations between estimated wet gradients from the WVR, ONSA and ONS1 using all data from the period 2013–2016. The data in the graphs with ONSA and ONS1 (left) have the original 5 min resolution, whereas the GPS data are averaged over 15 min when compared to the WVR data (middle and right). The correlation coefficients obtained when the east gradients from the WVR were correlated with the original total gradients from GPS were 0.633 for ONSA and 0.637 for ONS1. The corresponding values for the north gradients were 0.575 for WVR-ONSA and 0.571 for WVR-ONS1. This supports our assumption that the ECMWF hydrostatic gradients are reasonably accurate when carrying out a linear interpolation between the 6-hour samples.

We investigated if an average of the wet gradients from both GPS stations, ONSA and ONS1, estimated at the same time epoch, will improve the agreement with the WVR. We see an overall small improvement. For the east gradient the individual correlation coefficients were improved from 0.678 (ONSA) and 0.682 (ONS1) to 0.698. The corresponding values for the north gradient were increased from 0.639 (ONSA) and 0.635 (ONS1) to 0.666. Our interpretation is that by averaging the GPS gradients from ONSA and ONS1 the stochastic noise is reduced.

Correlation plots are shown in Figure 13 for each month of the four years. A clear seasonal dependence is seen, because the variability in the wet refractivity is larger during the warmer time periods, resulting in larger gradients and a larger dynamic range. We note that during October 2014 there were problems with the WVR (see Figure 6). During most of the days there is a significant data loss, likely due to rain, which could be the reason for the low correlation during this month. The other months with low correlations are March 2015 for both the east and the north component, and January and February 2016 for the north component. In all these cases there were no large gradients detected and this has an impact on the correlations. In Figure 8 of Lu et al. (2016) a correlation coefficient of 0.52 was reported for the months March–May, 2014, between GPS and WVR

gradients. Here we show that the variability from month to month is large and therefore the choice of the time period for gradient comparison studies is a critical issue.

Comparing the results obtained for ONSA with those from ONS1 they are almost identical (in both Figures 12 and 13) meaning that in this case there is no obvious improvement from the absorbing material below the antenna on ONSA. This is different to the previous finding where ONSA and SPT0, with microwave absorbing material, showed a better agreement with ECMWF gradients compared to the KIR0, MAR6, and VIS0 stations. Our assumption is that the lack of a concrete pillar with a metal mounting plate just below the antenna on ONS1, or any other objects affecting the electromagnetic environment at the antenna, eliminates the need for an absorber (see Figure 2).

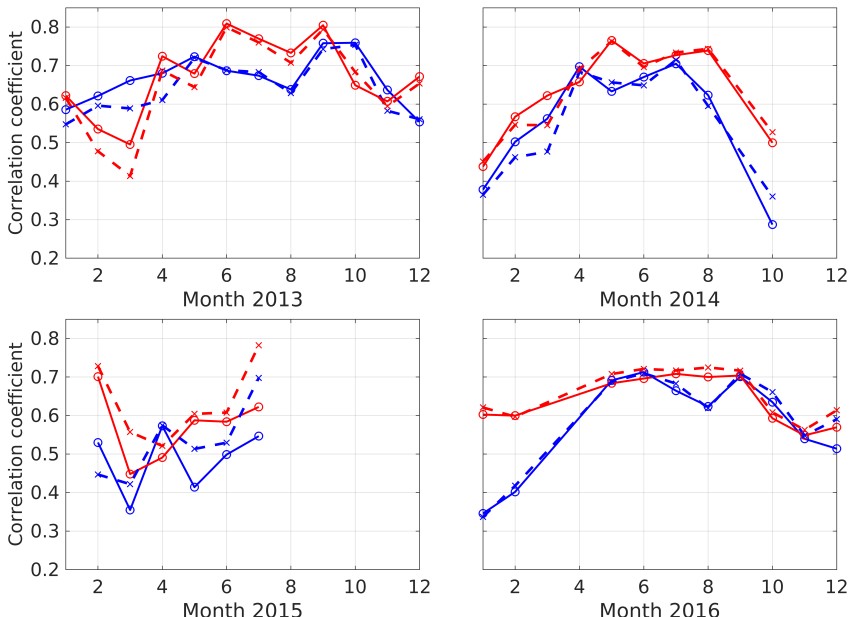

**Figure 13.** Correlations between estimated wet gradients from the WVR data and the GPS data from ONSA (solid lines) and ONS1 (dotted lines) averaged over 15 min when the hydrostatic gradients are removed from the total GPS gradients for each month of the four years. The east gradients are presented with red lines and the north gradients with blue lines.

## 5.3 GPS, VLBI, and WVR wet gradients during CONT14

The wet gradients from the two space geodetic techniques GPS and VLBI are compared to each other and to the WVR during the CONT14 campaign. Observations from several earlier CONT campaigns have been analysed in terms of gradients with different results depending on the station and the time of the campaign (Teke et al., 2013). We use this campaign as an example study of the short term variability of the wet gradients. The GPS gradients are those obtained from the VMF1 solution, unweighted, with a 3° elevation cutoff angle. The ECMWF data, see Figure 14, is only used to subtract the hydrostatic gradients from the total gradients estimated by VLBI and GPS. The time series of estimated gradients and ZWD are shown in Figure 15.

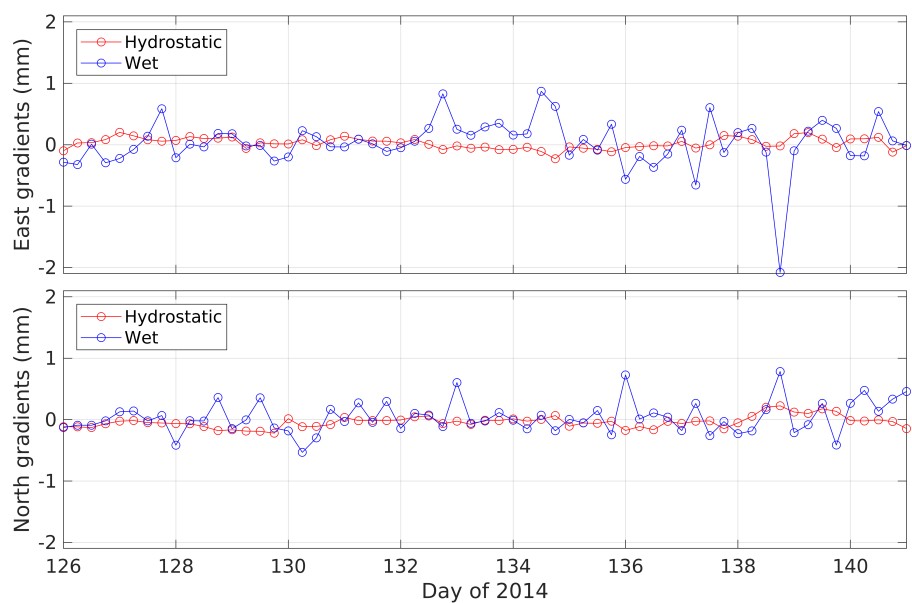

**Figure 14.** Time series of ECMWF hydrostatic and wet gradients during the CONT14 campaign.

Again we note that the size of the WVR gradients is larger compared to all other instruments. The VLBI gradients correlate with the gradients from the other instruments but their amplitudes are smaller. Given that the sampling of the atmosphere is much more sparse with the VLBI telescope, a short lived gradient in combination with the assumption of linear functions in 6-hour segments, will probably reduce the variability in the estimated amplitude.

Table 11 summarises the correlation coefficients for the east and the north VLBI wet gradients compared to those from the two GPS stations, ONSA and ONS1, and the WVR. Here we have correlated averages using data ±3 h around the VLBI gradient value every 6 h. In order to be consistent also the interpolated data from continuous VLBI segments are averaged in this way.

We note that the correlation coefficients are lower for the north component for all three comparisons, whereas the SDs are similar. The reason is that the size of the east gradients are larger compared to the north gradients during this 15-day period. Scatter plots (not shown) confirm what is indicated by the SDs, that the quality of the east and north components is similar.

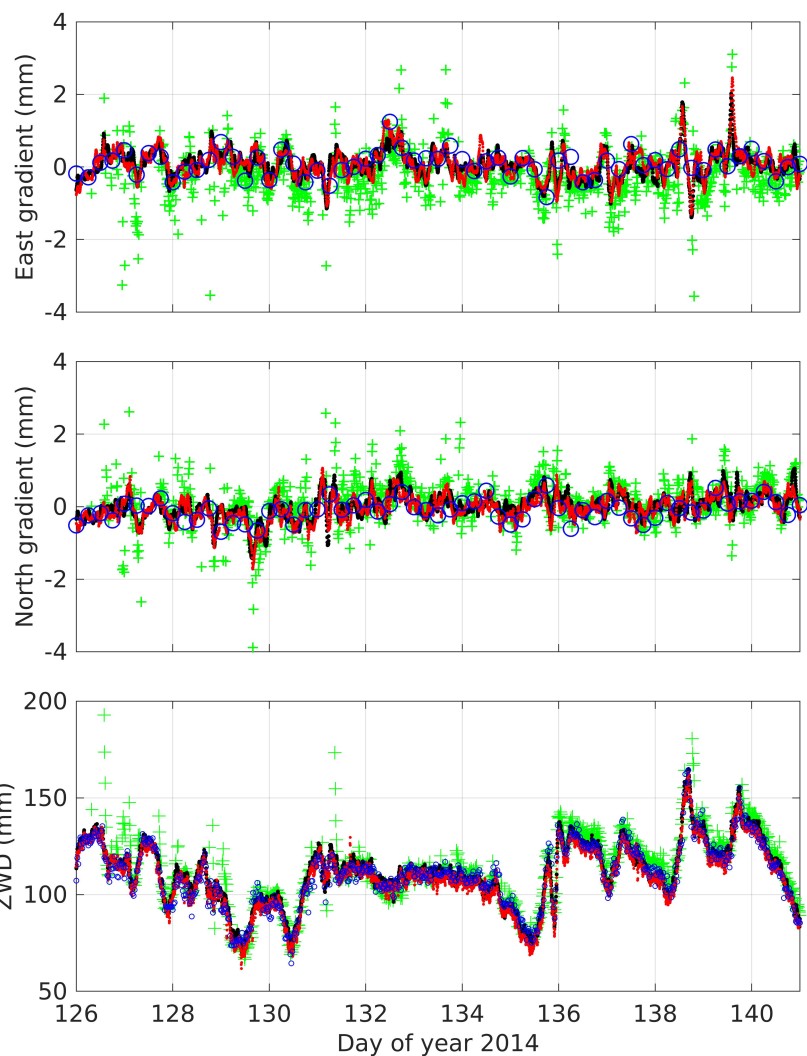

**Figure 15.** The wet gradients and the ZWD during the VLBI CONT14 campaign 6–20 May (days 126–140). The temporal resolution for the VLBI (blue circles) gradients is 6 h and the ZWD 30 min, 5 min for the GPS gradients for ONSA (red dots) and ONS1 (black dots), and 15 min for the WVR (green plus).

We attribute the lower correlation coefficients obtained between VLBI-GPS and VLBI-WVR using 6 h averages during the CONT14 campaign compared to GPS-WVR 15 min averages for the month of May 2014 in Figure 13 to the sparse sequential

**Table 11.** Comparison of estimated wet gradients from VLBI relative to GPS and WVR data.

| Reference instrument | Mean difference[a] | | Standard deviation | | Correlation coefficient | |
|---|---|---|---|---|---|---|
| | East (mm) | North (mm) | East (mm) | North (mm) | East (mm) | North (mm) |
| ONSA | 0.01 | −0.03 | 0.22 | 0.20 | 0.71 | 0.57 |
| ONS1 | 0.03 | −0.08 | 0.22 | 0.20 | 0.71 | 0.56 |
| WVR | 0.30 | −0.17 | 0.27 | 0.27 | 0.65 | 0.58 |

[a] The mean difference is VLBI−reference instrument.

sampling of the sky by the VLBI observations. On the other hand, averaging the WVR gradients over ±3 h reduces some of the noise seen in the 15 min values. The future use of the twin telescopes with faster slewing speeds at the site is likely to improve this situation. During CONT14 there were approximately 360 useful observations at Onsala per day. We expect this to increase by a factor of 6–7 when using the new VLBI Geodetic Observing System (VGOS) (Niell et al., 2018), which means

that the use of twin telescopes could result in 200 observations per hour. This in turn makes it possible to improve the temporal resolution of the estimated atmospheric gradients.

    Finally, we like to use this 15-day long time series for a discussion on gradient variability. At the end of day 135, see the ZWD plot in Figure 15, more humid air is starting to enter over the site. We note a sudden decrease, followed by a rapid increase. In Figure 16 we zoom in on the gradients and the ZWD during this period. Here we have an example with wet

gradients from GPS and WVR gradients when a warm front passage occurs in the evening of day 135. During this passage there is also a smaller drier air mass present causing a decrease followed by an increase in the ZWD. During this dip in ZWD the wind at the ground was from the west increasing from 7 m/s at 18 UT to 11 m/s at 24 UT. During the decrease in ZWD we see a clear positive east gradient and during the following increase in ZWD the east gradient has a negative peak. Also during the first few hours of day 136 a decrease in the ZWD corresponds to positive values for the east gradient, and the wind

continued to come from the west. This is as expected, but there are also variations in the north gradient during this period, consistently detected by the WVR and the GPS data, showing that the wind at the ground was not fully representative for all altitudes.

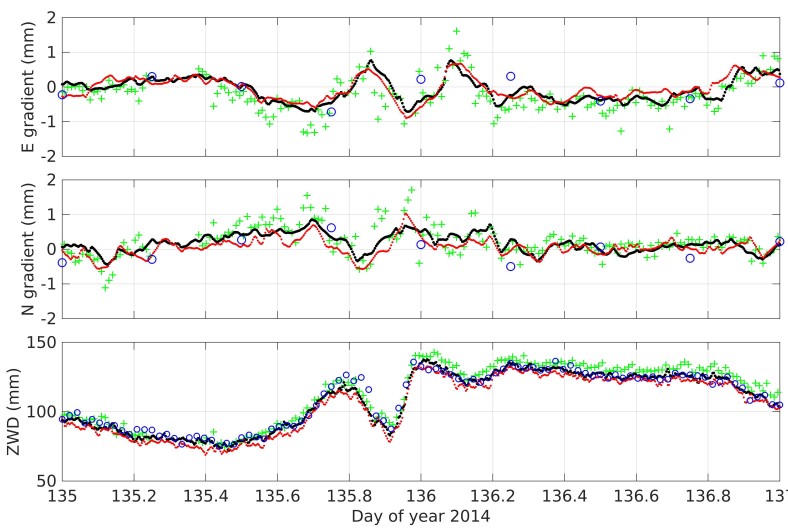

**Figure 16.** Zoom in on the time series in Figure 15. The symbols are as before: VLBI gradients (blue circles), GPS gradients for ONSA (red dots) and ONS1 (black dots), and WVR (green plus).

# 6 Conclusions and suggestions for future work

We have estimated linear horizontal gradients from GPS data acquired at five sites in Sweden. Averaging gradients in the east and the north direction over one month gives correlation coefficients of up to 0.9 when compared to gradients calculated from meteorological analyses of the ECMWF (Boehm and Schuh, 2007). The hydrostatic component gives the largest contribution to the monthly averages of the gradients.

We studied wet gradients estimated with a temporal resolution of 15 min from GPS and WVR data. We found that an elevation cutoff angle of 3° implies a better agreement when comparing GPS gradients with those from the WVR, in spite of the fact that the WVR does not observe the atmosphere below elevation angles of 20°. This confirms the result from Kačmařík et al. (2018) showing that low elevation observations are important for the accuracy of the estimated GPS gradients, although none of us has shown that 3° is an optimum cutoff angle.

We also note that using a 3° elevation cutoff angle in the GPS processing the amplitude of the GPS gradients decrease by approximately 20 % compared to using a 20° cutoff angle. We interpret this decrease as the result of two combined effects: (1) the decrease of mean amplitude and variability at the lower cutoff angle results from the averaging of a larger air mass (similar to averaging over longer periods); (2) the increase of mean amplitude and variability at the higher cutoff angle results from the increase of uncertainty and thus larger scatter in the estimates. The relative importance of these two effects are recommended to be studied further, e.g. by using simulations based on high resolution numerical weather models.

Correlation coefficients between wet gradients estimated from GPS and the WVR data can for specific months reach up to 0.8. We note a strong seasonal dependence, from 0.3 during months with smaller gradients to 0.8 during months with larger gradients, typically during the warmer, and more humid, part of the year. Related to this we suggest further studies of large wet gradients estimated from GPS, again in combination with meteorological high-resolution models for verification of the quality of the gradients.

In general we also note slightly higher correlation coefficients for the GPS derived gradients in the east compared to the north direction. We interpret this difference to be caused by an inhomogeneous spatial sampling on the sky, which is important when we assume that the linear model describing horizontal gradients has deficiencies. The different sampling on the sky is an important issue for any comparison between different techniques. This question remains unresolved and is also recommended for further studies.

Additional issues that deserve attention in future studies, in addition to similar studies in different climates, e.g. the tropics, can include multi-GNSS observations. At latitudes similar to those in this study, the use of GNSS satellites with a higher orbit inclination will reduce the part of the sky not sampled by GPS.

For VLBI the use of VGOS (twin) telescopes will also dramatically improve the sampling of the atmosphere. When WVR data are used to evaluate gradients from the space geodetic techniques one may consider to also apply different constraints for the temporal variability of these estimates. We suggest that future work on gradients should focus on the interplay between the elevation cutoff angle, the temporal resolution, and the constraint value using both single and multi-GNSS. Furthermore, we

believe that the outcome of such studies will be weather and site dependent which will make it difficult to arrive at just one optimal recommendation valid globally.

*Data availability.* The input GNSS data, in RINEX format, are available from EUREF, https://igs.bkg.bund.de/dataandproducts/browse. The input VLBI data are available from the IVS, ftp://ivs.bkg.bund.de/pub/vlbi/ivsdata/db/2014/. The ECMWF gradients are accessible from
5 the Technical University of Vienna, http://vmf.geo.tuwien.ac.at/trop_products/GNSS/LHG/. The estimated gradients from GPS, VLBI, and WVR data have been registered and archived by the Swedish National Data Service (SND): https://doi.org/10.5878/nswt-yr39.

*Author contributions.* Gunnar Elgered coordinated and wrote the major part of the manuscript and together with Tong Ning planned the different GNSS data analyses during the COST Action ES1206. Tong Ning performed the GNSS data analyses, resulting in the estimated gradients. Peter Forkman and Rüdiger Haas carried out the same task for WVR and VLBI data, respectively. All authors contributed in the
10 writing process, in particular to the sections presenting the results produced by each author and approved the entire manuscript before the submission.

*Competing interests.* The authors declare that they have no conflict of interest.

*Acknowledgements.* We appreciate the constructive comments and suggestions from the editor and the three referees. They resulted in a significantly improved paper, offering additional possible explanations to the obtained results as well as additional studies. For example, the
15 use of a 3° elevation cutoff angle in the GPS data processing was not included in our original manuscript. The map in Figure 1 was produced using the Generic Mapping Tools (Wessel and Smith, 1998).

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
