# Peer review of "On the information content in linear horizontal delay gradients estimated from space geodesy observations"

_Atmospheric Measurement Techniques, 2018_

## Referee Comment (RC1) · Anonymous Referee #1 · 7 Nov 2018

The study of the horizontal variability of the atmosphere is currently of great interest. Linear horizontal delay gradients are considered advanced GNSS meteorology products and it has been proved that they are a powerful tool to identify problems with GNSS data tracking. Although not yet assimilated into NWP models, they are fundamental for the reconstruction of the slant delay and, in turn, in the 3D water vapour fields derived by tomographic inversion of GNSS based slants. The analysis of the causes of the time variability on different time scales, from months to minutes, reported in the manuscript adds new insights in the research area of these advanced GNSS meteorology products. In the manuscript, tropospheric gradients estimated from GPS observations are evaluated with respect to independent techniques as WVR and VLBI and independent

data as ECMWF in order to assess their quality. I think it would be interesting in future to repeat the same kind of analysis in other regions. However, I would raise the following issues, which have to be clarified prior to the publication.

1. Mapping functions: GPS and VLBI data are analysed using different mapping functions: VMF1 for GPS and Niell for VLBI. No information about the gradient mapping function is given. I guess that in GPS data processing Bar-Sever et al. (1998) gradient mapping function is used, while in VLBI Chen and Herring (1997) gradient mapping function is applied. Kacmarik et al. (2018) recommend to agree on the gradient mapping function when tropospheric gradients derived from various sources are to be compared, since a systematic effect up to 0.3 mm is observed between Bar-Sever et al. (1998) and Chen and Herring (1997) gradient mapping functions. Having this in mind, I think that information on the gradients mapping function has to be provided in the manuscript and properly discussed.

2. Elevation cut-off: GPS data are processed at 100 and 200 elevation cut-off angle, while no info is provided for VLBI. Kacmarik et al. (2018) obtained better results using 30 elevation cut-off angle and GPS+GLONASS data. I recommend to process at least 1 GPS station with 30 elevation cut-off angle and evaluate the results. Should geodetic data processed with different cut-off angles depending on the application and on the tropospheric parameter of interest (ZTD or gradient)? A comment on this is really appreciated. The manuscript is within the scope of this special issue.

Kačmařík et al (2018), Sensitivity of GNSS tropospheric gradients to processing options, Ann. Geophys. Discuss., https://doi.org/10.5194/angeo-2018-93

Below specific comments.

Introduction Page 2 Line 6. I suggest adding the amount of improvement of multi-GNSS gradients compared to GPS-only gradients.

Cause of horizontal gradient In the manuscript, the mathematical model used to describe the tropospheric path delay is not reported. This can be added in this section and the title should be changed in 'model and cause of horizontal gradient'.

Instrumentation and data Page 3 Line 11-12. Complete the sentence 'We compare. . ...' adding at the end '. . .with respect to VLBI estimates, WVR and ECMWF data' I suggest adding in the section a table summarizing the characteristics of the instruments used for the evaluation, referring to the specific sub-section for further details.

GPS In this section, and also somewhere else in the manuscript, the term 'site' is used both referring to a local geographical area or referring to a unique geodetic marker. Please review it and use site for a local geographical area, where one or more geodetic markers are available, and station to indicate a unique geodetic marker at a site. Figure 2. You present the sky plot of GPS observation for May 12, 2014. Why did you select this specific day?

Gradients during the CONT14 VLBI campaign Figure 14. Add mean and std of the differences (GPS-VLBI), are these values affected by the different gradient mapping function?

Typos

Pag.2 Line 4: delete ')' after VLBI

Table 2 replace 'igs_1740.atx' with 'igs08_1740.atx', correct?

Pag.22 Line 20: horisontal -> horizontal?

––––––––––––––––––––––––––––––

---

## Referee Comment (RC2) · Anonymous Referee #2 · 10 Nov 2018

General comments

As expressed in the title, the manuscript deals with the information content in linear horizontal tropospheric delay gradients estimated from space geodesy observations, namely GPS and VLBI. The topic of the manuscript is highly actual. In past, the tropospheric delay gradients were estimated mainly for improving horizontal positioning (coordinate repeatability), although it was not always clear that it improved the troposphere modelling as the gradient parameters are highly sensitive to other error sources affecting the data analyses. So far, the gradients were also rarely estimated within an operational troposphere monitoring because the information content was often either

too noisy or too much smoothed by low temporal resolution or constraints. The situation is going to change in future when providing advanced tropospheric products on troposphere asymmetry monitoring, in particular with upcoming availability of more satellites from multi-GNSS constellations. Tropospheric delay gradients are also pre-requisite for delivering other products from GNSS such as retrieving slant delays, the reconstruction of three-dimensional refractivity field or generating severe weather event indicators. Attempts for developing assimilation techniques for GNSS-based tropospheric gradients emerged recently when it seems preferred way, compared the utilization of slant delays, due to the better production quality in (near) real-time. In this context, the manuscript contributes to a better understanding of how linear tropospheric delay gradients are able to characterize wet and hydrostatic effects of the neutral atmosphere on space geodetic observation analyses; both in actual situation or in a long-term trends and useable in geodetic, meteorological or climatology applications.

Generally, the manuscript is written carefully and results discussed in details. However, several suggestions are given below for completing the manuscript before its publishing.

1. Adding brief introduction of the model of calculating gradients from the NWM would be helpful for discussion of results, e.g. gradient mapping function, distribution of ray-tracing points, elevation angle cut-off.

2. Although the manuscript study a comparison of gradients, there are no information about gradient mapping functions used in estimating procedures of different techniques. Kačmařík et al. 2018 (submitted for discussions in ANGEO) showed that gradient mapping function could introduce systematic effects into estimated gradients. Similarly, no information about elevation-dependent weighting (if applied) was given neither for GPS nor VLBI. Please, add these information in corresponding tables or text paragraphs, and whenever useful, consider their impact in discussions as these might be more critical than the other processing settings.

3. The differences in size of estimated gradients estimated from different techniques are not discussed. These are visible in Figure 9 between GPS and Numerical Weather Model (NWM) and in Figures 11 and 12 for GPS compared to WVR. These could be attributed to various aspects such as used gradient mapping functions, limited resolution of numerical weather model data (ECMWF), observation sampling over the sky or others. Were similar characteristics common to the other stations?

For easier reading, I would also suggest to consider splitting sections 4.2 and, optionally 5.1, into two parts with more specific subtitles for more better clarity of different comparisons, see bellow in specific comments.

Specific comments

P1L11: 15-day long continuous

P2L4: VLBI, GPS and WVR (remove closing bracket).

P2L12: a 4-year period

P2L13: a 15-day long VLBI campaign

P2L20: over long-time scales

P6Tab2: add gradient mapping function and reference frame and PCV values applied (IGS08 or IGS14?)

P6L6: (consider modified wording) . . . estimated gradients are independent in adjacent epochs. . ..

P9L15: . . . as piece-wise linear offsets . . . Do you mean representation with a piece-wise linear function when represented with the interval end-point offsets? Or do you mean just constant offsets for individual intervals? Please, reword or clarify.

P11L5: the overall mean negative north gradients is also partly attributed to the flattening of the earth atmosphere, see Meindl et al. 2004, I suggest to add in the discussion

in this paragraph.

P12 Figure 9: discuss the different ranges of gradient sizes,

P13L6: by long-term averaging . . .

P13L18: . . . at this low humidity level . . .

P13L19: I suggest to add here new sub-sections for discussing long-term trends in gradients. I found it confusing when mixed in a single section.

P13L21: a possibility . . .

P13L22: trends in the total amplitude value of the gradients (I don't understand what is meant exactly by the trend in the total amplitude value of gradients. It would be helpful to clarify it here)

P13L22: A positive trend in the amplitude will occur if there is an increase in the variability at the side which can happen even if there are no trend. . . (confused again how to understand the meaning of the sentence).

P15Sec5: Consider modifying Sect 5.1 title by adding WVR so it is easier to distinguish which paragraphs compare GPS to WVR, and GPS to VLBI (5.2). Optionally, split 5.1 into sub-sections dealing with original and averaged comparisons.

P15L5: . . . sites share several error sources . . . (suggest to specify them more)

P16Fig11: I suggest also discussing more ranges of estimated gradients, which seem different for GPS and WVR. GPS gradients are generally smaller. E.g. it could be due to constraining in GPS solution, mapping function, elevation angle cut-off, elevation-dependent weighting or other effects. Similarly, it seems for GPS vs VLBI, where VLBI gradients seem to be more smoothed than GPS, most likely due to the 6-hour temporal resolution.

P16L7: wet gradients from both GPS stations, ONSA and ONS1, . . . (suggesting for a

better clarity)

P17Fig12: it seems that giving correlation coefficients in the text is enough, without further need to show both plots which characteristics are the same as in Figure 11.

P18Fig13: Consider merging four plots into a single one with the x-axis ranging in 2013 to 2016

P22L6: we find the wet component of the gradients cause most of the variability. (removed 'to')

P22L15: if small gradient trends . . . (remove plural)

––––––––––––––––––––––––––––––––

---

## Referee Comment (RC3) · Anonymous Referee #3 · 14 Jan 2019

**General comments**

There was a time when tropospheric delays were considered as error-prone parameters that had to be corrected by meteorological observations from other instruments. Successive methodological improvements have led Zenithal Wet Delay retrieved by GNSS to be sufficiently accurate for use in meteorology and climatology. The information content in linear horizontal delay gradients estimated from space geodesy observations is the next step, the central issue that must be treated rigorously so as not to lead to misinterpretations or over interpretations whose consequences can be unfortunate for the applications that come out of them.

[Figure]

At first sight, the article is presented as providing general answers to this problem. However, the results obtained are valid only in Sweden, in a particular meteorological context (the Icelandic low pressure system), in a particular geodesic context (the poorer sampling of GPS data on the sky north of the zenith direction due to the geometry of the GPS satellite constellation which is particularly the problem at high latitude) and mainly using the statistical notion of correlation coefficient. These results deserve to be verified on a global scale even if this study is an interesting intermediate step. However, the title should reflect the true scope of this study. Moreover it would have been interesting to provide more bibliographic references to list the results previously obtained for other regions and to compare them with the results of this study.

This study deals with the concept of total, hydrostatic or wet gradients, of pressure gradient, of temperature gradient, of water vapor gradient, of wet gradient retrieved by WVR. However, these notions are not sufficiently defined or sufficiently documented by bibliographic elements, which undermines the clarity of the article. The reproducibility of the results of this article should be facilitated: -> Where can we download GPS, WVR and ECMWF data? -> How to estimate the gradients with the WVR? -> What are the options used to process the data in detail? If there are studies, technical reports or more general articles that can provide quick and accurate answers to these questions without lengthening the article excessively: perfect, if not the addition of elements in the supplemental material could be a good option.

It would have been necessary to give some elements on the comparisons between ZWD estimated by the different techniques before deepening the comparison of the gradients.

One of the listed points of the summary is the comparison of the GPS gradients with the corresponding ones from the ECMW analyses. How GPS gradients can confirm known seasonal effects both in the hydrostatic and the wet components whereas GPS gradients are total gradients exclusively? In fact, it is an assumption that ECMWF hydrostatic gradients are reasonably accurate (P13L13). Before subtracting the ECMWF

hydrostatic gradients from the total GPS gradients, rigorously, it would have been necessary to ensure that the hydrostatic gradients calculated by the ECMWF and felt by the GPS measurements are equivalent, which seems very difficult to verify.

The main statistical comparison tool is based on the notion of linear correlation. However, the article does not explain the advantages, disadvantages and limitations of this approach without any specific bibliography for this type of study. Again, it undermines the clarity of the article and the scope of its conclusions. With just 13 lines in the article, the results of the CONT14 VLBI measurement campaign seem insufficiently exploited.

The sampling of the sky is a critical parameter according to the authors, what are the further studies to be conducted to avoid or reduce this problem?

Taking into account the preceding remarks and clarifying the points raised, the article could then serve as a reference for future studies. Here are my specific comments.

P1L1 : " . We assess the quality of estimated linear horizontal gradients in the atmospheric propagation delay " versus

P1L21 : " the reproducibility of estimated geodetic parameters " improper term ? / Repeatability?

P2 : Figure 1 is a very rough picture of the real situation. Orders of magnitude are given without any explanation. Is the Earth modeled as an infinite plane or sphericity is taken into account? " The scale heights, hs of the hydrostatic refractivity and the wet refractivity are approximately 8 km and 2 km, respectively. " 8 km / 2 km ... Proof? References?

P2L4-6 : " Gradinarsky et al. (2000) found that using different constraints for the variability of the horizontal gradient in the VLBI and GPS data analysis did not have a significant impact on the agreement with the WVR estimates. " Can you be more explicit and provide quantitative data?

P2L6-7 : " A more recent study by Li et al. (2015) reported on the improvement obtained by Using multi-GNSS constellations instead of GPS only. " Can you explicit with quantitative results?

P2L16-17 : OK it is known but provide the major references . . .

P2L18: " Hydrostatic gradients " These terms are not defined in the article and are not commonly used.

P2L18-20: unclear

P2L20: see IERS conventions (2010)

P3L1: Provide more recent references. What are the scientific questions raised by this climatic specificity?

P3L3: "Temperature and especially water vapour can show relatively much stronger horizontal gradients over small (kilometre) scales. The temporal variability is typically also much higher than that of the hydrostatic gradients, see e.g. Li et al. (2015). ": equations? References? Which temperature? Ground? Column? How are obtained these order of magnitude? [Typo : kilometer]

P3L6 : " be significant during a passage of a weather front, especially for distinct cold fronts." order of magnitude

P3L7-8: Provide references that study these phenomenons with GNSS data. P3L16-17: software and references . . .

P4L3 : Ning et al (2013) : It would be interesting to speak about an eventual update of the procedure . . . atx file should have been updated for instance . . .

P4L4 : " we calculated mean values over 15 min, 1 h, 6 h, 1 day, and 1 month. " Ok but why? Specify scientific questions in term of atmospheric processes

P4L5-7: Figure 4 shows Figure 1 is too simplistic. May be presenting the problem like this?

P4L9-11 : references or technical report to provide ? What are the problems of this technique? Advantage and limitations?

P6L1-2 : " Therefore, data taken during rain, or when the estimated amount of liquid water is >0.7 mm, are discarded from the analysis. " references ? If we do not pay attention: what are the consequences? Bias? Ok for rain but without rain : precision ?

P6L2-3 " when the WVR hardware has failed. " Why?

P7L21-2: → It would be interesting to provide a reference which explains the observations and the estimation of SWD with this instrument. It would be interesting to explain how the WVR gradients have been computed. → " where constraints with time are applied. " Specifically with your solution with GIPSY. → It would be interesting to recall what is observed and what is modeled with GNSS and WVR. Or provide references . . .

P8 Figure 7: → optionally add rainfall? → Difference between [2013,2016] and [2016,2017] about the maximum number of daily data : around 10000 / > 10000 / Homogenous methodology of WVR observation ? → Histogram?

P9L15-17 : Are you sure of the units about the constraints ?

P9L16-17 : inhomogeneity between mapping functions . . .

P9 part 3.4 : Focus on scientific and methodological questions ? Not enough details are given : impossible to reproduce the study.

P11L3 : Figure 9 : why do not use monthly running average ?

P11L4 : 10° or 20° ? What are the differences of the two GPS solutions? Which one is chosen? Why ?

P11L5 : " We can clearly see negative north gradient in the winter both in the GPS and the ECMWF results. " provide quantitative results

P11L7 : " the Icelandic low pressure system (Hewson and Longley, 1944). " Only one

reference ... 1944 . . .

P11L9-12 : add references / Why WVR data have not been used ?

P13 Table 4 : 6-hour resolution of ECMWF data It seems difficult to draw conclusions from hourly comparison between GPS and ECMWF.

P13L2-3 : " We assess the data quality, in terms of correlation coefficients, between the total GPS and ECMWF gradients estimated at the 5 GPS sites using data from 2006 to 2016. These are shown in Table 4. " The linear correlation coefficient is mainly used in this study: what are the advantages and disadvantages of the methodology followed?

P13L10-12 : " . 10 Another result worth noting is that the two sites with the highest correlation coefficients, and especially for the monthly averages, are ONSA and SPT0. These two sites are the only ones that are equipped with microwave absorbing material below the antenna. This could reduce the impact from unwanted multipath effects. The phenomenon calls for further studies. " It would be divergent with page 17 line 10 : "Comparing the results obtained for ONSA with those from ONS1 they are almost identical (in both Figures 11 and 13) meaning that in this case there is no obvious improvement from the absorbing material below the antenna on ONSA."

P13L13-15 : " Assuming that the ECMWF hydrostatic gradients, linearly interpolated between the 6 h values, are reasonably accurate we have the possibility to subtract this hydrostatic gradient from the estimated total GPS gradient in order to compare the wet gradients at these five sites " Provide a reference to justify the approach

P13L16 : " We note that when the wet gradients are averaged over one hour and one day " Did you subtract the daily average before calculating the hourly average?

P14L3 " Typically they are all well below 0.01 mm/year. " Have you tested the significance?

P15L5 " We expect that the two GPS sites share several error sources " OK, more

detail should be given about GPS errors

P15L6 " there is a significant common mode suppression of errors " GPS data have been processed by PPP ... Can you explain more what you mean by "a significant common mode suppression of errors"? Do you speak about the common modelling to process GPS data?

P15L6 : " be slightly overoptimistic. " that needs more investigation

P15 Figure 10 : There are differences that seem to be systematic over short periods of time ... (presence of dotted curves in this figure unlike Figure 11)

P15L17-18 : Amplitude of the North component versus amplitude of the East component ?

P15L4 " reduced at the order of 10 % " 10° to 20° reduces the correlation coefficients ... What happens if you reduce the cutoff from 10° to 5° or below ?

P17L2-3 " ). The other reason is the much higher variability in the time series from the WVR because no temporal constraints are used when estimating these gradients. " References about WVR and how its gradients have been estimated are necessary for a better understanding. Is it possible to add a stochastic constraint to estimate WVR gradients? I do not know if you can change the procedure to estimate tropospheric delays by WVR.

P17L6 : typo : Lu et al. (2016) Figure 8 of Lu et al. (2016)

P17L10 " there is no obvious improvement from the absorbing material below the antenna on ONSA." the cutoff angle is fixed at 10° ... The effect of the absorbing material would be shown using a lower cutoff angle.

P12-13 " ECMWF gradients compared to the KIR0, MAR6, and VIS0 sites. Our assumption is that the lack of a concrete pillar with a metal mounting plate just below the antenna on ONS1 eliminates the need for an absorber (see Figure 3). " Good

hypothesis that deserves to be confirmed: references on IGS network ?

P18 Figure 13 : The norm of monthly wet gradient as a bar plot would be interesting.

P19 part 5.2 : This part is a little disconnected from others and is not thorough enough to allow a clearer view of the contribution of VLBI to this study. More questions about the representativity of the gradients estimated by the geodetic techniques are araised. We would expect more answers on this issue.

P19 L6-7 : " We note that the agreement in general is better for the east component compared to the north " amplitudes of East and North Component ?

P19L7-8 : " where a large north gradient is not detected in the VLBI data. " How are estimated the VLBI gradients? Stochastic constraints? Impact of the 6 hour resolution? How gradients are modeled in the VLBI data processing? Step? Piecewise linear function?

P21 Figure 15 : " and the black dots are linearly interpolated VLBI results with a temporal resolution of 5 min in order to match the GPS data. " The interpolation must be consistent with the gradient modeling used for VLBI data processing. Can you clarify?

P21 Figure 15:" using mean values for the period of $\pm 3$ h around the time epochs of the VLBI values (6h:) " same remark as before: The 6-hour resampling of GPS estimates must be consistent with the gradient modeling used for VLBI data processing. Here, that implies that VLBI estimates are modeled as a step function.

P22L6-7: " When studying gradients averaged over shorter time scales, e.g. 15 min, we find the wet component of the gradients to cause most of the variability " Not exactly because you subtracted the hydrostatic gradients sampled at 6 h from the ECMWF. You did not analyze the variability of the hydrostatic gradients.

P22L10: "during the warmer, and more humid, part of the year " It would have been interesting to use IWV retrieved by GPS.

P22L11-12 " s in the east compared to the north direction. " It would have been interesting to better cross the amplitude of the gradients with the correlation coefficients obtained.

P22L12-14 " We interpret this difference to be caused by an inhomogeneous spatial sampling on the sky, which is important when we assume that the model describing linear horizontal gradients has deficiencies. The different sampling on the sky is an important issue for any comparison between different techniques. " This question remains unresolved and would have to be studied later. P22: Lack of " Data availability section"
* * *

---

## Author Response (AR1)

**Response to referees' comments on:**

**On the information content in linear horizontal gradients estimated from space geodesy observations**

*by Gunnar Elgered, Tong Ning, Peter Forkman, and Rüdiger Haas*

**Introduction**

We appreciate the referees' comments and for the time spent on the manuscript, not only for pointing out where clarifications were needed, and identifying a couple of mistakes, but also for suggesting additional comparisons that we think have made it possible to be more specific in some cases, and less specific in other cases. We first describe the major overall changes in the revised manuscript and then we give responses to the individual comments from the referees.

**Overall changes**

The original manuscript did not have any equations included. Basic equations for the atmospheric refractivity and the type of gradients assessed in the study are now added in Section 2.

The GPS data from the Onsala site during 2013-2016 were reprocessed using three different elevation cutoff angles, alternative mapping functions for the hydrostatic and wet delays, and with and without elevation dependent weighting. As pointed out by both Referees #1 and #2 the discussion paper by Kacmarik et al. (2018) in AMT had found that an elevation angle cutoff at 3° gave the best agreement for estimated gradients. We were now able to confirm this using water vapour radiometer (WVR) data. This means that the results in Section 5.1 are to a large extent new and the section is expanded, e.g. with two new Tables 6 and 7, and the new Figure 11, showing the total gradient sizes from GPS and WVR and the strong correlation between monthly means of gradient size and ZWD (these issues were suggested to discuss in the referees' comments). As suggested by Referee #2, Figure 12 in the original manuscript was removed.

Also gradients estimated from the WVR data were included in the VLBI comparison in Section 5.2 (for completeness and to confirm to what extent gradients seen by the space geodetic techniques, VLBI and GPS, were of atmospheric origin). We removed the old Figure 15, that showed correlation between VLBI and GPS gradients. We think it caused more confusion than understanding. When we now also added the WVR data it seemed reasonable to summarize these correlation coefficients in a table (Table 8 in the revised manuscript).

When WVR data were added to the 15-day long period of the CONT14 campaign we also included the new Figure 16 of the ECMWF gradients in order to show the impact (which is really small in terms of variability) of the hydrostatic gradients added to the WVR wet gradients. The variability seen in the WVR data during the CONT14 also motivated us to go into more detail presenting the zenith wet delays (new Figures 17 and 18) allowing us to discuss the fact that gradients tend to occur during changes of the air masses above the site.

In the following we deal with each referee's comments one by one. The text is colour coded roughly as follows:
Referees' comments are in black font.
Authors' general responses are in blue text and changes in the manuscript in red text.

**Referee #1:**

The study of the horizontal variability of the atmosphere is currently of great interest. Linear horizontal delay gradients are considered advanced GNSS meteorology products and it has been proved that they are a powerful tool to identify problems with GNSS data tracking. Although not yet assimilated into NWP models, they are fundamental for the reconstruction of the slant delay and, in turn, in the 3D water vapour fields derived by tomographic inversion of GNSS based slants. The analysis of the causes of the time variability on different time scales, from months to minutes, reported in the manuscript adds new insights in the research area of these advanced GNSS meteorology products. In the manuscript, tropospheric gradients estimated from GPS observations are evaluated with respect to independent techniques as WVR and VLBI and independent data as ECMWF in order to assess their quality. I think it would be interesting in future to repeat the same kind of analysis in other regions. However, I would raise the following issues, which have to be clarified prior to the publication.

1. Mapping functions: GPS and VLBI data are analysed using different mapping functions: VMF1 for GPS and Niell for VLBI. No information about the gradient mapping function is given. I guess that in GPS data processing Bar-Sever et al. (1998) gradient mapping function is used, while in VLBI Chen and Herring (1997) gradient mapping function is applied. Kacmarik et al. (2018) recommend to agree on the gradient mapping function when tropospheric gradients derived from various sources are to be compared, since a systematic effect up to 0.3 mm is observed between Bar-Sever et al. (1998) and Chen and Herring (1997) gradient mapping functions. Having this in mind, I think that information on the gradients mapping function has to be provided in the manuscript and properly discussed.

It is correct that the Bar-Sever gradient mapping function is used in all GPS solutions and that the Chen and Herring mapping function was used in the VLBI solution.
This is now clearly stated in Section 3.
We are aware of that the estimated amplitude of the gradients depend on the gradient mapping function chosen.
In the revised paper, we primarily investigated the impact of several different elevation cutoff angles on the resulting gradients. Mapping functions for gradients were not changed.

2. Elevation cut-off: GPS data are processed at 10° and 20° elevation cut-off angle, while no info is provided for VLBI. Kacmarik et al. (2018) obtained better results using 3° elevation cut-off angle and GPS+GLONASS data. I recommend to process at least 1 GPS station with 3° elevation cut-off angle and evaluate the results. Should geodetic data processed with different cut-off angles depending on the application and on the tropospheric parameter of interest (ZTD or gradient)? A comment on this is really appreciated. The manuscript is within the scope of this special issue.
Kačmařík et al (2018), Sensitivity of GNSS tropospheric gradients to processing options, Ann. Geophys. Discuss., https://doi.org/10.5194/angeo-2018-93

We reprocessed the data obtained from ONSA and ONS1 using elevation cutoff angles of 3°, 10°, and 20° from 2013 to 2016. The resulting gradients were compared to the ones obtained by primarily the WVR in Section 5.1, but the 3° solution also to VLBI gradient results in Section 5.2.
Two new tables with results were added and a short discussion is also added in the Conclusions about an optimum elevation cutoff angle.

Below specific comments.

Introduction Page 2 Line 6. I suggest adding the amount of improvement of multi-GNSS gradients compared to GPS-only gradients.

The following text is added in the revised paper "Using multi-GNSS observations, Li et al. (2015) found a significant increase in the correlation coefficients of gradient to about 0.6 when compared to ECMWF gradients, while the one for the GPS-only is usually below 0.5. In addition, they found that the RMS difference of the gradient is reduced to about 25–35 % by multi-GNSS processing."

Cause of horizontal gradient In the manuscript, the mathematical model used to describe the tropospheric path delay is not reported. This can be added in this section and the title should be changed in 'model and cause of horizontal gradient'.

Done.

Instrumentation and data Page 3 Line 11-12. Complete the sentence 'We compare ...' adding at the end '... with respect to VLBI estimates, WVR and ECMWF data' I suggest adding in the section a table summarizing the characteristics of the instruments used for the evaluation, referring to the specific sub-section for further details.

The sentence was modified, but slightly reformulated because we regard the gradients from all these sources to be estimates. When trying to design a table that would include the important characteristics from each instrument we did not end up with a solution that was sufficiently compact, so no new table was introduced.

GPS In this section, and also somewhere else in the manuscript, the term 'site' is used both referring to a local geographical area or referring to a unique geodetic marker. Please review it and use site for a local geographical area, where one or more geodetic markers are available, and station to indicate a unique geodetic marker at a site. Figure 2.

This makes sense and has been adopted in the manuscript.

You present the sky plot of GPS observation for May 12, 2014. Why did you select this specific day?

We selected this day because we have simultaneous observations from GPS and VLBI and the results for this day are further discussed in the corresponding result section.

This now explicitly pointed out in the caption to the figure.

Gradients during the CONT14 VLBI campaign Figure 14. Add mean and std of the differences (GPS-VLBI), are these values affected by the different gradient mapping function?

In the revised paper, we added Table 8 with mean and standard deviation (plus correlation coefficient) obtained when VLBI gradients are compared to GPS and WVR. Because we did not change the gradient mapping function we refer to Kacmarik et al. (2018) to point out that the estimated gradient amplitude will depend on the mapping function.

Typos
Pag.2 Line 4: delete ')' after VLBI

Corrected

Table 2 replace 'igs_1740.atx' with 'igs08_1740.atx', correct?

Corrected, also noted that igs08_1869.atx was used in the reprocessing of the 4 years of data from the Onsala site.

Pag.22 Line 20: horisontal -> horizontal?

Corrected

**Referee #2:**

As expressed in the title, the manuscript deals with the information content in linear horizontal tropospheric delay gradients estimated from space geodesy observations, namely GPS and VLBI. The topic of the manuscript is highly actual. In past, the tropospheric delay gradients were estimated mainly for improving horizontal positioning (coordinate repeatability), although it was not always clear that it improved the troposphere modelling as the gradient parameters are highly sensitive to other error sources affecting the data analyses. So far, the gradients were also rarely estimated within an operational troposphere monitoring because the information content was often either too noisy or too much smoothed by low temporal resolution or constraints. The situation is going to change in future when providing advanced tropospheric products on troposphere asymmetry monitoring, in particular with upcoming availability of more satellites from multi-GNSS constellations. Tropospheric delay gradients are also pre-requisite for delivering other products from GNSS such as retrieving slant delays, the reconstruction of three-dimensional refractivity field or generating severe weather event indicators. Attempts for developing assimilation techniques for GNSS-based tropospheric gradients emerged recently when it seems preferred way, compared the utilization of slant delays, due to the better production quality in (near) real-time. In this context, the manuscript contributes to a better understanding of how linear tropospheric delay gradients are able to characterize wet and hydrostatic effects of the neutral atmosphere on space geodetic observation analyses; both in actual situation or in a long-term trends and useable in geodetic, meteorological or climatology applications.

1. Adding brief introduction of the model of calculating gradients from the NWM would be helpful for discussion of results, e.g. gradient mapping function, distribution of raytracing points, elevation angle cut-off.
A description of the calculation of gradients from the NWM was added. However, it is our interpretation of the paper by Boehm and Schuh (2007) that the gradients are calculated using three vertical profiles of pressure, temperature, and humidity, without the use of a gradient mapping function, raytracing, or elevation angle cutoff.

2. Although the manuscript study a comparison of gradients, there are no information about gradient mapping functions used in estimating procedures of different techniques. Kačmařík et al. 2018 (submitted for discussions in ANGEO) showed that gradient mapping function could introduce systematic effects into estimated gradients. Similarly, no information about elevation-dependent weighting (if applied) was given neither for GPS nor VLBI. Please, add these information in corresponding tables or text paragraphs, and whenever useful, consider their impact in discussions as these might be more critical than the other processing settings.
In the revised paper, we added results obtained from comparisons of gradients given by different elevation cutoff angles (3°, 10°, and 20°), weighting and non-weighting, and different mapping functions (NMF and VMF1). Tables 6 and 7 and a corresponding discussion were added.

3. The differences in size of estimated gradients estimated from different techniques are not discussed. These are visible in Figure 9 between GPS and Numerical Weather Model (NWM) and in Figures 11 and 12 for GPS compared to WVR. These could be attributed to various aspects such as used gradient mapping functions, limited resolution of numerical weather model data (ECMWF), observation sampling over the sky or others. Were similar characteristics common to the other stations?
For easier reading, I would also suggest to consider splitting sections 4.2 and, optionally

5.1, into two parts with more specific subtitles for more better clarity of different comparisons, see bellow in specific comments.

The new Figure 11 helps us to discuss the (large) uncertainty in the estimated gradient amplitudes. We use ECMWF mainly to remove the hydrostatic gradients which seems to be possible to interpolate between the 6 h data points with sufficient accuracy (Li et al., 2015). The new Figures 9 and 16 showing time series of hydrostatic gradients from ECMWF also confirm the low variability during time periods of 6 h.

Section 4.2 was split into two parts. (We agree that these were two rather different topics).

We did not split Section 5.1. It has changed significantly with the testing of several different GPS solutions.

Specific comments

P1L11: 15-day long continuous

Corrected

P2L4: VLBI, GPS and WVR (remove closing bracket).

Corrected

P2L12: a 4-year period

Corrected

P2L13: a 15-day long VLBI campaign

Corrected

P2L20: over long-time scales

We are not sure about this. "long" is an adjective and "time scales" are two nouns. After talking to a native English speaker we decide to keep it as it is.

No change in the manuscript.

P6Tab2: add gradient mapping function and reference frame and PCV values applied (IGS08 or IGS14?)

The Bar-Sever et al. (1998) the mapping function was used for the gradient estimation. The reference frame is IGS08. Table 2 describing the GPS data processing has been updated with this information.

P6L6: (consider modified wording) ... estimated gradients are independent in adjacent epochs ...

Instead we write: "estimated gradients are independent of the ones estimated at adjacent epochs ..."

P9L15: ... as piece-wise linear offsets ... Do you mean representation with a piecewise linear function when represented with the interval end-point offsets? Or do you mean just constant offsets for individual intervals? Please, reword or clarify.

We changed the wording to "piecewise linear continuous function"

This is also illustrated in the new Figures 14 and 17, but of course the reader may not yet have seen these figures while reading this section.

P11L5: the overall mean negative north gradients is also partly attributed to the flattening of the earth atmosphere, see Meindl et al. 2004, I suggest to add in the discussion in this paragraph.

We do not find any statement in Meindl (2004) about the flattening of the earth atmosphere. We also find that Meindl (2004) only refer to the negative north gradients as a function of latitude. Therefore, we removed the word "pressure" from our text in order to give a correct citation.

P12 Figure 9: discuss the different ranges of gradient sizes,

Gradient sizes are now discussed together with the new Figure 11.

P13L6: by long-term averaging …
Corrected

P13L18: … at this low humidity level …
Corrected

P13L19: I suggest to add here new sub-sections for discussing long-term trends in
gradients. I found it confusing when mixed in a single section.
Done

P13L21: a possibility …
Corrected

P13L22: trends in the total amplitude value of the gradients (I don't understand what is
meant exactly by the trend in the total amplitude value of gradients. It would be helpful
to clarify it here)
The total amplitude is > 0. If the variability increases along one coordinate there will be larger
gradients, both positive and negative. This can happen without a net increase (trend) in the east and
north gradient component.
This text is rewritten and the mathematical definition of "total amplitude" is added.

P13L22: A positive trend in the amplitude will occur if there is an increase in the variability
at the side which can happen even if there are no trend … (confused again how
to understand the meaning of the sentence).
See previous comment.

P15Sec5: Consider modifying Sect 5.1 title by adding WVR so it is easier to distinguish
which paragraphs compare GPS to WVR, and GPS to VLBI (5.2). Optionally, split 5.1
into sub-sections dealing with original and averaged comparisons.
Titles are modified — additionally now also WVR data are compared in Section 5.2.

P15L5: … sites share several error sources … (suggest to specify them more)
We now mention satellite clock and orbit errors and mapping functions.

P16Fig11: I suggest also discussing more ranges of estimated gradients, which seem
different for GPS and WVR. GPS gradients are generally smaller. E.g. it could be due
to constraining in GPS solution, mapping function, elevation angle cut-off, elevation dependent
weighting or other effects. Similarly, it seems for GPS vs VLBI, where VLBI
gradients seem to be more smoothed than GPS, most likely due to the 6-hour temporal
resolution.
We think the amplitudes of the GPS and WVR gradients are now also addressed by the new Figure 11.
Concerning the VLBI gradients we think it is difficult to make a general statement about their
amplitude size. However, given that the sampling of the atmosphere is much more sparse, a short lived
gradient in combination with assumption of linear functions in 6-hour segments, will probably reduce
the variability in the estimated amplitude. We added this comment when discussing the new Figure 15.

P16L7: wet gradients from both GPS stations, ONSA and ONS1, … (suggesting for a better clarity)
Corrected

P17Fig12: it seems that giving correlation coefficients in the text is enough, without
further need to show both plots which characteristics are the same as in Figure 11.
We agree and remove the figure but keep the text.

P18Fig13: Consider merging four plots into a single one with the x-axis ranging in 2013 to 2016

We did consider this, but think that it is easier to identify the specific month and compare it between the different years. We keep it as it is.

P22L6: we find the wet component of the gradients cause most of the variability. (removed 'to')

Corrected

P22L15: if small gradient trends ... (remove plural)

Corrected

**Referee #3:**

There was a time when tropospheric delays were considered as error-prone parameters
that had to be corrected by meteorological observations from other instruments.
Successive methodological improvements have led Zenithal Wet Delay retrieved by
GNSS to be sufficiently accurate for use in meteorology and climatology. The information
content in linear horizontal delay gradients estimated from space geodesy observations
is the next step, the central issue that must be treated rigorously so as not to
lead to misinterpretations or over interpretations whose consequences can be unfortunate
for the applications that come out of them. At first sight, the article is presented as providing general
answers to this problem.
However, the results obtained are valid only in Sweden, in a particular meteorological
context (the Icelandic low pressure system), in a particular geodesic context (the poorer
sampling of GPS data on the sky north of the zenith direction due to the geometry
of the GPS satellite constellation which is particularly the problem at high latitude)
and mainly using the statistical notion of correlation coefficient. These results deserve
to be verified on a global scale even if this study is an interesting intermediate step.
However, the title should reflect the true scope of this study.

We do not agree that all results are only valid in Sweden. The meteorological processes mentioned,
such as changes of air masses and systematic patterns in the pressure, are frequent in many regions.
Also the possible instrumental effects related to anti reflecting material at the antenna is a general
result. Since we clearly state in the abstract that we only use 5 sites in Sweden we want the title
unchanged. Although we are not native English speakers it is our interpretation that the first two words
of title indicate that it is a modest contribution to a complex subject.

Moreover it would have been interesting to provide more bibliographic references to list the results
previously obtained for other regions and to compare them with the results of this study.
This study deals with the concept of total, hydrostatic or wet gradients, of pressure
gradient, of temperature gradient, of water vapor gradient, of wet gradient retrieved by
WVR. However, these notions are not sufficiently defined or sufficiently documented by
bibliographic elements, which undermines the clarity of the article. The reproducibility
of the results of this article should be facilitated: -> Where can we download GPS,
WVR and ECMWF data? -> How to estimate the gradients with the WVR? -> What are
the options used to process the data in detail? If there are studies, technical reports or
more general articles that can provide quick and accurate answers to these questions
without lengthening the article excessively: perfect, if not the addition of elements in
the supplemental material could be a good option.
It would have been necessary to give some elements on the comparisons between
ZWD estimated by the different techniques before deepening the comparison of the gradients.

We believe that we have addressed all these points in the revised manuscript. They overlap with
several of the specific points from all three referees. We will come back to these in the specific
comments below.

One of the listed points of the summary is the comparison of the GPS gradients with
the corresponding ones from the ECMW analyses. How GPS gradients can confirm
known seasonal effects both in the hydrostatic and the wet components whereas GPS
gradients are total gradients exclusively? In fact, it is an assumption that ECMWF hydrostatic
gradients are reasonably accurate (P13L13). Before subtracting the ECMWF hydrostatic gradients
from the total GPS gradients, rigorously, it would have been necessary
to ensure that the hydrostatic gradients calculated by the ECMWF and felt by
the GPS measurements are equivalent, which seems very difficult to verify.

We do say that it is an assumption. In order to a give a bit more motivation, showing that it is a
reasonable assumption, in the revised manuscript we have two new Figures (9 and 16) that show the

different behaviour both in amplitude and in temporal variability of the wet and hydrostatic gradients from ECMWF.
The assumption is also motivated by that when WVR gradients are correlated with GPS gradients the correlation increases when the hydrostatic gradient from ECMWF is removed from the GPS total gradients. This improvement in correlation coefficients is now explicitly given in the caption to the new Figure 13.

The main statistical comparison tool is based on the notion of linear correlation. However, the article does not explain the advantages, disadvantages and limitations of this approach without any specific bibliography for this type of study. Again, it undermines the clarity of the article and the scope of its conclusions.
We have added standard deviations (SD) and state that correlation coefficients also are presented motivated by the different gradient amplitudes. They are used in a relative sense if an agreement is better or worse given identical input data but processed differently. Although we note that the comparison of the agreement between WVR and different GPS solution give the same result using SD as using correlation coefficients.

With just 13 lines in the article,
the results of the CONT14 VLBI measurement campaign seem insufficiently exploited.
The sampling of the sky is a critical parameter according to the authors, what are the further studies to be conducted to avoid or reduce this problem?
We have now added WVR gradients and make additional comparisons (see the overall changes at the beginning of the document).
We have added a short paragraph about the future of geodetic VLBI with the VGOS system.

Taking into account the preceding remarks and clarifying the points raised, the article could then serve as a reference for future studies.

Here are my specific comments.

P1L1 : " . We assess the quality of estimated linear horizontal gradients in the atmospheric propagation delay " versus
The "versus" comes in the 3$^{rd}$ sentence of the abstract which should be fine, or did we not understand?

P1L21 : " the reproducibility of estimated geodetic parameters " improper term ? / Repeatability?
According to e.g. NIST https://www.nist.gov/pml/nist-tn-1297-appendix-d-clarification-and-additional-guidance, the so called repeatability conditions are:
- the same measurement procedure
- the same observer
- the same measuring instrument, used under the same conditions
- the same location
- repetition over a short period of time

We think that for long time series of geodetic parameters, the term reproducibility should be used.

P2 : Figure 1 is a very rough picture of the real situation. Orders of magnitude are given without any explanation. Is the Earth modeled as an infinite plane or sphericity is taken into account? " The scale heights, hs of the hydrostatic refractivity and the wet refractivity are approximately 8 km and 2 km, respectively. " 8 km / 2 km ... Proof? References?

With the added theoretical background we can now refer to Equation (2) for the refractivities and we pointed out that there are large variations as well as that the sampling of the volume can and will be different for different instruments. The text of the figure is rewritten saying that it is a sketch, that the scale heights vary a lot, and referring to Eq. (2). We think every reader knows that the mapping functions used later in the modelling takes the sphericity of the earth into account, or ...?

P2L4-6 : " Gradinarsky et al. (2000) found that using different constraints for the variability of the horizontal gradient in the VLBI and GPS data analysis did not have a significant impact on the agreement with the WVR estimates. " Can you be more explicit and provide quantitative data?
The above sentence is replaced by:
Gradinarsky et al. (2000) found that when varying the constraint for the gradient variability from 0.2 to 5.6 mm/$\sqrt{\rm h}$ the weighted root-mean-square (rms) difference compared to the WVR gradients varied between 0.8 and 1.0 mm for both the GPS and the VLBI gradients.

P2L6-7 : " A more recent study by Li et al. (2015) reported on the improvement obtained by Using multi-GNSS constellations instead of GPS only. " Can you explicit with quantitative results?
The following text replaces the one above in the revised paper "Using multi-GNSS observations, Li et al. (2015) found a significant increase in the correlation coefficient to about 0.6 when compared to ECMWF gradients, while the one for the GPS only was typically below 0.5. In addition, they found that the RMS difference of the gradient is reduced to about 25–35 % by multi-GNSS processing."

P2L16-17 : OK it is known but provide the major references ...
A reference to Rüeger (2002) is added together with the equation for refractivity and additional references follow in the same section.

P2L18: " Hydrostatic gradients " These terms are not defined in the article and are not commonly used.
These are now defined in the theoretical background in the beginning of Section 2.

P2L18-20: unclear
See the response above.

P2L20: see IERS conventions (2010)
We added also this reference in the theoretical background text describing the model.

P3L1: Provide more recent references. What are the scientific questions raised by this climatic specificity?
It is a well known meteorological feature (that affects space geodesy data). It is a confirmation that the gradients estimated from space geodesy data are correct (in this sense).
Two additional references are added.

P3L3: "Temperature and especially water vapour can show relatively much stronger horizontal gradients over small (kilometre) scales. The temporal variability is typically also much higher than that of the hydrostatic gradients, see e.g. Li et al. (2015). ": equations? References? Which temperature? Ground? Column? How are obtained these order of magnitude? [Typo : kilometer]
Equations have been added in Section 2.
We have also added the two new Figures 9 and 16, showing time series of hydrostatic and wet gradients from the ECMWF data.
We use British English and according to our dictionary the spelling is "kilometre"

P3L6 : " be significant during a passage of a weather front, especially for distinct cold fronts." order of magnitude

With the extension of Section 5.2 in the revised manuscript gradients associated with the change of air masses are shown. This is however, not very helpful here in the introductory part. Since we did not find a reference of gradients estimated during the passage of a "distinct cold front". We changed the text and referred to Kacmarik et al. (2018) and their estimated gradients during an occlusion front.

P3L7-8: Provide references that study these phenomenons with GNSS data.

We are not aware of published results that identify estimated GNSS gradients with a specific source, except for the fronts mentioned above. The following examples mention meteorological phenomena that have horizontal variability in the partial pressure of water vapour and with the equations now included in the beginning of Section 2, this should be clear. The text is rephrased and now first mention variability in the partial pressure of water vapour.

P3L16-17: software and references ...
Table 2 is expanded and updated

P4L3 : Ning et al (2013) : It would be interesting to speak about an eventual update of the procedure ... atx file should have been updated for instance ...

The atx file has been updated. This is specified in Table 2, including a footnote for the reprocessing done for the 4 years of data from the Onsala site for the revised version of the manuscript.

P4L4 : " we calculated mean values over 15 min, 1 h, 6 h, 1 day, and 1 month. " Ok but why? Specify scientific questions in term of atmospheric processes
Examples of atmospheric processes that affect the 3D refractivity over time were mentioned in the previous section. The 1 h was a mistake. The new text is:
" we calculated mean values over 15 min, 6 h, 1 day, and 1 month
in order to match the temporal resolution of the comparison data
and to study the variability of the wet and the hydrostatic gradients over different time scales."

P4L5-7: Figure 4 shows Figure 1 is too simplistic. May be presenting the problem like this?
We now point out the issue of sampling already in the caption to Figure 1.

P4L9-11 : references or technical report to provide ? What are the problems of this technique? Advantage and limitations?
A general reference to Elgered and Jarlemark (1998) is added. In the caption to Figure 6 we now mention that observations cannot be done in directions close to the sun and not below elevation angles of 20°.

P6L1-2 : " Therefore, data taken during rain, or when the estimated amount of liquid water is >0.7 mm, are discarded from the analysis. " references ? If we do not pay attention: what are the consequences? Bias? Ok for rain but without rain : precision ?
A reference to Westwater and Guiraud (1980) is added and we especially mention "large positive errors in the wet delay".

P6L2-3 " when the WVR hardware has failed. " Why?
We added brief information for to 2 long data gaps:
The first long data gap in 2014–2015 was caused by a broken mechanical waveguide switch
and the second long gap in 2015–2016 was due to broken cables in the so called cable wrap. The cable wrap was redesigned.

P7L21-2: → It would be interesting to provide a reference which explains the observations

and the estimation of SWD with this instrument. It would be interesting to explain
how the WVR gradients have been computed. → " where constraints with time are
applied. " Specifically with your solution with GIPSY. → It would be interesting to recall
what is observed and what is modeled with GNSS and WVR. Or provide references ...
The GNSS gradients were estimated using a random walk model with a standard deviation (SD) of 0.3
mm/sqrt(h)) that was taken from Bar-Sever et al. (1998). A reference to Jarlemark et al. (1998) is
added in Table 2 for the GPS constraint for the ZWD.
The reference to Elgered and Jarlemark (1998) above and references therein explains the SWD/ZWD
calculation with the WVR. We state that the WVR gradients use the "four-parameter model" in Davis
et al. (1993).

P8 Figure 7: → optionally add rainfall? → Difference between [2013,2016] and
[2016,2017] about the maximum number of daily data : around 10000 / > 10000 /
Homogenous methodology of WVR observation ? → Histogram?
We think it would be too detailed to add rainfall information (since it occurs often and is the major
cause for data loss except for the 2 long gaps). We now note that in the text and stress that the same
observation sequence is used over the 4-year period as well as mention that observations close to the
sun are removed (in the caption to Figure 7).

P9L15-17 : Are you sure of the units about the constraints ?
Yes, we now also note that they are not modelled as stochastic processes.

P9L16-17 : inhomogeneity between mapping functions ...
In the revised paper, we also processed the GPS data using the same mapping function (NMF) and
found that NMF or VMF1 did not have a large impact on the gradient correlation with the WVR.

P9 part 3.4 : Focus on scientific and methodological questions ? Not enough details
are given : impossible to reproduce the study.
Details have been added both on the motivation for the CONT campaigns and missing information
about gradient mapping function and elevation cutoff angle.

P11L3 : Figure 9 : why do not use monthly running average ?
We could have done that but chose to illustrate the 11 years of data for each month to visualize any
seasonal variation.

P11L4 : 10° or 20° ? What are the differences of the two GPS solutions? Which one
is chosen? Why ?
In the revised paper, we added more results on the gradients given by different GPS solutions, i.e.,
different elevation cutoff angles, weighting and non-weighting, different mapping functions. We used
the gradients given by the GPS 3° cutoff angle solution for all other comparison due to a better
agreement when compared to WVR gradients.

P11L5 : " We can clearly see negative north gradient in the winter both in the GPS and
the ECMWF results. " provide quantitative results
We added -0.2 mm as a mean value.

P11L7 : " the Icelandic low pressure system (Hewson and Longley, 1944). " Only one reference ...
1944 ...
Two more recent references were added in Section 2. Here we refer back to this.

P11L9-12 : add references / Why WVR data have not been used ?
Two additional sea breeze references were added in Section 2. Unfortunately, WVR data from
scanning the sky are not available for the 11-year period only for 2013-2016. We say that sea breeze is

one possible cause, we do not claim that this is what we see. Perhaps in a future collaboration with meteorological expertise we could study sea breeze conditions using both GNSS and the WVR.

P13 Table 4 : 6-hour resolution of ECMWF data It seems difficult to draw conclusions from hourly comparison between GPS and ECMWF.
This was a mistake — thank you for making us aware of it. It is corrected to "six hourly".

P13L2-3 : " We assess the data quality, in terms of correlation coefficients, between the total GPS and ECMWF gradients estimated at the 5 GPS sites using data from 2006 to 2016. These are shown in Table 4. " The linear correlation coefficient is mainly used in this study: what are the advantages and disadvantages of the methodology followed?
Correlation coefficients are used in a relative sense by comparing the agreement at different stations. In Section 5 where we also use WVR data (that in general result in larger gradient amplitudes) we also use standard deviation to compare the agreements. The results are consistent.

P13L10-12 : " . 10 Another result worth noting is that the two sites with the highest correlation coefficients, and especially for the monthly averages, are ONSA and SPT0. These two sites are the only ones that are equipped with microwave absorbing material below the antenna. This could reduce the impact from unwanted multipath effects. The phenomenon calls for further studies. " It would be divergent with page 17 line 10 : "Comparing the results obtained for ONSA with those from ONS1 they are almost identical (in both Figures 11 and 13) meaning that in this case there is no obvious improvement from the absorbing material below the antenna on ONSA."
We give a possible explanation for this: "Our assumption is that the lack of a concrete pillar with a metal mounting plate just below the antenna on ONS1 eliminates the need for an absorber"

P13L13-15 : " Assuming that the ECMWF hydrostatic gradients, linearly interpolated between the 6 h values, are reasonably accurate we have the possibility to subtract this hydrostatic gradient from the estimated total GPS gradient in order to compare the wet gradients at these five sites " Provide a reference to justify the approach
See above — actually there was no interpolation. It is the "six hourly" values that are compared. We now refer to this comparison to be "six hourly"? This is done in both Table 4 and Table 5.

P13L16 : " We note that when the wet gradients are averaged over one hour and one day " Did you subtract the daily average before calculating the hourly average?
The 6 hour averages are the averages of the gradients over the 6 h. No daily average was subtracted. As mentioned above, hourly is changed to 6 h.

P14L3 " Typically they are all well below 0.01 mm/year. " Have you tested the significance?
No we did not, but we should have done that. Now we have calculated two different formal uncertainties and expanded the text. It is clear (we think) that no atmospheric gradient trend has been detected, but we still mention the highest trend estimated, and warn for instrumental effects when searching for trends in the future.

P15L5 " We expect that the two GPS sites share several error sources " OK, more detail should be given about GPS errors
We now mention satellite clock and orbit errors when observing the same satellites and we use the same mapping function (same as Referee #2)

P15L6 " there is a significant common mode suppression of errors " GPS data have been processed by PPP ... Can you explain more what you mean by "a significant common mode suppression of errors"? Do you speak about the common modelling to

process GPS data?

Yes, a common modelling and common errors associated with the same observed satellites (see the previous comment).

P15L6 : " be slightly overoptimistic. " that needs more investigation

We removed "slightly". It is just overoptimistic, because of the shared common errors mentioned above, but we cannot quantify how much.

P15 Figure 10 : There are differences that seem to be systematic over short periods of time ... (presence of dotted curves in this figure unlike Figure 11)

In the new Figure 12, we plot the GPS gradients with the original temporal resolution of 5 min. When compared to WVR data, in Figure 13 (middle and right), the GPS gradients were averaged to a 15 min temporal resolution in order to match the WVR data. We now state this in the figure captions.

P15L17-18 : Amplitude of the North component versus amplitude of the East component?

We did several plots of east versus north gradients and found that they were not correlated at all, so we did not pursue this further.

P15L4 " reduced at the order of 10 % " 10° to 20° reduces the correlation coefficients ... What happens if you reduce the cutoff from 10° to 5° or below ?

The question is dealt with in the new GPS solutions using cutoff angles of 3°, 10°, and 20°. Section 5.1 is significantly expanded with these new results.

P17L2-3 " ). The other reason is the much higher variability in the time series from the WVR because no temporal constraints are used when estimating these gradients. " References about WVR and how its gradients have been estimated are necessary for a better understanding. Is it possible to add a stochastic constraint to estimate WVR gradients? I do not know if you can change the procedure to estimate tropospheric delays by WVR.

The description of the WVR and the estimated gradients is updated in Section 3. In principle, there is nothing that would not allow constraints to be applied in the WVR gradient estimation. However, that is a big effort (at least for us). We choose to discuss this as possible future work in the Conclusions section.

P17L6 : typo : Lu et al. (2016) Figure 8 of Lu et al. (2016)

Corrected

P17L10 " there is no obvious improvement from the absorbing material below the antenna on ONSA." the cutoff angle is fixed at 10° ... The effect of the absorbing material would be shown using a lower cutoff angle.

We processed the ONSA data using a 3° elevation cutoff angle and the result is the same, still there is no clear difference between ONSA and ONS1.

P12-13 " ECMWF gradients compared to the KIR0, MAR6, and VIS0 sites. Our assumption is that the lack of a concrete pillar with a metal mounting plate just below the antenna on ONS1 eliminates the need for an absorber (see Figure 3). " Good hypothesis that deserves to be confirmed: references on IGS network ?

Sorry, we have not been able to find any independent confirmation, someone has to be first ... However, to be fair, when the Swedish Mapping, Cadastral and Land Registration Authority designed the monument for ONS1 it was suspected that the original design of monuments in the SWEPOS reference network was not optimal.

P18 Figure 13 : The norm of monthly wet gradient as a bar plot would be interesting.

In the original manuscript we did not have any time series plot in Section 5.1. This comment, plus a few other comments, addresses the issue of the different gradient amplitudes. The new Figure 11 address these issues. Here the absolute value of the monthly means of the wet gradients are shown together with the ZWD monthly means from ONSA, ONS1, and the WVR. The idea is that these qualitative graphs show correlations that are clear to the eye offer a better understanding compared to additional correlation plots.

P19 part 5.2 : This part is a little disconnected from others and is not thorough enough to allow a clearer view of the contribution of VLBI to this study. More questions about the representativity of the gradients estimated by the geodetic techniques are araised. We would expect more answers on this issue.
When we added the WVR data for comparisons during this 15-day period we find that the temporal resolution of 6 h is limiting the study. The main result, we think, that is discussed in the updated text, is that we can try to motivate additional gradient studies using the upcoming VGOS.

P19 L6-7 : " We note that the agreement in general is better for the east component compared to the north " amplitudes of East and North Component ?
Better language: "Again we note that the agreement, in terms of correlation coefficients, is better for the east component compared to the north component."

P19L7-8 : " where a large north gradient is not detected in the VLBI data. " How are estimated the VLBI gradients? Stochastic constraints? Impact of the 6 hour resolution? How gradients are modeled in the VLBI data processing? Step? Piecewise linear function?
The text in Section 3 has been updated to describe that the gradients are estimated as piecewise linear continuous functions and not a stochastic process.

P21 Figure 15 : " and the black dots are linearly interpolated VLBI results with a temporal resolution of 5 min in order to match the GPS data. " The interpolation must be consistent with the gradient modeling used for VLBI data processing. Can you clarify?
The figure has been removed from the manuscript. Instead a table with correlation coefficients for VLBI-GPS and VLBI-WVR has been added.

P21 Figure 15:" using mean values for the period of ±3 h around the time epochs of the VLBI values (6h:) " same remark as before: The 6-hour resampling of GPS estimates must be consistent with the gradient modeling used for VLBI data processing. Here, that implies that VLBI estimates are modeled as a step function.
Since the figure is removed it does not matter. (However, it can be consistent if also the continuous VLBI segments are averaged around the given value ± 3 h.)

P22L6-7: " When studying gradients averaged over shorter time scales, e.g. 15 min, we find the wet component of the gradients to cause most of the variability " Not exactly because you subtracted the hydrostatic gradients sampled at 6 h from the ECMWF. You did not analyze the variability of the hydrostatic gradients.
We have now added a plot (Figure 16) with ECMWF hydrostatic and wet gradients in Section 5.2. It is not 100 % safe but looking at the change from one 6-hour value to the next indicates that not much is happening in between.

P22L10: "during the warmer, and more humid, part of the year " It would have been interesting to use IWV retrieved by GPS.
The new Figure 11 addresses this issue. (ZWD is roughly proportional to the IWV.)

P22L11-12 " s in the east compared to the north direction. " It would have been interesting

to better cross the amplitude of the gradients with the correlation coefficients obtained.

We do not understand what is meant, unless it is covered by the new Figure 11 in combination with the old figure with the new number 14? No other action has been taken related to this comment.

P22L12-14 " We interpret this difference to be caused by an inhomogeneous spatial sampling on the sky, which is important when we assume that the model describing linear horizontal gradients has deficiencies. The different sampling on the sky is an important issue for any comparison between different techniques. " This question remains unresolved and would have to be studied later.

Yes, that is what we meant. We have added the suggested sentence.

P22: Lack of " Data availability section"

We now give the IP addresses for the input data. The gradient time series estimated by us using GPS, VLBI, and WVR data have been archived and approved in terms of the documentation by the Swedish National Data Service (SND) and a doi number will be sent to us within days, so that it can be published in the final version. For the time being a compressed file is available via DropBox: https://www.dropbox.com/s/lg4sctpm6qrfto4/Gradient%20data.zip?dl=0

[revised manuscript text omitted]

**2 Cause of horizontal gradients and models**

The delay of space geodetic signals propagating through the atmosphere depends of the refractive index $n$. Because the values are typically just above 1 it is practical to define the refractivity, $N = 10^6 \left(n - 1\right)$. A common expression used for the refractivity is (Rüeger, 2002)

$$N = k_1 \frac{p_d}{T} + k_2 \frac{e}{T} + k_3 \frac{e}{T^2} \tag{1}$$

where $p_d$ is the partial pressure of the dry constituents of air in hPa, $e$ is the partial pressure of water vapour in hPa (i.e., the total pressure $P = p_d + e$), and $T$ is the absolute temperature in K. The values of $k_1$, $k_2$, and $k_3$ are estimated from laboratory

experiments. For space geodetic applications it is meaningful to combine the terms resulting in two refractivity components, often referred to the hydrostatic ($N_h$) and the wet ($N_w$) components (Davis et al., 1985):

$$N = N_h + N_w = k_1 \frac{P}{T} + k_3' \frac{e}{T^2} \tag{2}$$

5     In order to define the integrated horizontal delay gradient, that can be inferred from ground-based observations, we follow Davis et al. (1993) and start by expressing the refractivity as a function of the height $z$, the horizontal vector $\boldsymbol{x}$, consisting of one east and one north component, and the time $t$:

$$N(\boldsymbol{x}, z; t) = N_\circ(z; t) + \boldsymbol{\xi}(z; t) \cdot \boldsymbol{x} \tag{3}$$

where $N_\circ(z; t)$ is the vertical profile of the refractivity at $\boldsymbol{x} = 0$ and $\boldsymbol{\xi}(z; t)$ is the vertical profile of the horizontal gradient at
10   $\boldsymbol{x} = 0$:

$$\xi_i(z; t) = \left. \frac{\partial N(\boldsymbol{x}, z; t)}{\partial x_i} \right|_{\boldsymbol{x} = 0} \tag{4}$$

where the index $i = 1, 2$ denotes the east and the north direction, respectively. This leads to the following expression for the integrated horizontal delay gradient:

$$\boldsymbol{\Xi} = 10^{-6} \int\limits_0^\infty dz \; z \; \boldsymbol{\xi}(z) \tag{5}$$

15   Hence, the vector $\boldsymbol{\Xi}$ has one east and one north component, which in turn can be separated into one hydrostatic and one wet component according to Eq. (2). The atmospheric volume that is more or less homogeneously sampled by ground-based sensors such as a GNSS station, a VLBI station, or a WVR, is illustrated in Figure 1.

Hydrostatic gradients are usually dominated by pressure gradients and exist mainly over global and regional scales (e.g.  synoptic scale weather systems). For example the north gradient has a clear dependence  on latitude
20   when averaged over long time scales. This has been shown by Meindl et al. (2004) using GPS data. For the area of interest in this study we specifically mention the Icelandic low pressure system that typically evolves in the winter and disappears in the summer, e.g. (Hewson and Longley, 1944; Thompson and Wallace, 1998; Sanchez-Franks et al., 2016).

Temperature and especially water vapour can show  strong horizontal gradients over small (kilometre) scales  and the temporal variability is typically also much higher than that of the hydrostatic gradients, see e.g.
25   Li et al. (2015). Hence, the large local gradients over a  station are mainly caused by the variability in water vapour and the wet refractivity. Gradients can be significant during a passage of a weather front, e.g.

[Figure]

**Figure 1.**  A sketch showing that the atmospheric volume determining the estimated atmospheric parameters is a cone originating at the upward pointing arrow. The typical scale heights, $h_s$ of the hydrostatic refractivity and the wet refractivity are  of the order of 8 km and 2 km, respectively, but vary a lot globally and in time according to Eq. (2). 
[revised manuscript text omitted]
  $\left( \sqrt{\Xi_e^2 + \Xi_n^2} \right)$ at the station. A positive trend in the  total amplitude can also occur if there is an increase in the variability at the  station, which can happen even if there  is no trend, neither in the east, nor in the north  gradient. For these five  stations we have estimated linear gradients in the east and the north direction as well as for the total gradient amplitudes over the 11 years. The trends are indeed very small, typically well below 0.01 mm/year. The highest value is −0.02 mm/year for the wet gradient in the north direction at the SPT0  station. If this trend originates from the atmosphere it is a local effect because it is 6 times as large as the wet north gradient trend at the nearby ONSA  station. A typical formal 1-sigma uncertainty of 0.01 mm/year is obtained if we assume that the deviations from the model is white noise, but 0.04 mm/year is estimated by taking the short term temporal correlation of the deviations into account using the model presented by *Nilsson and Elgered* (2008). In addition to that the estimated trends are small relative to their uncertainties we cannot assume perfectly stable hardware at the  station. For example, hardware problems giving a large impact on the estimated gradients have been reported by Douša et al. (2017). Given these circumstances it seems unlikely to detect any trends in gradients caused by the atmosphere unless there is a dramatic local effect of the weather conditions at the site.

**5 Gradients at the Onsala site**

**5.1 Wet gradients from GPS and WVR**

For the Onsala site we have total gradients from the two GPS stations, hydrostatic and wet gradients from ECMWF, and
10 wet gradients from the WVR  for the time period 2013–2016. Gradients in the east and the
north  directions are estimated from the GPS data for five different solutions. We use three different elevation cutoff
angles for the VMF1 zenith delay mapping functions. One additional solution is carried out with elevation dependent weighting
$(\sin(\varepsilon))$ and in the fifth solution the VMF1 mapping functions are replaced by the NMF. As stated earlier the gradient mapping
function presented by Bar-Sever et al. (1998) is used in all cases.
15

 When we use the independent WVR data to assess the quality of the gradients estimated
20 for ONSA and ONS1, the hydrostatic gradients from ECMWF (see Figure 9), linearly interpolated to match the time epochs
of the GPS gradients, are subtracted from the estimated total GPS gradients. Thereafter we form 15 min averages for the east
and the north wet gradients from GPS and compare to the corresponding WVR results.  An overview of
the data is presented in Figure 11 in terms of monthly means of total gradient size and ZWD. The GPS solution is the one
with a 3° elevation cutoff angle, no weighting, and the VMF1 mapping functions. Here we note that the WVR gives much
25 larger gradients. This depends mainly on that no constraints are applied in the WVR data analysis. The WVR gradients for
one 15-minute period do not depend on earlier or later estimates. The WVR is also less of an all weather instrument, being
sensitive to liquid water in the sensed atmosphere. This is likely the cause for positive systematic errors in the ZWD as well
as occasional overestimates of gradient amplitudes. In this context the 17 % difference in gradient amplitude depending on
mapping function used reported by Kačmařík et al. (2018) is less critical. The uncertainty of the estimated gradient amplitude,
30 to which the assumption of a linear model for the atmosphere is also contributing, is significantly larger.

The results for the different GPS solutions are summarised in Tables 6 and 7. Because of the different gradient amplitudes
from the WVR and GPS, we present mean values and SD of the differences as well as correlations coefficients. Table 6 shows
the results when the total gradients from the stations ONSA and ONS1 are compared to each other. Table 7 shows the results
when the wet gradients from ONSA and ONS1 are compared to the WVR gradients. We note that in both tables the best
agreement between the gradients estimated is obtained for an elevation cutoff angle equal to 3°. This confirms the results
presented by Kačmařík et al. (2018) using a GNSS station network in central Europe. This result was not expected by us, given
that the WVR has an elevation cutoff angle of 20° (in order to avoid ground-noise pickup) the GPS solution using the same
5 cutoff angle would show a better agreement. One interpretation of this result is that for the temporal resolutions of 5–15 min the
low elevation observations are important in order to distinguish the gradient parameters relative to other estimated parameters

[Figure]

**Figure 11.** Time series of monthly means of wet total gradients, $\sqrt{\Xi_{e,wet}^2 + \Xi_{n,wet}^2}$ (top) and ZWD (bottom) from GPS and WVR. When forming monthly means the correlation coefficients become high. The total wet gradients: WVR vs. ONSA/ONS1 are both 0.85 and ONSA-ONS1 is 0.99. For the ZWD: WVR vs. ONSA/ONS1 are both 0.97 and ONSA-ONS1 is 0.999. When correlating the wet total gradients with the ZWD we obtain 0.95 for the WVR, 0.93 for ONSA and 0.92 for ONS1.

[revised manuscript text omitted]
 15.  The addition of the hydrostatic gradients from ECMWF to the WVR wet gradients did not add any significant variability to the WVR gradients, see Figure 16.

Again we note that the size of the WVR gradients is larger compared to all other instruments. The VLBI gradients correlate with the gradients from the other instruments but their amplitudes are smaller. Given that the sampling of the atmosphere is much more sparse with the VLBI telescope, a short lived gradient in combination with the assumption of linear functions in 6-hour segments, will probably reduce the variability in the estimated amplitude.

[Figure]

**Figure 15.** Time series of estimated total gradients during the VLBI CONT14 campaign 6–20 May (days 126–140). The temporal resolution is 6 h for the VLBI gradients (blue circles connect with a solid line), 5 min for the GPS gradients for ONSA (red dots) and ONS1 (black dots), and 15 min for the WVR wet plus the ECMWF hydrostatic gradients (green plus). Also included are the ECMWF total gradients with a 6 h resolution (cyan squares).

Table 8 summarises the correlation coefficients for the east and the north VLBI gradients compared to those from the two GPS  stations, ONSA and ONS1,  and the WVR. Here we have correlated averages using

[Figure]

**Figure 16.** Time series of ECMWF hydrostatic and wet gradients during the CONT14 campaign.

data ±3 h around the VLBI gradient value every 6 h. In order to be consistent also the interpolated data from continuous VLBI segments are averaged in this way.

Again we note that the agreement, in terms of correlation coefficients, is better for the east component compared to the north component. There is a specific example seen in Figure 15 during day 132 (May 12) where a large north gradient is not detected in the VLBI data. The independent wet gradients obtained from the WVR (plus the ECMWF hydrostatic gradient) confirm that this gradient was originating from the atmosphere. The left plot in Figure 8 may explain why the north gradient has a larger uncertainty at this specific time.

We attribute the  lower correlation coefficients obtained between  VLBI-GPS and VLBI-WVR using 6 h averages during the CONT14 campaign compared to GPS-WVR 15 min averages for the month of May 2014 in Figure 14 to the sparse sequential sampling of the sky by the VLBI observations. On the other hand, averaging the WVR gradients over ±3 h reduces some of the noise seen in the 15 min values. The future use of the twin telescopes with faster slewing speeds at the site is likely to improve this situation. During CONT14 there were approximately 360 useful observations at Onsala per day. We expect this to increase by a factor of 6–7 when using the new VLBI Geodetic Observing System (VGOS) (Niell et al., 2018), which means that the use of twin telescopes could result in 200 observations per hour. This in turn makes it possible to improve the temporal resolution of the estimated atmospheric  gradients.

Finally, we like to use this 15-day long time series for a discussion on gradient variability. Figure 17 shows the ZWD from the two GPS stations and the WVR. At the end of day 135 more humid air is starting to enter over the site. We note a sudden

**Table 8.** Comparison of estimated linear gradients from VLBI relative to GPS and WVR data.

| Reference Instrument | Mean Difference[a] | | Standard Deviation | | Correlation Coefficient | |
|---|---|---|---|---|---|---|
| | East (mm) | North (mm) | East (mm) | North (mm) | East (mm) | North (mm) |
| ONSA | 0.01 | −0.03 | 0.22 | 0.20 | 0.70 | 0.69 |
| ONS1 | 0.03 | −0.08 | 0.22 | 0.20 | 0.69 | 0.67 |
| WVR | 0.23 | −0.17 | 0.27 | 0.27 | 0.64 | 0.65 |

[a] The mean difference is VLBI−reference instrument.

short decrease, followed by a rapid increase. In Figure 5.2 we zoom in on the gradients and the ZWD during this period. Here
5  we have an example where the GPS and WVR gradients occur when different air masses pass over the site. We believe that this example illustrates a situation where GPS/GNSS data in the future can be used to evaluate high resolution numerical weather models.

[Figure]

**Figure 17.**  The zenith wet delay (ZWD) from ONSA (red dots), ONS1 (black dots), and  WVR (green plus) during the CONT14 campaign.

[Figure]

 , ECMWF total  gradients  with a 6  h  resolution (cyan squares).

 , ECMWF total  gradients  with a 6  h  resolution (cyan squares).

**Figure 18.** Zoom in on the time series with gradients  in Figure 15 and ZWD in Figure 17. The  symbols are as before: VLBI gradients (blue circles connect with a solid line),  GPS gradients for ONSA (red dots) and ONS1 (black dots), WVR wet plus the  ECMWF hydrostatic gradients (green plus)

 , ECMWF total  gradients  with a 6  h  resolution (cyan squares).

[revised manuscript text omitted]

---

## Editor Decision (ED1)

Editor comments on manuscript amt-2018-318-version 3 "On the information content in linear horizontal delay gradients estimated from space geodesy observations" by Gunnar Elgered and co-authors.

I think the authors addressed well the major questions and comments raised by the referees. Several new figures and Tables have been added which enhance the purpose and the discussion of the paper. However, due to these additions, the focus on the main questions is sometimes blurred and the reasons and objectives for the various intercomparisons are becoming unclear. I suggest some revisions to smooth the flow of the discussion and sharpen the message. Some points deserved also for further discussion.

General comments:

Goal of the study: it is stated in the Abstract and in the Introduction that the goal is to assess the quality of the GPS gradient estimates based on comparisons with independent data: ECMWF model, WVR and VLBI observations. State more clearly which scientific questions you are addressing and justify the study scenario (why you test several specific processing options) and the use of each of the data sources (how each of the data sources helps you to address part of the questions). The specific advantages/disadvantages and the complementarity of the reference data sources and their uncertainties should be introduced as well.

My feeling is that the study would be better presented as an inter-comparison/inter-validation rather than an assessment of GPS gradients because the gradients from the various reference data sources don't agree very well… I think understanding properly the uncertainties and limitations of all the data sources used in this study is the main point of the paper and it needs to be enhanced. In this respect, I suggest that you also compare the ECMWF wet gradients to the WVR data in Section 5.1. Probably this sub-section should be split in two (as also suggested by one of the reviewers) and the GPS processing tests and the other tests (GPS vs. WVR and ECMWF vs. WVR) might be addressed separately.

Section 4. Comparisons with ECMWF at seasonal scale: two features in the ECMWF gradients are highlighted at ONSA station: 1) the persistent NS hydrostatic gradient and its seasonal variation (with winter maxima); 2) the increased EW wet gradient in summer. The hydrostatic feature is attributed to the influence of the Icelandic low pressure. I think the Icelandic low is not the only pressure system influencing the surface pressure variations in the region and it is located quite a distance to the West from the study area, so its direct influence would rather be in the EW direction. If the main seasonal changes are in the NS gradient, there must be influence of other centres (e.g. Arctic Oscillation), unless the gradients are of a more local nature. Can you elaborate be a bit more on the processes involved and their impact on the regional pressure field (e.g. inspect mean sea level pressure maps from ECMWF analyses and time series of observed surface pressure in Iceland and Sweden, etc.). This would help to better understand the information content in the GPS gradients.

Regarding the 2nd feature, sea breeze circulation is hypothesized which is a rather local phenomenon driven by a temperature and pressure gradient between sea and land. Again, can you be a bit more descriptive and quantitative about the local pressure, temperature, and humidity gradients in the vicinity of the GPS stations to support this assumption? Note that all these variables are available from the ECWMF analyses.

You wrote that the results for the four other stations are identical (except KIR0, so rather write "for the three nearby stations MAR6, SPT0, and VIS0"). However, I would not expect that the changes in the wet gradients are identical because sea breeze is a local feature which depends on the

orientation of the coastline (and not all coastlines in your study area are identically oriented). Please reformulate and be more specific on which features are seen and explained at the other sites.

In Figure 10, the dispersion of the results for each month (which I interpret as internannual variability) is much larger in the GPS series than in the ECMWF series. Can you comment on this feature? Is GPS over-estimating the gradients and their variability or is ECMWF underestimating them?

Section 5 now includes a lot of new material but the objectives and methods become a bit blurred. Please add a short rationale at the beginning of each sub-section and/or before you comment the results. E.g. What is the purpose of including systematically the two collocated GPS stations ONS1 and ONSA in all graphs? Why do you compare both total gradients and wet gradients in this section? (wouldn't be better to separate hydrostatic and wet gradients? Or focus only on wet gradients?).

Water Vapour Radiometer results: the WVR gives much larger wet gradients. Your interpretation is that this is due to the absence of constraints between estimates. I don't understand how the increased variability would increase the amplitude of the mean gradients. Or in the opposite situation, why would adding constraints in the processing change the monthly means? I think it would only reduce the short-term variability. Probably the contamination from liquid water in the measurements is a better explanation (but I am not an expert in these measurements). Could they explain the spikes in Fig. 15 and 17? Did you check the results after a more severe editing of the WVR data?

P20L29: I think the uncertainty due to the mapping function is still critical. The 17% reported by Kacmarik 2018 which you mention would explain about half of the bias seen in Fig. 11. There might also be a similar uncertainty associated with the mapping function used in the estimation of the WVR gradients. Can you comment on this? (btw, the type of mapping function is not specified in Section 3.2).

You conclude that the results for a cutoff angle of 3° are the best, though the difference with a cutoff angle of 10° is very small. Is this difference statistically significant?

P21: "The solution giving the best agreement, when comparing gradients from ONSA and ONS1 data with each other, is the one with elevation dependent weighting, whereas the comparisons with the WVR, for both ONSA and ONS1 give the best agreement without weighting." This result merits further comments and interpretation.

Specific comments:

Please be consistent in the denomination of the gradients throughout the paper. In some places you write "linear horizontal gradients" (e.g. captions of Table4), or "linear gradients" (P18L10), and most of the time "gradients". I think the latter is fine.

Abstract: P1L4-5: "GPS gradients confirm known seasonal effects both in the hydrostatic and the wet components" this sentence suggests that GPS is used to confirm the seasonal feature, but maybe you rather want to write that "GPS is able to reproduce / detect / monitor" the known seasonal effects since the purpose here is to assess GPS… By the way, I have 2 more comments: 1) I am not sure the seasonal effects in delay gradients in this region is something well known (the fact that it is well represented in the ECMWF model doesn't mean that it is well known) and 2) GPS cannot confirm the hydrostatic and wet components because GPS is measuring the total gradients… So this sentence needs be reformulated.

Section 2: I think the basic definitions and modelling of gradients from the paper by Davis et al., 1993, don't need to be repeated here.  If you need to refer to specific equations and variables you can cite them from Davis' paper, or alternatively move the equations in an Appendix. In this section, mainly Equation (6) and some text from paragraph P4L7-17 might be kept and moved to the beginning of the section. Additional references worth mentioning because they describe atmospheric processes that can be sensed by the GPS gradients are given below (Koulali Idrissi et al., 2011, and Nahmani et al., 2019).

Figure 1 is a very simplified version of the reality. I suggest that you remove it as also suggested by a reviewer. Furthermore, the numerics that are given in the caption are not used in the text.

Figure 2: add latitude and longitude ticks.

Table 2: add the information on elevation depending weighting

Section 3.2: how many observations are available in each estimation batch? What is the mapping functions used for the ZWD and gradient parameters?

P9L20: "adjacent" -> "successive"?

Section 3: it would be interesting to compare the formal errors for the estimated parameters (ZWD and gradients) from the 3 techniques (GPS, WVR, VLBI).

Figure 9: not easy to see something… Maybe plot just one year for the 6-hourly data. The monthly values are more or less shown in Fig. 10, and the SD values should be given in a table similar to Table 5 (see comment below).

Figure 10: I guess each symbol in the plots is for one year, but why are more than 11 symbols plotted for some of the months? Are the two stations ONSA and ONS1 superposed maybe?

Section 4.2: could focus on wet gradients only

Add a scatter plot of GPS and ECMWF wet gradients (similar to Fig. 13) to compare the range of values from the two data sources.

P18L8: "We assess the data quality…" If you mean the quality of GPS data, this assumes that ECMWF is of higher quality which is probably not the case (this point should be discussed).

P18L15-20: Regarding the positive impact of microwave absorbing material at ONSA and SPT0, statistical significance of the differences between correlation coefficients in Table 4 should be tested to have an objective basis for this statement as the differences are very small. Moreover, it would be good to include results for ONS1 here as it is claimed later (P24-25) that thanks to the different mechanical structure at station ONS1, there is no need for microwave absorber. This has an important practical implication.

P19: Long term trends: did you estimate linear trends directly from the gradient components or from their anomalies? I guess even if the seasonal cycle is small, it should be removed. You don't write if you analyse the GPS gradient trends only. How do they compare to the ECMWF trends? How do the trends for the hydrostatic and wet components differ?

P19L8: "Given that horizontal gradients in general are small and that the larger values typically occur for a short time" how short a time do you mean here? Give numbers and associated processes.

P19L9-10: "An estimated gradient has a direction and from a time series we estimate trends for the east and the north gradients. Combining these two trends … offers the possibility to also search for trends in the total amplitude value of the gradient at the station" => "From the time series of the NS and EW gradient components we can compute trends for the two components but also for the amplitude of the gradient vector (refer to an equation that can be given in section 2 or in the Appendix)".

P19L11-12: "total amplitude value of the gradient" and "total amplitude"… => Remove "total" (amplitude or magnitude involves both components by definition). Instead use "total" when you refer to the sum of the hydrostatic and wet components. But is it really useful to show the amplitude instead of the two components here as in all other sections the two components are shown?

P19L12-15: I don't understand how a trend in amplitude can happen if there is not a trend in the EW or NS components…

P19L15-: the discussion is not easy to follow. It would help if you could include the trend estimates in Table 5.

Table 5: "horizontal wet gradient" -> "wet gradient"

I suggest that you add a similar table but with the ECMWF results. This would help to further document the consistency/differences between the two data sources at regional scale and at different time scales.

Section 5.1: the title could be changed to "test of GPS processing variants" or something similar

P20L29-30: "The uncertainty of the estimated gradient amplitude, to which the assumption of a linear model for the atmosphere is also contributing, is significantly larger." I guess by "a linear model for the atmosphere" you mean a piece-wise linear function of time? However, I don't understand the general meaning of the sentence (what is larger than what? and why?).

P21 and Fig. 11: "wet total gradient" same comments as above => "amplitude of the wet gradient"

Fig. 11: here you could also plot the standard deviations of the 15 min data for each month for the 3 data sources. I am wondering how different the standard deviations are (I guess WVR SDs are much larger). Another option would be to plot the SDs for the 2 components (NS and EW) in a separate Figure in the same format as Fig. 14.

In the captions of Fig. 11: are all these correlation numbers really useful? Here we don't have other stations to compare with (like in Table 4) and other results (Table 5) are given for EW and NS components. I suggest that you remove the numbers. From the figure it is quite clear that the 2 GPS series are in better agreement with each other than with the WVR series.

Fig. 12 and 13 could be merged.

Fig. 13: use the same x and y limits in all plots.

P24, Fig. 14 shows the results for 4 years. Can you comment on the similarities/differences between years?

Could the month-to-month and year-to-year variability in the correlation coefficients shown in Fig. 14 be due to occasional spurious values in the WVR time series?

Fig. 14: here you could add the SDs for the 2 components (NS and EW) to support the idea that the variability in the wet refractivity is larger in summer (P24).

Section 5.2: The title doesn't include ECMWF.

Why do you compare total gradients in Fig. 15? I think it is sufficient to show the hydrostatic gradients in Fig. 16 (very small variability) and compare only wet gradients in Fig. 15. However, it is difficult to distinguish the different data sources in Fig. 15. Maybe consider using different colours, or removing one GPS series, or split in two figures.

P27L4-5: "The left plot in Figure 8 may explain why the north gradient has a larger uncertainty at this specific time". I am not sure I understand the explanation. Can you be more specific?

Table 8: total gradients? 6-hourly data? Please add VLBI-ECMWF.

P27L16: Figure 17 and 18: case study of day 135-136: can you describe briefly the meteorological situation?

P28L4 "Figure 5.2" => Figure 18.

Figure 17: maybe only wet gradients are necessary here. Don't connect the VLBI symbols (it gives the impression that the variation is mis-represented whereas it is just under-sampled).

Fig. 17 could be merged with Fig. 15.

Additional references:

Please see the paper by Koulali Idrissi et al. (2011) who analysed the seasonal variations of GPS gradients in a different region, and the paper by Nahmani et al., 2019, who shows the changes in GPS gradients during the passage of MCSs.

Koulali Idrissi, A., D. Ouazar, O. Bock and A. Fadil, Study of seasonal-scale atmospheric water cycle with ground-based GPS receivers, radiosondes and NWP models over Morocco (2012) Atmos. Res., 104–105, February 2012, Pages 273–291; https://doi.org/10.1016/j.atmosres.2011.11.002

Nahmani, S., Bock, O., and Guichard, F.: Sensitivity of GPS tropospheric estimates to mesoscale convective systems in West Africa, Atmos. Chem. Phys. Discuss., https://doi.org/10.5194/acp-2018-1242, in review, 2019.

---

## Author Response (AR2)

**Response to editor's comments on:**

*On the information content in linear horizontal gradients estimated from space geodesy observations*

*by Gunnar Elgered, Tong Ning, Peter Forkman, and Rüdiger Haas*

**Introduction**

We are grateful for the many good ideas proposed as well as making us aware of a couple of mistakes. Most of them are accepted and implemented as suggested. In a few cases, we have made compromises. The details are described below where the editor's comments are in black text and our responses are in red text.

**General comments:**

Goal of the study: it is stated in the Abstract and in the Introduction that the goal is to assess the quality of the GPS gradient estimates based on comparisons with independent data: ECMWF model, WVR and VLBI observations. State more clearly which scientific questions you are addressing and justify the study scenario (why you test several specific processing options) and the use of each of the data sources (how each of the data sources helps you to address part of the questions). The specific advantages/disadvantages and the complementarity of the reference data sources and their uncertainties should be introduced as well.

A new subsection at the end of Section 3 includes formal uncertainties and specifies the comparisons made over different time scales. In Section 4 we focus on comparisons between total gradients from GPS and ECMWF data with a temporal resolution limited to 6 h. Because the ECMWF offer both wet and hydrostatic gradients we also study them separately as monthly means (Figure 9) and the trends (Table 8). In Section 5 we focus on comparisons of wet gradients from GPS and the WVR with a temporal resolution of 15 min. One exception is in Section 5.3 where we use the CONT14 VLBI campaign as a case study, although the temporal resolution of the VLBI data is 6 h.

My feeling is that the study would be better presented as an inter-comparison/inter-validation rather than an assessment of GPS gradients because the gradients from the various reference data sources don't agree very well… I think understanding properly the uncertainties and limitations of all the data sources used in this study is the main point of the paper and it needs to be enhanced. In this respect, I suggest that you also compare the ECMWF wet gradients to the WVR data in Section 5.1. Probably this sub-section should be split in two (as also suggested by one of the reviewers) and the GPS processing tests and the other tests (GPS vs. WVR and ECMWF vs. WVR) might be addressed separately.

We agree that it is difficult to state that any one of the different data sources is superior in accuracy. The word "assessment" is interpreted differently by different persons whereas a comparison is just a comparison. With this in mind we have changed the wording (avoiding the word assessment except in a couple of places and also added motivations for the work in the Introduction.

The former Section 5.1 is now also split into two sections as suggested. The ECMWF gradients did not add any information when compared to the VLBI and the WVR data (Subsection 5.3). In Subsections 5.1 and 5.2 we focus on the 15-minute data from the WVR and GPS. However, we added the Table 7 with ECMWF results for the 5 stations in Subsection 4.2 and the trends in Subsection 4.3.

Section 4. Comparisons with ECMWF at seasonal scale: two features in the ECMWF gradients are highlighted at ONSA station: 1) the persistent NS hydrostatic gradient and its seasonal variation (with winter maxima); 2) the increased EW wet gradient in summer. The hydrostatic feature is attributed to the influence of the Icelandic low pressure. I think the Icelandic low is not the only pressure system influencing the surface pressure variations in the region and it is located quite a distance to

the West from the study area, so its direct influence would rather be in the EW direction. If the main seasonal changes are in the NS gradient, there must be influence of other centres (e.g. Arctic Oscillation), unless the gradients are of a more local nature. Can you elaborate be a bit more on the processes involved and their impact on the regional pressure field (e.g. inspect mean sea level pressure maps from ECMWF analyses and time series of observed surface pressure in Iceland and Sweden, etc.). This would help to better understand the information content in the GPS gradients.

Our interpretation is that the "Icelandic low" (documented already in 1944) is a phenomenon which is a component also in the North Atlantic Oscillation and the Arctic oscillation, see Thompson and Wallace (1998). The text is expanded by mentioning all three phenomena. When studying the mean sea level pressure distribution over Sweden there is mainly a north-south component in the winter. We now, in Section 4.2, mention the ERA-40 Atlas (openly available via internet), which shows this north-south winter gradient in the mean sea level pressure very clearly.

Regarding the 2nd feature, sea breeze circulation is hypothesized which is a rather local phenomenon driven by a temperature and pressure gradient between sea and land. Again, can you be a bit more descriptive and quantitative about the local pressure, temperature, and humidity gradients in the vicinity of the GPS stations to support this assumption? Note that all these variables are available from the ECWMF analyses.

We have rewritten this part to stress that this is one possible explanation. For this paper we do not have the expertise in meteorological high-resolution models available to verify which large east gradients that clearly are associated with sea breeze conditions. (Also, we do only right now have access to the ECMWF gradients calculated by the Vienna group.) We believe that a separate study is needed in order to obtain good and reliable results. It is therefore mentioned as future work in the conclusions section.

You wrote that the results for the four other stations are identical (except KIR0, so rather write "for the three nearby stations MAR6, SPT0, and VIS0"). However, I would not expect that the changes in the wet gradients are identical because sea breeze is a local feature which depends on the orientation of the coastline (and not all coastlines in your study area are identically oriented). Please reformulate and be more specific on which features are seen and explained at the other sites.

This was a mistake. The text has been rewritten. Now we first state that the results are similar, rather than identical, and then we comment on the differences, the drier air in Kiruna and the sea breeze, or other coast line phenomena, at the Onsala site.

In Figure 10, the dispersion of the results for each month (which I interpret as internannual variability) is much larger in the GPS series than in the ECMWF series. Can you comment on this feature? Is GPS over-estimating the gradients and their variability or is ECMWF underestimating them?

It is impossible to say. However, the issue of gradient sizes estimated from GPS data is now further illuminated with the new Table 11. Based on this, one may guess that if we would have used a 3° cutoff angle also in the 11-year study, the GPS gradients would have been approximately 20 % lower.

Section 5 now includes a lot of new material but the objectives and methods become a bit blurred. Please add a short rationale at the beginning of each sub-section and/or before you comment the results. E.g. What is the purpose of including systematically the two collocated GPS stations ONS1 and ONSA in all graphs? Why do you compare both total gradients and wet gradients in this section? (wouldn't be better to separate hydrostatic and wet gradients? Or focus only on wet gradients?).

The reason to include both ONSA and ONS1 is because of the different antenna environments. We now state that in the beginning of Section 5.

It does not really matter if one studies wet or total gradients over these short time scales. Nevertheless, to make it less confusing for the reader, we now focus on total gradients in Section 4

and wet gradients in Section 5 (where we only use ECMWF hydrostatic gradients to infer the wet gradients from GPS and VLBI).

Water Vapour Radiometer results: the WVR gives much larger wet gradients. Your interpretation is that this is due to the absence of constraints between estimates. I don't understand how the increased variability would increase the amplitude of the mean gradients. Or in the opposite situation, why would adding constraints in the processing change the monthly means? I think it would only reduce the short-term variability. Probably the contamination from liquid water in the measurements is a better explanation (but I am not an expert in these measurements). Could they explain the spikes in Fig. 15 and 17? Did you check the results after a more severe editing of the WVR data?

You are correct, it is only over short time scales that the absence of constraints is relevant.
We carried out a test when editing away all data when the liquid water content > 0.3 mm. The impact on overall gradient sizes was insignificant. This finding is added to the text in a list with possible explanations, e.g. Table 11 showing the impact on the estimated gradient amplitude for the different cutoff angles.

P20L29: I think the uncertainty due to the mapping function is still critical. The 17% reported by Kacmarik 2018 which you mention would explain about half of the bias seen in Fig. 11. There might also be a similar uncertainty associated with the mapping function used in the estimation of the WVR gradients. Can you comment on this? (btw, the type of mapping function is not specified in Section 3.2).

We made an error in this discussion. Both the WVR and GPS use the Bar-Sever gradient mapping function. The mapping function for the WVR is now stated in Section 3.2. As mentioned above we now list possible explanations for differences in the gradient size from WVR and GPS data.

You conclude that the results for a cutoff angle of 3° are the best, though the difference with a cutoff angle of 10° is very small. Is this difference statistically significant?

The changes are from 0.66 to 0.68 in the east and 0.62 to 0.64 in the north component.
The 95 % confidence interval for correlations of these sizes, and approximately 80,000 data pairs is ±0.004, so yes, in a statistical sense the differences are significant. The confidence interval is now specifically mentioned.

P21: "The solution giving the best agreement, when comparing gradients from ONSA and ONS1 data with each other, is the one with elevation dependent weighting, whereas the comparisons with the WVR, for both ONSA and ONS1 give the best agreement without weighting." This result merits further comments and interpretation.

The following text is added:
"The choice of elevation cutoff angle is a compromise between having a good geometry and avoiding the effects of signal multipath. Our interpretation is that the gradients from ONSA and ONS1 are estimated based on very similar observational directions and have common error sources, such as orbit errors, resulting in correlations around 0.9. In order to increase an already high correlation the observations at the lowest elevation angles are not that important, since multipath effects will be more and more different the closer to the horizon the observations are made. When ONSA and ONS1 gradients are compared to those from the WVR the situation is different because these gradients are independent and the geometry of the GPS observations becomes more important in order to estimate a more accurate gradient, although we note that the correlation is here reduced to around 0.6."

**Specific comments:**

Please be consistent in the denomination of the gradients throughout the paper. In some places you write "linear horizontal gradients" (e.g. captions of Table4), or "linear gradients" (P18L10), and most of the time "gradients". I think the latter is fine.

Now we just have "linear horizontal gradients" in the title and when first mentioned in the abstract and in the introduction, just to be clear about which type of gradients we study. Thereafter, we just use "gradients".

Abstract: P1L4-5: "GPS gradients confirm known seasonal effects both in the hydrostatic and the wet components" this sentence suggests that GPS is used to confirm the seasonal feature, but maybe you rather want to write that "GPS is able to reproduce / detect / monitor" the known seasonal effects since the purpose here is to assess GPS… By the way, I have 2 more comments: 1) I am not sure the seasonal effects in delay gradients in this region is something well known (the fact that it is well represented in the ECMWF model doesn't mean that it is well known) and 2) GPS cannot confirm the hydrostatic and wet components because GPS is measuring the total gradients… So this sentence needs be reformulated.

The sentence in the abstract is rewritten without using the "well known" wording and instead noting that effects due to wet and hydrostatic components are detected.

Section 2: I think the basic definitions and modelling of gradients from the paper by Davis et al., 1993, don't need to be repeated here. If you need to refer to specific equations and variables you can cite them from Davis' paper, or alternatively move the equations in an Appendix. In this section, mainly Equation (6) and some text from paragraph P4L7-17 might be kept and moved to the beginning of the section. Additional references worth mentioning because they describe atmospheric processes that can be sensed by the GPS gradients are given below (Koulali Idrissi et al., 2011, and Nahmani et al., 2019).

The equations were added in the previous revision as an introduction to the model, Equation (6), where the latter was requested by the referees. They would be helpful for the less experienced reader but as the manuscript now also is much longer we agree, and keep only Equation (6) now Equation (1). We also added the definition of gradient amplitude.
The two suggested references are now complementing the previous ones.

Figure 1 is a very simplified version of the reality. I suggest that you remove it as also suggested by a reviewer. Furthermore, the numerics that are given in the caption are not used in the text.

We think that the figure is a nice introduction to make the reader aware of the different volumes being averaged. On the other hand the manuscript is now much longer — see previous comment — and the figure is removed.

Figure 2: add latitude and longitude ticks.
Corrected in the new figure.

Table 2: add the information on elevation depending weighting
Footnote "b" of the table is updated. This is now Table 1. The previous Table 1 has been moved to the end of Section 3.

Section 3.2: how many observations are available in each estimation batch? What is the mapping functions used for the ZWD and gradient parameters?
This information is added to the caption of the new Figure 5. (100 observations in 15 minutes result in 10,000 observations per day, see Figure 6.)
In order to avoid ground-noise pickup the WVR provides observations of the wet delay in the different directions above 20°. Therefore a simple sin(EL) mapping function is used to relate these

slant wet delays to the equivalent ZWD. The model for gradient estimation uses the Bar Sever gradient mapping function. This is now described in the text.

P9L20: "adjacent" -> "successive"?
We now use "successive", also in Section 5 (confirmed by a native English speaker).

Section 3: it would be interesting to compare the formal errors for the estimated parameters (ZWD and gradients) from the 3 techniques (GPS, WVR, VLBI).
We have added Subsection 3.5 in which Table 4 presents the typical formal errors for the remote sensing techniques.

Figure 9: not easy to see something… Maybe plot just one year for the 6-hourly data. The monthly values are more or less shown in Fig. 10, and the SD values should be given in a table similar to Table 5 (see comment below).
The figure was added in the previous revision due to a referee comment that we had not shown that the hydrostatic gradients were less variable compared to the wet. We see this figure as introductory material (similar to the photos of the GPS antenna installations and the WVR) since it is our input data calculated by the Vienna group. It visualizes the differences for six hourly and monthly values for the hydrostatic and wet gradients and is now referred to in the beginning of Subsection 4.3. One can argue that one year is sufficient because the different years look very similar, on the other hand, showing the four years illustrates the small differences from year to year, so we prefer to show the full four years for the Onsala site, which is later studied in Section 5.

Figure 10: I guess each symbol in the plots is for one year, but why are more than 11 symbols plotted for some of the months? Are the two stations ONSA and ONS1 superposed maybe?
A mistake was made in the GPS plot, including monthly data from before 2006, when no ECMWF data were available. A correct plot is inserted.

Section 4.2: could focus on wet gradients only
Add a scatter plot of GPS and ECMWF wet gradients (similar to Fig. 13) to compare the range of values from the two data sources.
The range of values can now be compared using the new Table 7 (ECMWF data) and Table 6 (GPS data). For the monthly mean values also the upper two plots in Figure 9 is meaningful in this sense. We produced a couple of scatter plots but decided that they did not add any additional information to the two tables.

P18L8: "We assess the data quality…" If you mean the quality of GPS data, this assumes that ECMWF is of higher quality which is probably not the case (this point should be discussed).
Correct, a better wording is "We study the differences ..."

P18L15-20: Regarding the positive impact of microwave absorbing material at ONSA and SPT0, statistical significance of the differences between correlation coefficients in Table 4 should be tested to have an objective basis for this statement as the differences are very small. Moreover, it would be good to include results for ONS1 here as it is claimed later (P24-25) that thanks to the different mechanical structure at station ONS1, there is no need for microwave absorber. This has an important practical implication.
The 95 % confidence interval for the correlation coefficient of 0.90 at ONSA is now explicitly stated. The significance of the larger value at SPT0 (compared to the other 3 stations) is less but still approximately at the 95 % level.

ONS1 data are only available for four of the years studied and since the statistical significance for the 11 years is just above what is required, 4 years of data will not be convincing. This issue should be revisited after a few years from today.

P19: Long term trends: did you estimate linear trends directly from the gradient components or from their anomalies? I guess even if the seasonal cycle is small, it should be removed. You don't write if you analyse the GPS gradient trends only. How do they compare to the ECMWF trends? How do the trends for the hydrostatic and wet components differ?
We did not remove a seasonal cycle. It would only make sense for the amplitude, but no trend is detected — the "signals" are too small.
Table 8, summarizing the estimated trend values from both ECMWF and GPS, has been added.

P19L8: "Given that horizontal gradients in general are small and that the larger values typically occur for a short time" how short a time do you mean here? Give numbers and associated processes.
New text:
"Given that gradients in general are less than 1 mm and that the larger values typically occur over time scales from minutes to a few hours ..."

P19L9-10: "An estimated gradient has a direction and from a time series we estimate trends for the east and the north gradients. Combining these two trends … offers the possibility to also search for trends in the total amplitude value of the gradient at the station" => "From the time series of the NS and EW gradient components we can compute trends for the two components but also for the amplitude of the gradient vector (refer to an equation that can be given in section 2 or in the Appendix)".
The text is modified and an equation defining the gradient amplitude is added in Section 2 as suggested.

P19L11-12: "total amplitude value of the gradient" and "total amplitude"… => Remove "total" (amplitude or magnitude involves both components by definition). Instead use "total" when you refer to the sum of the hydrostatic and wet components. But is it really useful to show the amplitude instead of the two components here as in all other sections the two components are shown
We changed the wordings as suggested. Yes, we think it is meaningful to here show the trends for both the components and the amplitude. It relates to the next comment. New text: "There can be a trend in the amplitude, even if there is no trend neither in the east nor in the north components. The amplitude is by definition never negative. A trend of larger east gradients can be balanced by a similar trend in larger west gradients resulting in no net trend in the east gradient component, but a trend in the gradient amplitude."

P19L12-15: I don't understand how a trend in amplitude can happen if there is not a trend in the EW or NS components…
See previous comment. The text is now (hopefully) more understandable.

P19L15-: the discussion is not easy to follow. It would help if you could include the trend estimates in Table 5.
We have changed the wording but did not find it easy to expand the table mixing trends and correlations. Instead the new Table 8 was added (see above).

Table 5: "horizontal wet gradient" -> "wet gradient"

I suggest that you add a similar table but with the ECMWF results. This would help to further document the consistency/differences between the two data sources at regional scale and at different time scales.

Done — this is now Table 7.

Section 5.1: the title could be changed to "test of GPS processing variants" or something similar

We have split the previous Section 5.1 into two (as suggested). The 1st one of these is now titled: "Test of GPS processing variants relative to WVR data".

P20L29-30: "The uncertainty of the estimated gradient amplitude, to which the assumption of a linear model for the atmosphere is also contributing, is significantly larger." I guess by "a linear model for the atmosphere" you mean a piece-wise linear function of time? However, I don't understand the general meaning of the sentence (what is larger than what? and why?).

This part of text is totally rewritten and the new Table 11 is used instead to show the averaging effect for different elevation cutoff angles.

P21 and Fig. 11: "wet total gradient" same comments as above => "amplitude of the wet gradient"

Done!

Fig. 11: here you could also plot the standard deviations of the 15 min data for each month for the 3 data sources. I am wondering how different the standard deviations are (I guess WVR SDs are much larger). Another option would be to plot the SDs for the 2 components (NS and EW) in a separate Figure in the same format as Fig. 14.

The SD plots are added.

In the captions of Fig. 11: are all these correlation numbers really useful? Here we don't have other stations to compare with (like in Table 4) and other results (Table 5) are given for EW and NS components. I suggest that you remove the numbers. From the figure it is quite clear that the 2 GPS series are in better agreement with each other than with the WVR series.

Yes, we agree and removed these sentences.

Fig. 12 and 13 could be merged.

These are now Figures 11 and 12. Figure 11 includes all simultaneously estimated original total gradients from ONSA and ONS1 whereas Figure 12 depicts the wet gradients and only includes the results when WVR data are available at the same time as ONSA or ONS1 data. Therefore, we prefer to have them separated and to present both of them. Then we can also make the point (in the text) about lower correlations between ONSA-ONS1 for wet gradients compared to the total gradients.

Fig. 13: use the same x and y limits in all plots.

Done!

P24, Fig. 14 shows the results for 4 years. Can you comment on the similarities/differences between years?
Could the month-to-month and year-to-year variability in the correlation coefficients shown in Fig. 14 be due to occasional spurious values in the WVR time series?

In October 2015, yes as already mentioned in the caption.
For the other months with low correlations the reason was simply that no large gradients occurred for that component during that month. This is now explicitly stated in the text.

Fig. 14: here you could add the SDs for the 2 components (NS and EW) to support the idea that the variability in the wet refractivity is larger in summer (P24).

We tried, but the plots became too "busy". With the added SDs in Figure 10 we have shown the correlations between ZWD, its SD, and the wet gradient amplitudes over the seasons.

Section 5.2: The title doesn't include ECMWF.
Why do you compare total gradients in Fig. 15? I think it is sufficient to show the hydrostatic gradients in Fig. 16 (very small variability) and compare only wet gradients in Fig. 15. However, it is difficult to distinguish the different data sources in Fig. 15. Maybe consider using different colours, or removing one GPS series, or split in two figures.
We have done the change and now study primarily the wet gradients in the whole Section 5. We increased the height of the figure (relative to the width) to improve the visibility, and included also the ZWD (as suggested below).

P27L4-5: "The left plot in Figure 8 may explain why the north gradient has a larger uncertainty at this specific time". I am not sure I understand the explanation. Can you be more specific?
This discussion was not conclusive and has been removed. Surprisingly, in spite of the fact that the hydrostatic gradients were relatively small during CONT14, the results for just the wet gradients became a bit different, which is seen when comparing the new Table 12 with the old Table 8.

Table 8: total gradients? 6-hourly data? Please add VLBI-ECMWF.
The ECMWF gradients do not add any information to the case study. Except that the hydrostatic gradients in Figure 14 are used to calculate wet gradients from the VLBI and the GPS data. The CONT14 session is too short to provide meaningful statistics. We prefer to use this example as a case study (as well as pointing out that future VLBI data with more frequent observations should improve the accuracy of the VLBI gradients).

P27L16: Figure 17 and 18: case study of day 135-136: can you describe briefly the meteorological situation?
We now describe briefly the warm front passage with some additional information about wind at the ground. We also added some text relating ZWD variations to the gradients (see also comment below on Figure 17).

P28L4 "Figure 5.2" => Figure 18.
Sorry, but we do not understand this comment/request.

Figure 17: maybe only wet gradients are necessary here. Don't connect the VLBI symbols (it gives the impression that the variation is mis-represented whereas it is just under-sampled).
Although we do not compare the ZWD time series the ZWD helps to get the picture of changing air masses which sometimes is the likely cause of the wet gradients. (This is more obvious in Figure 16 with a better temporal resolution on the x axis.) Of course, it would then have been sufficient with only one ZWD time series, but since they are available and consistent (given their uncertainties) we chose to include the WVR, and the GPS with the better temporal resolutions, but add also the VLBI gradients because the case study is defined by the VLBI experiment.
The lines between the VLBI symbols have been removed as suggested.

Fig. 17 could be merged with Fig. 15.
Done!

[revised manuscript text omitted]

$$N = k_1 \frac{p_d}{T} + k_2 \frac{e}{T} + k_3 \frac{e}{T^2}$$

experiments. . For space geodetic applications it is meaningful to combine the terms resulting in two refractivity components, often referred to the hydrostatic ($N_h$) and the wet ($N_w$) components (Davis et al., 1985):

$$N = N_h + N_w = k_1 \frac{P}{T} + k'_3 \frac{e}{T^2}$$

In order to define the integrated horizontal delay define one hydrostatic and one wet component (Davis et al., 1985). For a
5   horizontally stratified atmosphere it is then common practise to use equivalent zenith values for these components. Additionally we may define a horizontal linear gradient, that can be inferred from ground-based observations , we follow Davis et al. (1993) and start by expressing the refractivity as a function of the height $z$, the horizontal vector $\boldsymbol{x}$, (Davis et al., 1993), consisting of one east and one north component, and the time $t$:

$$N(\boldsymbol{x}, z; t) = N_\circ(z; t) + \boldsymbol{\xi}(z; t) \cdot \boldsymbol{x}$$

10   where $N_\circ(z;t)$ is the vertical profile of the refractivity at $\boldsymbol{x} = 0$ and $\boldsymbol{\xi}(z;t)$ is the vertical profile of the horizontal gradient at $\boldsymbol{x} = 0$:

$$\xi_i(z;t) = \left. \frac{\partial N(\boldsymbol{x}, z; t)}{\partial x_i} \right|_{\boldsymbol{x}=0}$$

where the index $i = 1, 2$ denotes the east and the north direction, respectively. This leads to the following expression for the integrated horizontal delay gradient:

15   $$\boldsymbol{\Xi} = 10^{-6} \int_0^\infty dz \, z \, \boldsymbol{\xi}(z)$$

Hence, the vector $\boldsymbol{\Xi}$ has one east and one north component, which in turn also can be separated, into one hydrostatic and one wet component according to Eq. (??). The atmospheric volume that is more or less homogeneously sampled by ground-based sensors such as a GNSS station, a VLBI station, or a WVR, is illustrated in Figure ??..

A sketch showing that the atmospheric volume determining the estimated atmospheric parameters is a cone originating at
20   the upward pointing arrow. The typical scale heights, $h_s$ of the hydrostatic refractivity and the wet refractivity are of the order of 8 km and 2 km, respectively, but vary a lot globally and in time according to Eq. (??). 
[revised manuscript text omitted]
. ~~An overview of the data is presented in Figure 10 in terms of monthly means of total gradient size and ZWD. The GPS solution is the one with a 3 elevation cutoff angle, no weighting, and the VMF1 mapping functions. Here we note that the WVR gives much larger gradients. This depends mainly on that no constraints are applied in the WVR data analysis. The WVR gradients for one 15-minute period do not depend on earlier or later estimates. The WVR is also less of an all weather instrument, being sensitive to liquid water in the sensed atmosphere. This is likely the cause for positive systematic errors in the ZWD as well as occasional overestimates of gradient amplitudes. In this context the 17 % difference in gradient amplitude depending on mapping function used reported by Kačmařík et al. (2018) is less critical. The uncertainty of the estimated gradient amplitude, to which the assumption of a linear model for the atmosphere is also contributing, is significantly larger.~~

~~Time series of monthly means of wet total gradients, $\sqrt{\Xi_{e,wet}^2 + \Xi_{n,wet}^2}$, (top) and ZWD (bottom) from GPS and WVR. When forming monthly means the correlation coefficients become high. The total wet gradients: WVR vs. ONSA/ONS1 are both 0.85 and ONSA-ONS1 is 0.99. For the ZWD: WVR vs. ONSA/ONS1 are both 0.97 and ONSA-ONS1 is 0.999. When correlating the wet total gradients with the ZWD we obtain 0.95 for the WVR, 0.93 for ONSA and 0.92 for ONS1.~~

[revised manuscript text omitted]

---

## Editor Decision (ED2)

Editor comments on manuscript amt-2018-318-version 4 "On the information content in linear horizontal delay gradients estimated from space geodesy observations" by Gunnar Elgered and co-authors.

I would like to congratulate the authors for the implementation of the corrections and suggestions provided in my previous report and their careful review and revision of the text and figures. I have two concerns related to some new results included in the revised manuscript and a few additional minor comments.

Table 11 shows the mean values of gradient amplitudes from two GPS stations for 3 different cutoff angles (3°, 10°, and 20°). The result is quite striking. The mean amplitude is clearly increasing when the cutoff is increasing, or the other way round, the mean amplitude is decreasing when the cutoff is decreasing. You adopt the 2$^{nd}$ point of view and speculate that the decrease is due to averaging of a larger air-mass, a result that is compared to averaging over longer time periods. I am not convinced by this explanation. I think that changing the cutoff angle has primarily an impact on the correlation between gradients and other parameters and on the accuracy of the estimated parameters. Actually, decreasing the cutoff is expected to improve the accuracy and provide more realistic gradients estimates. I think this is supported by the enhanced agreement with WVR results (the 3° GPS solution agrees better with WVR results). On the other hand, increasing the cutoff to 20° seems rather an unfavourable situation for estimating accurate gradients and it may be that the GPS gradient estimates are actually biased in this case, as well as the WVR gradient maybe? The increased standard deviation of differences and decreased correlation coefficients between GPS and WVR indeed support the idea that GPS and WVR agree less well in that case, i.e. the uncertainty in the GPS gradients is larger as also predicted by the larger formal error. I think that this effect is larger than the effect of sensing different air-masses at lower elevations. At least both points of view should be discussed and if you want to maintain your idea, more evidence should be provided to support it.

My second concern is with your statement that there can be a trend in amplitude without a trend in components (also noted in the previous review). Since the gradient is a vector, if its length increases, the components necessarily increase. This statement should be removed or clarified if I didn't get what you mean.

Below are a few additional minor suggestions.

Though you slightly extended the last paragraph describing the organisation of the paper, I think there is still a paragraph missing in the Introduction on the motivation of this work and the rationale of the study scenario. Please add a short paragraph between line 10 and 11.

P2L26-28: Meindl et al. (2004) discussed the global north-south temperature gradient. In your work, regional high and low pressure systems are more important. I suggest to remove the sentence related to work of Meindl et al. (2004), and rewrite this part as follows: "Hydrostatic gradients are usually dominated by pressure gradients and exist mainly over regional scales (e.g. persistent high and low pressure systems) and synoptic scale (e.g. weather systems). For the area of interest in this study we specifically mention the Icelandic low pressure system…"

P3L8: linear in what? Do you mean that the processes cannot be represented by gradients that are a linear function of time?

PL21: Add "While total gradients are estimated, they can be interpreted as the sum of hydrostatic and wet components as well. In the following we will subtract the hydrostatic component computed from ECMWF from the total GPS gradient to get the GPS wet gradient."

P12: Are the ECMWF gradients computed from operational analyses or a reanalysis? Note that operational analyses should not be used to analyse long time series because of changes in the model and assimilation system over time.

P13L8-9 (line numbering in the manuscript doesn't match) why is the formal error for the north component larger?

Figure 9: add some comments on the year-to-year variability in the results and differences between GPS and ECMWF.

P17L11: The sentence suggests that ECMWF gradients are more accurate if used to validate GPS gradients. I suggest to reformulate as "The correlations seen in all cases confirm that a consistent atmospheric signal in terms of gradients is detected by the GPS observations and ECMWF analyses."

P17L14: rather than being better modelled by the ECMWF analyses, I think that larger scale features agree better because the representativeness differences between the gridded model fields and GPS point observations are smaller.

P17L25: "…but the relative differences between the sites" it is not clear what relative differences are meant, suggest to be more specific: "GPS gradients are larger by a factor of ~1.5 and this factor is roughly the same for all sites."

P17 last sentence: "e.g. instrumental" can you be more specific?

P19: I think the section on trend results is not relevant. First, it poses the problem of the homogeneity of the data (both GPS and ECMWF). Second, the values reported in Table 8 are confirmed in the text to be statistically insignificant. Third, the main interest of inspecting trends is said to be for the detection of hardware problems. This is well illustrated in Dousa et al., 2017, but not here. I suggest to remove this section and Table 8 and replace it with one or two summarizing sentences in the Conclusion section.

Table 11: over which period of time are the mean values computed?

P23: Section 5.2: discussion of the factors that can cause a difference in GPS and WVR gradient amplitudes should be revised (see above).

I don't understand point 2) Constraints on the variability should not impact the mean amplitude.

At bottom of page "Before studying the correlation…" remove this sentence as GPS and WVR were already compared in the previous section.

P25L13: As expected => As seen previously from total gradients…

Figure 12: use the same range for x and y – axis in all plots.

P30: remove the last sentence (it is not demonstrated neither in this paper nor in general that the GPS gradients can help to validate high resolution NWM models). Also P32 remove "both for evaluation of the performance of the model and".

P32. First sentence: the study doesn't really explain the GPS gradients based on meteorological phenomena, but rather presents a statistical analysis and comparison with other data sources. Please correct the sentence.

"horizontal gradients" => GPS gradients

Revise the interpretation of impact of cutoff on GPS gradients

"simultaneously estimated" => remove "simultaneously"

---

## Author Response (AR3)

**Response to editor's comments on:**

*On the information content in linear horizontal gradients estimated from space geodesy observations*

*by Gunnar Elgered, Tong Ning, Peter Forkman, and Rüdiger Haas*

*6 May 2019*

**Introduction**
The comments are appreciated and well taken. Our responses for each comment follow below, where the editor's comments are in black text and our responses are in red text. Other updates have also been made, mainly language oriented. For example, the first three sections giving background information are now mainly written in the past tense, whereas the results, conclusions, and suggestions for future work are mainly in present and future tense.

I would like to congratulate the authors for the implementation of the corrections and suggestions provided in my previous report and their careful review and revision of the text and figures. I have two concerns related to some new results included in the revised manuscript and a few additional minor comments.

Table 11 shows the mean values of gradient amplitudes from two GPS stations for 3 different cutoff angles (3°, 10°, and 20°). The result is quite striking. The mean amplitude is clearly increasing when the cutoff is increasing, or the other way round, the mean amplitude is decreasing when the cutoff is decreasing. You adopt the 2nd point of view and speculate that the decrease is due to averaging of a larger air-mass, a result that is compared to averaging over longer time periods. I am not convinced by this explanation. I think that changing the cutoff angle has primarily an impact on the correlation between gradients and other parameters and on the accuracy of the estimated parameters. Actually, decreasing the cutoff is expected to improve the accuracy and provide more realistic gradients estimates. I think this is supported by the enhanced agreement with WVR results (the 3° GPS solution agrees better with WVR results). On the other hand, increasing the cutoff to 20° seems rather an unfavourable situation for estimating accurate gradients and it may be that the GPS gradient estimates are actually biased in this case, as well as the WVR gradient maybe? The increased standard deviation of differences and decreased correlation coefficients between GPS and WVR indeed support the idea that GPS and WVR agree less well in that case, i.e. the uncertainty in the GPS gradients is larger as also predicted by the larger formal error. I think that this effect is larger than the effect of sensing different air-masses at lower elevations. At least both points of view should be discussed and if you want to maintain your idea, more evidence should be provided to support it.

We agree, increasing the formal error, and the uncertainty, will imply a larger scatter which in turn will increase the mean amplitude. We think it is quite clear that both these effects will contribute to the observed increase of the mean amplitude with higher elevation cutoff angle in former Table 11 (now Table 10). More work is required in order to determine their relative importance, e.g. a high resolution (100 m ?) numerical weather model to simulate the averaging effect in a turbulent atmosphere, which we do not have access to. Therefore, we stop here and describe both effects and introduce one more suggestion for future work.

My second concern is with your statement that there can be a trend in amplitude without a trend in components (also noted in the previous review). Since the gradient is a vector, if its length increases, the components necessarily increase. This statement should be removed or clarified if I didn't get what you mean.

We think this is matter of definitions. The gradient components (east and north) can be both negative and positive. We include the sign when we calculate the trends for each component, which means that a trend in any of the two components is a kind of tilt of the atmosphere. The variability may increase without any tilt involved, causing the amplitude to have a positive trend, but not the components. This is illustrated by a simulation in the following two graphs:

[Figure]

Let us assume that the left and right graphs depict all estimated/observed gradients during year 1 and year 2, respectively. The mean values are both < 1 μm, whereas the mean amplitudes are 0.08 mm and 0.10 mm, respectively. We have a trend in the amplitude but not in the east component. (For simplicity this simulation sets the north gradient to always be zero.) Based on this discussion we have rewritten the corresponding text since we think it is important that we define what we have estimated, in spite of the fact that we could only place an upper bound on any gradient trend (discussed further below).

Below are a few additional minor suggestions.
Though you slightly extended the last paragraph describing the organisation of the paper, I think there is still a paragraph missing in the Introduction on the motivation of this work and the rationale of the study scenario. Please add a short paragraph between line 10 and 11.

Done

P2L26-28: Meindl et al. (2004) discussed the global north-south temperature gradient. In your work, regional high and low pressure systems are more important. I suggest to remove the sentence related to work of Meindl et al. (2004), and rewrite this part as follows: "Hydrostatic gradients are usually dominated by pressure gradients and exist mainly over regional scales (e.g. persistent high and low pressure systems) and synoptic scale (e.g. weather systems). For the area of interest in this study we specifically mention the Icelandic low pressure system…"

The text is rewritten. Although we believe that hydrostatic gradients are dominated by gradients in pressure there is no need to make such a general statement. Meindl et al. (2004) used a European and a global network and did not really go into detail about the cause for the stronger north gradients by trying to quantify the influence from temperature and pressure gradients. They wrote: "It is also interesting to note that the north components seem to have a maximum at mid-latitudes and decrease towards the poles and equator."

(Note that the pressure is in the nominator and the temperature is in the denominator in the expression for the refractivity.) Because the work by Meindl et al. (2004) included three of the five stations in our study we think it is appropriate to make a reference to their (early) work.

P3L8: linear in what? Do you mean that the processes cannot be represented by gradients that are a linear function of time?
No, we meant horizontally linear, thinking of the processes mentioned in the previous paragraph. We inserted "horizontally".

PL21: Add "While total gradients are estimated, they can be interpreted as the sum of hydrostatic and wet components as well. In the following we will subtract the hydrostatic component computed from ECMWF from the total GPS gradient to get the GPS wet gradient."
Done (assuming that it was referring to page 3 — it made sense).

P12: Are the ECMWF gradients computed from operational analyses or a reanalysis? Note that operational analyses should not be used to analyse long time series because of changes in the model and assimilation system over time.
It is not stated in the paper by Boehm and Schuh (2007). They just write "data of the European Centre for Medium-range Weather Forecasts (ECMWF) model". We suspect that the gradients are calculated from the operational analysis but since we do not detect any gradient trends from neither the ECMWF gradient time series, nor the GPS wet gradient time series it is not critical.

P13L8-9 (line numbering in the manuscript doesn't match) why is the formal error for the north component larger?
We added the following sentence: "The reason is that we lose many observations of satellites located in the north, see Figure 3."

Figure 9: add some comments on the year-to-year variability in the results and differences between GPS and ECMWF.
As an introduction to the details below we added the following sentences at the end of the 1st paragraph: "In the top graphs, comparing ECMWF and GPS gradients, we note that the GPS gradients show a larger variability. There are also differences between the east and the north gradients both in the mean over the year and in the seasonal variations."

P17L11: The sentence suggests that ECMWF gradients are more accurate if used to validate GPS gradients. I suggest to reformulate as "The correlations seen in all cases confirm that a consistent atmospheric signal in terms of gradients is detected by the GPS observations and ECMWF analyses."
Changed

P17L14: rather than being better modelled by the ECMWF analyses, I think that larger scale features agree better because the representativeness differences between the gridded model fields and GPS point observations are smaller.
We deleted "which is better modelled by the ECMWF data." The next sentence is sufficient for us to explain what we think is the cause.

P17L25: "…but the relative differences between the sites" it is not clear what relative differences are meant, suggest to be more specific: "GPS gradients are larger by a factor of ~1.5 and this factor is roughly the same for all sites."
Changed

P17 last sentence: "e.g. instrumental" can you be more specific?
The end of the paragraph is rewritten: "… e.g. multipath effects, become important. Variations in the electromagnetic environment that change the impact of the signal multipath at a station may be due to e.g. snow, rain, vegetation, and soil moisture."

P19: I think the section on trend results is not relevant. First, it poses the problem of the homogeneity of the data (both GPS and ECMWF). Second, the values reported in Table 8 are confirmed in the text to be statistically insignificant. Third, the main interest of inspecting trends is said to be for the detection of hardware problems. This is well illustrated in Dousa et al., 2017, but not here. I suggest to remove this section and Table 8 and replace it with one or two summarizing sentences in the Conclusion section.
We did not include Table 8 in the original submission because of the sizes of the uncertainties. The (old) Table 8 is now removed but we need to describe what we did in order to mention it later in the conclusions. (This part of the conclusions is not changed.) Let us compare to some work of our fellow astronomers, i.e. to publish the result from a non-detection. After a search for a specific molecule in a gaseous nebula without a detection, they can place an upper bound on the abundance. In our case we can set an upper bound on any gradient trend for an eleven-year long time series. In this context the reference to Dousa et al. (2017) is relevant, because even if the probability of finding a true gradient trend (larger than its uncertainty) is very small, the exercise itself is motivated. The title of the subsection was modified to "Search for long term trends" which already from the beginning indicate that they are difficult to find.

Table 11: over which period of time are the mean values computed?
Here we note a mistake in the corresponding text. Our first version of the table had separate columns for each year. However, the values were almost identical so we made the table simpler and gave one value for the whole 4-year period. This is now corrected in the text:
" … for each year." → " … for the 4-year period 2013–2016."
We also added "15-minute" in the title of the table as an additional clarification.

P23: Section 5.2: discussion of the factors that can cause a difference in GPS and WVR gradient amplitudes should be revised (see above).
It has been revised as described above.

I don't understand point 2) Constraints on the variability should not impact the mean amplitude.
We added to the following text: "A constraint has a similar impact as a low-pass filter. Peaks with a short duration, requiring rapid changes, will be reduced. This is a valid argument for the mean gradient amplitude, but not for the individual east and north components, because their mean values are expected to be very close to zero regardless of the value of the constraint."

At bottom of page "Before studying the correlation…" remove this sentence as GPS and WVR were already compared in the previous section.

Done + we now specifically mention ONSA and ONS1 in the next sentence.

P25L13: As expected => As seen previously from total gradients…

Changed

Figure 12: use the same range for x and y – axis in all plots.

Done

P30: remove the last sentence (it is not demonstrated neither in this paper nor in general that the GPS gradients can help to validate high resolution NWM models). Also P32 remove "both for evaluation of the performance of the model and".

The last sentence on P30 was removed.
Regarding the sentence on P32 it was intended to be interpreted as suggested future work. Regarding the specific wording "both for evaluation ...". It is difficult to do both at the same time, we agree. On the other hand model problems could also be identified, and especially if independent WVR gradients agree with the GPS gradients, but that should be obvious, so we adopt the suggested shorter version. In order to be clear that the conclusion section is not only including what we have shown but also what we propose to investigate if it is possibly to show, the title of the section now also states suggestions for future work.

P32. First sentence: the study doesn't really explain the GPS gradients based on meteorological phenomena, but rather presents a statistical analysis and comparison with other data sources. Please correct the sentence.

As a first introductory sentence to the conclusions it works well with: "We have estimated linear horizontal gradients from GPS data from five sites in Sweden."

"horizontal gradients" => GPS gradients

Changed

Revise the interpretation of impact of cutoff on GPS gradients

Done (as described above)

"simultaneously estimated" => remove "simultaneously"

Done

[revised manuscript text omitted]

---

## Editor Decision (ED3)

Editor comments on manuscript amt-2018-318-version 5 "On the information content in linear horizontal delay gradients estimated from space geodesy observations" by Gunnar Elgered and co-authors.

Thank you for providing a revised manuscript and answers to my comments.

The two major points I mentioned in the previous review have been clarified but the text still needs be clarified to make these two points fully understandable. A few other points need also to be further discussed and explained. Below are my comments and suggestions for revision.

Please note also that the reference numbering in the revised manuscript did not work properly (all references in the text appear as ??). In order to avoid publication delays, be careful that this is corrected in the next version.

Section 5.2 on the discussion of the difference of amplitude of gradients from GPS and WVR

P24L290-294: suggested revision:

"(2) The WVR gradients for one 15-minute period do not depend on earlier or later estimates whereas the GPS gradients are estimated using constraints on the variability. A constraint has a similar impact as a low-pass filter (peaks with a short duration will be reduced). "

=> the variability of the GPS estimates is actually controlled by the random walk parameter. A value of 0.3 mm/sqrt(h) is given in Table 1. How was this value chosen? Based on the results of this study would you suggest to increase this value? Did you test this? According to the paper by Gradinarski et al., 2000, increasing the RW parameters doesn't improve the RMSD. A short discussion of this point might be relevant at this stage.

P24L295-302: suggested revision:

"(3) The fact that the WVR and the GPS gradients are computed for different elevation cutoff angles has two possible impacts: (i) the larger volume sensed by GPS (with a 3 degrees cutoff angle) includes different air masses and introduces an averaging effect that reduces the variability and mean amplitude of gradients, similar to averaging over longer time periods as shown in Table 6; (ii) the higher cutoff angle of the WVR observations (20 degrees) results in larger formal errors, and thus larger variability and larger gradient amplitudes. Table 10 shows the impact of changing the elevation cutoff angle from the GPS observations for ONSA and ONS1 over the 4-year period 2013–2016."

P24L303-304: suggested revision:

"We conclude that the use of different constraints and cutoff angles are the likely explanations for the differences in gradient amplitudes estimated from GPS and WVR data but cannot based on these results determine their relative importance."

=> Add the standard deviation of gradient amplitude in Table 10 to support the idea that variability is increasing with cutoff angle. Add also the WVR results (mean and standard deviation of gradient amplitude) to be compared to the GPS results at 20°.

Figure 10: add the information that the GPS results are for 3° and no elevation dependent weighting. Add which time resolution is used to compute the monthly mean and SD.

Section 4.3 on the trends

Thank you for proving a graphical explanation of your reasoning in the answer. Now I understand what you were meaning. So yes, I agree that the mean gradient components can have no trends while their mean absolute value can have a trend. However, this situation is purely theoretical and not based on observation (at least not documented in the manuscript). It is also a bit misleading because it appears as a contradiction where there is none. Indeed, since the gradient is a vector, it is important to check mainly its amplitude and direction rather than its components because amplitude and direction have more physical sense. But actually, I think that this discussion is unnecessary in the manuscript.

Moreover, I have a concern with the two following sentences:

P20L225: "A positive trend corresponds to a larger variability" => this is incorrect if you consider a trend in the variability of the direction of the gradient vector with constant amplitude. And a positive trend can also be simply due to an increase in amplitude without a change in direction, i.e. not only due to larger variability.

P20L227: "A trend of larger east gradients can be balanced by a similar trend in larger west gradients". This sentence sounds awkward. A directional trend would be either eastward or westward but not both… I think what you mean is that more positive values (eastward gradient) can be balanced by more negative values (westward gradient) resulting in a null mean value, but this situation is rather speculative and unnecessary.

=> As a consequence, I recommend that you remove the sentences between L223 and L228 "An estimated gradient has a direction… but a trend in the gradient amplitude."

The rest of the Section 4.3 should also be revised because both the GPS and ECMWF trends have large uncertainties. The GPS series are not homogenized and the ECMWF product used in this study is probably elaborated from operational ECMWF data (i.e. not homogeneous) and has some inherent limitations discussed by Kacmarik et al., 2018 (see the comparison of the gradient product by Böhm and Schuh, 2007, and other ray-traced products in their final publication). Landskron, D. & Böhm (2018) recommend also to use their new refined discrete gradient product computed from ERA-Interim reanalysis instead of the earlier product by Böhm and Schuh (2007).

Landskron, D. & Böhm, J. J Geod (2018) 92: 1387. https://doi.org/10.1007/s00190-018-1127-1

Kačmařík, M., Douša, J., Zus, F., Václavovic, P., Balidakis, K., Dick, G., and Wickert, J.: Sensitivity of GNSS tropospheric gradients to processing options, Ann. Geophys. Discuss., https://doi.org/10.5194/angeo-2018-93, in review, 2018.

P20L235-236: "Nevertheless, to study long time series of estimated gradients is motivated by the monitoring of the quality of the data from a GPS station." This statement should be made earlier (e.g. at beginning of the sub-section or in the Introduction). If made in this sub-section, it should be better explained in the context of trend analysis (e.g. hardware malfunctioning can produce drifts or breaks in gradient series and thus artificial trends).

As suggested in the previous review, this sub-section could be totally removed, or at least limited to the analysis of trends in total GPS gradients (not using this particular ECMWF gradient product for which the uncertainty may not be adapted for trend estimation).

Section 4.2 on the GPS and ECMWF results

The differences between the GPS and ECMWF gradients in Table 6 and 7 should be further discussed, not just the differences between stations. There are large differences in the mean gradients and a systematic underestimation of the variability in the ECMWF data by a factor of 2.

The quality of the ECWMF gradient product by Böhm and Schuh (2007) should be better stated (see below) or assessed/discussed in this section. This could be added as a specific objective of this study.

Section 3.4 on ECMWF data

There are different methods used to compute horizontal gradients from ECMWF data and several products exist. The product used here is usually referred to as LHG (linear horizontal gradients) by Böhm and Schuh (2007). Note that it has been replaced by a refined discrete gradient product using ERA-Interim reanalysis (Landskron and Böhm, 2018). Other ray-tracing methods used to compute gradients from NWM data were developed by Zus et al., 2012, and Zus et al., 2015, and are discussed in Kačmařík et al., 2018 (ANGEO, in review).

Zus, F., Bender, M., Deng, Z., Dick, G., Heise, S., Shang-Guan, M. and Wickert, J.: A methodology to compute GPS slant 15 total delays in a numerical weather model, Radio Science, 47, RS2018, doi:10.1029/2011RS004853, 2012.

Zus, F., Dick, G., Heise, S. and Wickert, J.: A forward operator and its adjoint for GPS slant total delays, Radio Science, 50, 393–405, doi: 10.1002/2014RS005584, 2015.

Minor comments on the revised manuscript (version 5):

P2L35-38: the question about the seasonal changes of gradients was previously studied by Koulali et al., 2012, please cite here.

"if these make sense given present knowledge about the meteorological conditions" change to "if they can be explained by the influence of regional-scale weather systems".

The scientific question leading to the "comparison of GPS gradients with high temporal resolution" is not clearly formulated. Why or for which applications is a 15-minute resolution more interesting/useful than a 6-hour resolution?

Add a scientific question regarding the assessment of the ECMWF LHG data by comparison with GPS?

P18L213: the "SD" of the GPS gradients are larger … (add SD and maybe define the acronym here if used first time).

P18L215 and 216: "significantly smaller": how is the significance measured? "comparable": what is the limit used to distinguish between smaller and comparable? Based on the values in Table 6, the ratio of daily values for KIR0 and SPT0 is 0.32/0.38 = 0.84, so KIR0 is 16% smaller, but the ratio of monthly values is 0.13/0.16 = 0.81, so KIR0 is 19% smaller in that case, so not comparable… Please revise the statements and refer to the numbers in the Table(s).

P18L217-218: "at this level the hydrostatic gradient and other effects, e.g. signal multipath effects, become important." The SD of monthly hydrostatic and wet gradients over 4 years are given in Fig. 8. They should be quoted here and used to estimate the contribution of both components to the total gradient (if they are valid for the 11-y period, if not compute the SD for 11-year here).

Why would "signal multipath effects" by larger in the monthly SD than in the 6h and daily data?

P21L252-255: Reformulate as: "The GPS wet gradients for ONSA and ONS1 are computed by subtracting the hydrostatic gradients from ECMWF, linearly interpolated to match the time epochs of the GPS gradients, from the total GPS gradients."

P21L262 "This confirms the results presented by Kacmarík et al. (2018) using a GNSS station network in central Europe." Be more specific or don't cite Kacmarík et al. (2018) here because they did not use a WVR.

P22L277: "the correlation is here reduced to around 0.6" rather 0.66 at 3° (average of 0.64 and 0.68).

P33L387-388: "gradients calculated from meteorological analyses of the ECMWF" refer to the LHG product used in this study as the results may depend on the NWM model, the computation method and mapping function used in the method as discussed by Kačmařík et al., 2018.

P33L389-391: "No significant long-term trends were detected for the GPS gradients. If small gradient trends are detected in the future, we recommend to critically assess if they could be caused by station problems or confirmed by a nearby (or even collocated) station." I think these sentences should be made less general because the study was conducted in a limited area. It cannot be said that there are no detectable trends in other parts of the world, though it is true in general that when a trend is detected the time series be visualised and data inspected to check the nature of the trend…

P33L392-393: quantify "most of the variability"

P33L393: "implies a better agreement" => based on the correlation coefficients mainly (the SD of differences may actually decrease because the GPS gradients at 3° are of smaller amplitude).

P33L394: Kačmařík et al., 2018, did not use a WVR, and they only compared 3° and 7° solutions. A thorough optimization wrt cutoff angle might require testing more values.

P33L395: "Related to this is…" suggests that it explains the above results though it is the opposite! Change to: "despite…"

P33L396…

"We interpret this result as the combined impact of two possible causes" => "We interpret this difference as the result of two combined effects"

"(1) the decrease of mean amplitude and variability at the lower cutoff angle results from the averaging of a larger air mass (similar to averaging over longer periods)"

"(2) the increase of mean amplitude and variability at the higher cutoff angle results from the increase of uncertainty and thus larger scatter in the estimates".

---

## Author Response (AR4)

**Response to editor's comments (23 May 2019) on:**

*On the information content in linear horizontal delay gradients estimated from space geodesy observations*

*by Gunnar Elgered, Tong Ning, Peter Forkman, and Rüdiger Haas*

*4 June 2019*

The comments are appreciated and well taken. Our responses for each comment follow below, where the editor's comments are in black text and our responses are in red text. We added some clarifications and corrected language errors. These are seen in the attached output from DiffLaTeX.

Thank you for providing a revised manuscript and answers to my comments.
The two major points I mentioned in the previous review have been clarified but the text still needs be clarified to make these two points fully understandable. A few other points need also to be further discussed and explained. Below are my comments and suggestions for revision.

Please note also that the reference numbering in the revised manuscript did not work properly (all references in the text appear as ??). In order to avoid publication delays, be careful that this is corrected in the next version.
Yes, we apologize.

Section 5.2 on the discussion of the difference of amplitude of gradients from GPS and WVR
P24L290-294: suggested revision:
"(2) The WVR gradients for one 15-minute period do not depend on earlier or later estimates whereas the GPS gradients are estimated using constraints on the variability. A constraint has a similar impact as a low-pass filter (peaks with a short duration, requiring rapid changes, will be reduced). This is a valid argument for the mean gradient amplitude, but not for the mean values of the individual east and north components, because they are expected to be very close to zero regardless of the value of the constraint."
Done (parenthesis added)

=> the variability of the GPS estimates is actually controlled by the random walk parameter. A value of 0.3 mm/sqrt(h) is given in Table 1. How was this value chosen? Based on the results of this study would you suggest to increase this value? Did you test this? According to the paper by Gradinarski et al., 2000, increasing the RW parameters doesn't improve the RMSD. A short discussion of this point might be relevant at this stage.
The following text was added in Section 3.1 (commenting on this specific value in Table 1):
*"In order to investigate the impact of different constraints on the estimated gradients in addition to the value of 0.3 mm/sqrt(h) suggested by Bar-Sever et al. (1998), we also processed two days of GPS data for ONSA using the constraint values: 0.6, 0.9, and 1.2 mm/sqrt(h). The resulting gradients were compared to those estimated from the WVR data. The result shows no significant difference in the gradient amplitudes or the correlations. This is consistent with the result presented by Gradinarsky et al. (2000)."*

P24L295-302: suggested revision:
"(3) The fact that the WVR and the GPS gradients are computed for different elevation cutoff angles has two possible impacts: (i) the larger volume sensed by GPS (with a 3 degrees cutoff angle) includes different air masses and introduces an averaging effect that reduces the variability and mean amplitude of gradients, similar to averaging over longer time periods as shown in Table 6; (ii) the higher cutoff angle of the WVR observations (20 degrees) results in larger formal errors, and thus larger variability

and larger gradient amplitudes. Table 10 shows the impact of changing the elevation cutoff angle from the GPS observations for ONSA and ONS1 over the 4-year period 2013–2016."
Done.

P24L303-304: suggested revision:
"We conclude that the use of different constraints and cutoff angles are the likely explanations for the differences in gradient amplitudes estimated from GPS and WVR data but cannot based on these results determine their relative importance."
=> Add the standard deviation of gradient amplitude in Table 10 to support the idea that variability is increasing with cutoff angle. Add also the WVR results (mean and standard deviation of gradient amplitude) to be compared to the GPS results at 20°.
Our main idea with this table was to present that the size of the estimated gradients increased with elevation cutoff angle, but we agree that it is relevant to also present the increase in the SD. Two columns with SD (one for ONSA and one for ONS1) are added in Table 10. Since the corresponding WVR results are only available for a 20° elevation cutoff angle, these are reported in a footnote.

Figure 10: add the information that the GPS results are for 3° and no elevation dependent weighting. Add which time resolution is used to compute the monthly mean and SD.
Done (in the caption of the figure).

Thank you for proving a graphical explanation of your reasoning in the answer. Now I understand what you were meaning. So yes, I agree that the mean gradient components can have no trends while their mean absolute value can have a trend. However, this situation is purely theoretical and not based on observation (at least not documented in the manuscript). It is also a bit misleading because it appears as a contradiction where there is none. Indeed, since the gradient is a vector, it is important to check mainly its amplitude and direction rather than its components because amplitude and direction have more physical sense. But actually, I think that this discussion is unnecessary in the manuscript. Moreover, I have a concern with the two following sentences:
P20L225: "A positive trend corresponds to a larger variability" => this is incorrect if you consider a trend in the variability of the direction of the gradient vector with constant amplitude. And a positive trend can also be simply due to an increase in amplitude without a change in direction, i.e. not only due to larger variability.
P20L227: "A trend of larger east gradients can be balanced by a similar trend in larger west gradients". This sentence sounds awkward. A directional trend would be either eastward or westward but not both… I think what you mean is that more positive values (eastward gradient) can be balanced by more negative values (westward gradient) resulting in a null mean value, but this situation is rather speculative and unnecessary.
=> As a consequence, I recommend that you remove the sentences between L223 and L228 "An estimated gradient has a direction… but a trend in the gradient amplitude."
We discuss Section 4.3 further below.

The rest of the Section 4.3 should also be revised because both the GPS and ECMWF trends have large uncertainties. The GPS series are not homogenized and the ECMWF product used in this study is probably elaborated from operational ECMWF data (i.e. not homogeneous) and has some inherent limitations discussed by Kacmarik et al., 2018 (see the comparison of the gradient product by Böhm and Schuh, 2007, and other ray-traced products in their final publication). Landskron, D. & Böhm (2018) recommend also to use their new refined discrete gradient product computed from ERA-Interim reanalysis instead of the earlier product by Böhm and Schuh (2007).
Landskron, D. & Böhm, J. J Geod (2018) 92: 1387. https://doi.org/10.1007/s00190-018-1127-1
Kačmařík, M., Douša, J., Zus, F., Václavovic, P., Balidakis, K., Dick, G., and Wickert, J.: Sensitivity of GNSS tropospheric gradients to processing options, Ann. Geophys. Discuss., https://doi.org/10.5194/angeo-2018-93, in review, 2018.
P20L235-236: "Nevertheless, to study long time series of estimated gradients is motivated by the monitoring of the quality of the data from a GPS station." This statement should be made earlier (e.g.

at beginning of the sub-section or in the Introduction). If made in this sub-section, it should be better explained in the context of trend analysis (e.g. hardware malfunctioning can produce drifts or breaks in gradient series and thus artificial trends).

As suggested in the previous review, this sub-section could be totally removed, or at least limited to the analysis of trends in total GPS gradients (not using this particular ECMWF gradient product for which the uncertainty may not be adapted for trend estimation).

*It is true that this sub-section can be removed. It is not the focus of the study. However, it is of some interest that we do not find any trends, neither in the GPS data (that are not homogenised, although we guess that adding offsets to the ZWD will probably have a very small impact on estimated gradients) nor in the ECMWF data (which are not the "state-of the-art"). Finally, we decided to remove the sub-section.*

The differences between the GPS and ECMWF gradients in Table 6 and 7 should be further discussed, not just the differences between stations. There are large differences in the mean gradients and a systematic underestimation of the variability in the ECMWF data by a factor of 2.

*The possible shortcomings with the ECMWF gradients have now been mentioned in Section 3.4 (see below). Based on this the following discussion is given in Section 4.2:*

*"When comparing the two tables it is clear that there are differences in the mean values of up 0.2 mm. These differences are mainly in the east component whereas there are consistent negative values for the north component. The SD of the GPS gradient are larger than the ECMWF gradients themselves. We note that the empirical factors used when calculating the ECMWF gradients (see Section 3.4) may not be correct for these stations. A related possible reason is that not all variations in the water vapour content are detected due to the poor spatial and temporal resolutions of the ECMWF data."*

The quality of the ECWMF gradient product by Böhm and Schuh (2007) should be better stated (see below) or assessed/discussed in this section. This could be added as a specific objective of this study.

*(see next comment)*

Section 3.4 on ECMWF data

There are different methods used to compute horizontal gradients from ECMWF data and several products exist. The product used here is usually referred to as LHG (linear horizontal gradients) by Böhm and Schuh (2007). Note that it has been replaced by a refined discrete gradient product using ERA-Interim reanalysis (Landskron and Böhm, 2018). Other ray-tracing methods used to compute gradients from NWM data were developed by Zus et al., 2012, and Zus et al., 2015, and are discussed in Kačmařík et al., 2018 (ANGEO, in review).

Zus, F., Bender, M., Deng, Z., Dick, G., Heise, S., Shang-Guan, M. and Wickert, J.: A methodology to compute GPS slant 15 total delays in a numerical weather model, Radio Science, 47, RS2018, doi:10.1029/2011RS004853, 2012.

Zus, F., Dick, G., Heise, S. and Wickert, J.: A forward operator and its adjoint for GPS slant total delays, Radio Science, 50, 393–405, doi: 10.1002/2014RS005584, 2015.

*The following paragraph is added in Section 3.4:*

*"There are alternative methods to derive gradients from ECMWF data using ray tracing methods, see e.g. (Zus et al. 2015) and references therein. More recently gradient calculations based on the ERA-Interim analyses were recommended by Landskron and Böhm (2018). In spite of the possible shortcomings of the ECMWF gradients by Boehm and Schuh (2007) they are an independent source of information which we use for comparisons with estimated GPS gradients in terms of seasonal variations and differences between sites for different time scales."*

Minor comments on the revised manuscript (version 5):

P2L35-38: the question about the seasonal changes of gradients was previously studied by Koulali et al., 2012, please cite here.

"if these make sense given present knowledge about the meteorological conditions" change to "if they can be explained by the influence of regional-scale weather systems".

*Rephrased, and the citation is added.*

The scientific question leading to the "comparison of GPS gradients with high temporal resolution" is not clearly formulated. Why or for which applications is a 15-minute resolution more interesting/useful than a 6-hour resolution?

The text appearing before the paragraph presenting the structure of the paper now reads:

*"Here we report on comparisons between GPS and WVR gradients, with a temporal resolution of 15 minutes, over a more or less continuous 4-year period. With such a resolution it is for example possible to study convective systems (Brenot et al. 2013) and the relation between the temporal variability of the gradients and the zenith wet delay (ZWD) during the passage of weather fronts."*

Add a scientific question regarding the assessment of the ECMWF LHG data by comparison with GPS?

Given that we only use one product based on ECMWF data (which is not the most modern one) we refrain from making this a scientific question that is relevant for this study.

P18L213: the "SD" of the GPS gradients are larger … (add SD and maybe define the acronym here if used first time).

"SD" added (SD is defined in Section 3.4)

P18L215 and 216: "significantly smaller": how is the significance measured? "comparable": what is the limit used to distinguish between smaller and comparable? Based on the values in Table 6, the ratio of daily values for KIR0 and SPT0 is 0.32/0.38 = 0.84, so KIR0 is 16% smaller, but the ratio of monthly values is 0.13/0.16 = 0.81, so KIR0 is 19% smaller in that case, so not comparable… Please revise the statements and refer to the numbers in the Table(s).

It is hard to remember exactly why the present wording was chosen, possibly it was more related to absolute differences in mm rather than in percentages. We have rewritten the text avoiding the subjective words "comparable" and "significant". It now reads:

*"We note that the SD obtained for the KIR0 station for 6 h and one day are smaller. This is likely a consequence of the lower humidity at the station. For monthly averages, these differences are reduced and the SD for all stations are in the range 0.13–0.18 mm indicating that the hydrostatic gradients and other effects, e.g. signal multipath, become relatively more important."*

P18L217-218: "at this level the hydrostatic gradient and other effects, e.g. signal multipath effects, become important." The SD of monthly hydrostatic and wet gradients over 4 years are given in Fig. 8. They should be quoted here and used to estimate the contribution of both components to the total gradient (if they are valid for the 11-y period, if not compute the SD for 11-year here).

The relative importance of monthly SD for the wet and the hydrostatic gradients were calculated for the five sites using the ECMWF data covering the 11 years. The results for ONSA are consistent with the values for the 4 years presented in Figure 8. The following text is added (after the rewritten text in the previous point):

*"The relative importance of hydrostatic and wet gradients was illustrated in Figure 8 using four years of data from the ONSA station. Using all eleven years of ECMWF data, all sites have standard deviations of the hydrostatic east and north monthly gradients in the range from 0.05 mm to 0.07 mm, whereas the standard deviations for the monthly wet gradients show a dependence with latitude, from 0.03 mm at KIR0 in the north to 0.06 mm at ONSA in the south."*

Why would "signal multipath effects" by larger in the monthly SD than in the 6h and daily data?

We missed the word relatively which is now added (see the new text above).

P21L252-255: Reformulate as: "The GPS wet gradients for ONSA and ONS1 are computed by subtracting the hydrostatic gradients from ECMWF, linearly interpolated to match the time epochs of the GPS gradients, from the total GPS gradients."

Done

P21L262 "This confirms the results presented by Kacmarík et al. (2018) using a GNSS station network in central Europe." Be more specific or don't cite Kacmarík et al. (2018) here because they did not use a WVR.

We removed this sentence (and the citation) because a similar statement is modified in the conclusions, just mentioning the importance of low elevation observations (not a WVR).

P22L277: "the correlation is here reduced to around 0.6" rather 0.66 at 3° (average of 0.64 and 0.68).

We now give that actual values:

*"... the correlation coefficients are here reduced, to 0.68 and 0.64 for the east and the north component, respectively."*

P33L387-388: "gradients calculated from meteorological analyses of the ECMWF" refer to the LHG product used in this study as the results may depend on the NWM model, the computation method and mapping function used in the method as discussed by Kačmařík et al., 2018.

A citation to Boehm and Schuh (2007) is added.

P33L389-391: "No significant long-term trends were detected for the GPS gradients. If small gradient trends are detected in the future, we recommend to critically assess if they could be caused by station problems or confirmed by a nearby (or even collocated) station." I think these sentences should be made less general because the study was conducted in a limited area. It cannot be said that there are no detectable trends in other parts of the world, though it is true in general that when a trend is detected the time series be visualised and data inspected to check the nature of the trend…

We agree that our findings are not at all general (but the recommendation to be critical is valid globally). This conclusion has, however, been removed since Section 4.3 was removed.

P33L392-393: quantify "most of the variability"

Because we have no value for the variability of the hydrostatic gradients for 15-minute periods we reformulate this sentence to:

"We studied wet gradients estimated with a temporal resolution of 15 min from GPS and WVR data."

P33L393: "implies a better agreement" => based on the correlation coefficients mainly (the SD of differences may actually decrease because the GPS gradients at 3° are of smaller amplitude).

Yes, it can be concluded from the correlation coefficients, but the gradient amplitudes in the reference data set from the WVR are still based on the 20° elevation cutoff angle, and the WVR values are slightly higher, meaning that also a reduction of the SD indicates a better agreement. We keep the original text.

P33L394: Kačmařík et al., 2018, did not use a WVR, and they only compared 3° and 7° solutions. A thorough optimization wrt cutoff angle might require testing more values.

The text is rewritten, see next comment ...

P33L395: "Related to this is…" suggests that it explains the above results though it is the opposite! Change to: "despite…"

It was not our intention that it shall be interpreted as an explanation, rather it is an additional observation. The text relevant to these last four comments now read:

[revised manuscript text omitted]

---

## Editor Decision (ED4)

Editor comments on manuscript amt-2018-318-version 6 "On the information content in linear horizontal delay gradients estimated from space geodesy observations" by Gunnar Elgered and co-authors.

Thank you for providing a revised manuscript and answers to my comments. The main points have been well clarified. You also followed my earlier recommendation to remove section 4.3. I think this helps focusing the study on its main goals for which robust answers are now brought. Below are a few final remarks and requirements. Once these corrections are implemented the paper will be ready to go for typesetting and publication.

Comment on the text that you added in Section 3.1: "In order to investigate the impact of different constraints on the estimated gradients in addition to the value of 0.3 mm/sqrt(h) suggested by Bar-Sever et al. (1998), we also processed two days of GPS data for ONSA using the constraint values: 0.6, 0.9, and 1.2 mm/sqrt(h). The resulting gradients were compared to those estimated from the WVR data. The result shows no significant difference in the gradient amplitudes or the correlations. This is consistent with the result presented by Gradinarsky et al. (2000)."

=> How were these two days chosen? What was the meteorological context? In calm weather no significant impact is indeed expected, but in strong convective situations and/or frontal passages a larger RW parameter can make a difference (Nahmani et al., 2019). I think the choice of this parameter is still open to discussion. Don't you think that a larger value may lead to an overall increase in the gradient variability and improve the consistency between the GPS and WVR results discussed in Section 5.2? This might be suggested along with the sentence P24L24-25 ("We conclude that the constraints and the sampling … but cannot based on these results determine their relative importance.")

The quality of the ECWMF gradient product used in this study should still be better stated in Section 3.4 and mentioned when interpreting results in Section 4.2.

Section 3.4: suggested rewriting/reorganisation:

"The Technical University of Vienna provides hydrostatic and wet gradients based on ECMWF data for many space geodetic sites globally. The product used here is usually referred to as LHG (linear horizontal gradients) and is described by Böhm and Schuh (2007). It is available during certain time periods from the mid of 2005 and is more continuous from 2006. It is computed from profiles of hydrostatic and wet refractivity with a temporal resolution of 6 h, and a spatial resolution of 0.25 (30 km). The profile closest to the site is used together with one profile to the east and one profile to the north to calculate the refractivity gradient profiles. These are thereafter integrated to give the delay gradients. Because it was observed that on average the gradients computed in this way overestimate the more accurate gradients estimated from slant profiles, they are scaled by empirically derived factors, 0.53 for the hydrostatic gradients and 0.71 for the wet gradients (Böhm and Schuh, 2007). This computation method and rescaling provide gradient estimates of limited accuracy but they still represent valuable and independent source of information which are used here for comparisons with estimated GPS gradients.

There are alternative methods to derive gradients from Numerical Weather Model data using ray tracing methods, see e.g. (Zus et al., 2015) and references therein. More recently the Technical University of Vienna also introduced a new gradient product based on a least-squares adjustment of the ERA-Interim analyses (Landskron and Böhm, 2018).

In this study we used the LHG data from 2006 to 2016, resulting in a time series of 11 years. As an introduction, examples of the ECMWF hydrostatic and wet gradients are illustrated in Figure 8.

Worth noting is that the wet gradients dominate for the temporal resolution of 6 h and vary with the season, whereas the wet and the hydrostatic gradients show similar standard deviations (SD) for the monthly averages."

Section 4.2

P18L18-20: "When comparing the two tables it is clear that there are differences in the mean values of up 0.2 mm. These differences are mainly in the east component whereas there are consistent negative values for the north component. The SD of the GPS gradients are larger than the ECMWF gradients themselves."

=> last sentence: "The SD of the GPS gradients are larger than the ECMWF gradients by a factor of 2."

P18L20-22: "We note that the empirical factors used when calculating the ECMWF gradients (see Section 3.4) may not be correct for these stations. A related possible explanation is that not all variations in the water vapour content are detected due to the poor spatial and temporal resolutions of the ECMWF data. "

Comment on the first sentence: You don't have any clue that the empirical factors are not correct for these stations. And the issue with the empirical factors was actually not discussed in Section 3.4 (you just mentioned the values of the factors). In the proposed reformulation given above the reason for the factors is given and the limited accuracy is mentioned.

Comment on the second sentence: the operational ECMWF model is a global model with rather high resolution for a global model, hence the "poor resolution" comment is not fair… Maybe this model is just not adapted to your application?

=> Suggested rewriting of L20-22: "The differences may be explained by at least two reasons. First, the ECMWF gradient data used here have some intrinsic shortcomings (see Section 3.4). Second, not all variations in the water vapour content observed by the GPS receivers are actually represented in the ECMWF model due to its rather coarse spatial and temporal resolutions (Bock and Parracho, 2019)."

Section 5.2

P24L12-16: the revision suggested in the previous review was not clear. It was suggested to delete "requiring rapid changes" and the last sentence ("This is valid argument…regardless of the value of the constraint"). These parts of text were crossed out in the previous review but maybe this didn't appear in your copy. So the suggested rewriting is:

"(2) The WVR gradients for one 15-minute period do not depend on earlier or later estimates whereas the GPS gradients are estimated using constraints on the variability. A constraint has a similar impact as a low-pass filter (peaks with a short duration will be reduced)."

Additional references:

Bock, O. and Parracho, A. C.: Consistency and representativeness of integrated water vapour from ground-based GPS observations and ERA-Interim reanalysis, Atmos. Chem. Phys. Discuss., https://doi.org/10.5194/acp-2019-28, in review, 2019.

Nahmani, S., Bock, O., and Guichard, F.: Sensitivity of GPS tropospheric estimates to mesoscale convective systems in West Africa, Atmos. Chem. Phys. Discuss., https://doi.org/10.5194/acp-2018-1242, in review, 2019.